# Global subsidence of river deltas

L. O. Ohenhen[1 ✉], M. Shirzaei[2,3], J. L. Davis[4], A. Tiwari[5], R. Nicholls[6,7], O. Dasho[2,3], N. Sadhasivam[2,3], K. Seeger[8,9], S. Werth[2,3], A. J. Chadwick[4], F. Onyike[2,3], J. Lucy[2,3], C. Atkins[2,3], S. Daramola[10], A. Ankamah[2], P. S. J. Minderhoud[9,11,12], J. Oelsmann[13] & G. C. Yemele[11]

River deltas sustain dense human populations, major economic centres and vital ecosystems worldwide[1,2]. Rising sea levels and subsiding land threaten the sustainability of these valuable landscapes with relative sea-level rise and associated flood, land loss and salinization hazards[1–3]. Despite these risks, vulnerability assessments are impeded by the lack of contemporary, high-resolution, delta-wide subsidence observations[4]. Here we present spatially variable surface-elevation changes across 40 global deltas using interferometric synthetic aperture radar. Using this dataset, we quantify delta surface-elevation loss and show the prevalence and severity of subsidence in river deltas worldwide. Our analysis of three key anthropogenic drivers of delta elevation changes shows that groundwater storage has the strongest relative influence on vertical land motion in 10 of the 40 deltas. The other deltas are either influenced by multiple drivers or dominated by sediment flux or urban expansion. Furthermore, we find that contemporary subsidence surpasses absolute (geocentric) sea-level rise as the dominant driver of relative sea-level rise for most deltas over the twenty-first century. These findings suggest the need for targeted interventions addressing subsidence as an immediate and localized challenge, in parallel with broader efforts to mitigate and adapt to climate change-driven global sea-level rise.

River deltas, which occupy only 1% of land area, are among the most vital landforms on Earth[1]. Globally, deltas host an estimated 350–500 million people (representing 4–6% of the global population), including 10 of the 34 megacities of the world[1–3]. These dynamic landforms serve important socioeconomic, ecological and energy-related functions[5,6]. They sustain agricultural productivity and fisheries, their ecosystems sustain important biodiversity and their infrastructure, such as ports and transportation networks, anchors maritime trade vital to national, regional and global economies[5–7].

This recognized importance, which makes deltas indispensable, also increases their exposure to compounding climatic, environmental and anthropogenic threats[2,8–10]. As low-lying landforms, with extensive areas less than 2 m above sea level[11], deltas are acutely susceptible to rising sea level, storm surge, land subsidence, shifting temperature and rainfall patterns, and other environmental pressures, which are amplified by climate change[2,3,7–10,12,13]. These pressures degrade agricultural land; disrupt freshwater availability; exacerbate coastal and fluvial flooding; promote wetland loss, saltwater intrusion and shoreline retreat; and threaten infrastructure in deltas[2,5,6,14,15]. Beyond direct physical impacts, the interplay of these hazards also creates potential cascading socioeconomic consequences. For example, land loss and freshwater scarcity may drive displacement and migration, heightening competition for dwindling resources and fuelling social tensions[16,17]. Together, these intersecting climatic, environmental, human-driven pressures and multi-hazards render deltas the most fragile landscapes on Earth, with their low elevation and high urban exposure placing them at the forefront of climate and environmental risks[3,5,9] (Extended Data Fig. 1).

Among these threats, land subsidence often emerges as an important contributor to risks in global river deltas[1–3,12,18,19]. This predominantly human-driven process is just as, or more, influential than climate-induced sea-level rise (SLR) in the twenty-first century[3,20,21], with subsidence control now providing an important component of future coastal adaptation strategies[22,23]. Despite its perceived importance, land subsidence remains underrepresented in global assessments of delta vulnerability[9,24] largely because of the lack of modern, high-resolution subsidence observations[4,13]. Even with recent advances in space-based geodetic monitoring, high-resolution synoptic measurements of subsidence rates remain scarce, as most observations remain restricted to main urban centres within deltas, neglecting rural and ecologically critical zones[4]. Understanding delta-wide spatial characteristics of contemporary land elevation changes is important for informing their sustainable management.

Here, we present high-spatial-resolution datasets of surface-elevation change derived from Sentinel-1 synthetic aperture radar (SAR) interferometry across 40 deltas globally (Fig. 1). These datasets capture delta-wide temporal trends, subsidence rates and horizontal motion at 75 m resolution, spanning five continents and 29 countries. Our analysis encompasses all major river deltas with a population exceeding

[1]Department of Earth System Science, University of California, Irvine, Irvine, CA, USA. [2]Department of Geosciences, Virginia Tech, Blacksburg, VA, USA. [3]Institute for Water, Environment and Health, United Nations University, Richmond Hill, Ontario, Canada. [4]Lamont-Doherty Earth Observatory, Columbia University, New York, NY, USA. [5]Texas A&M AgriLife Research Center, Corpus Christi, TX, USA. [6]Tyndall Centre for Climate Change Research, University of East Anglia, Norwich, UK. [7]School of Engineering, University of Southampton, Southampton, UK. [8]Institute of Geography, University of Cologne, Cologne, Germany. [9]Soil Geography and Landscape Group, Wageningen University and Research, Wageningen, The Netherlands. [10]Department of Civil and Environmental Engineering, Virginia Tech, Blacksburg, VA, USA. [11]Department of Civil, Environmental and Architectural Engineering, University of Padova, Padova, Italy. [12]Department of Groundwater and Water Security, Deltares Research Institute, Utrecht, The Netherlands. [13]Department of River-Coastal Science and Engineering, Tulane University, New Orleans, LA, USA. ✉e-mail: oohenhen@uci.edu

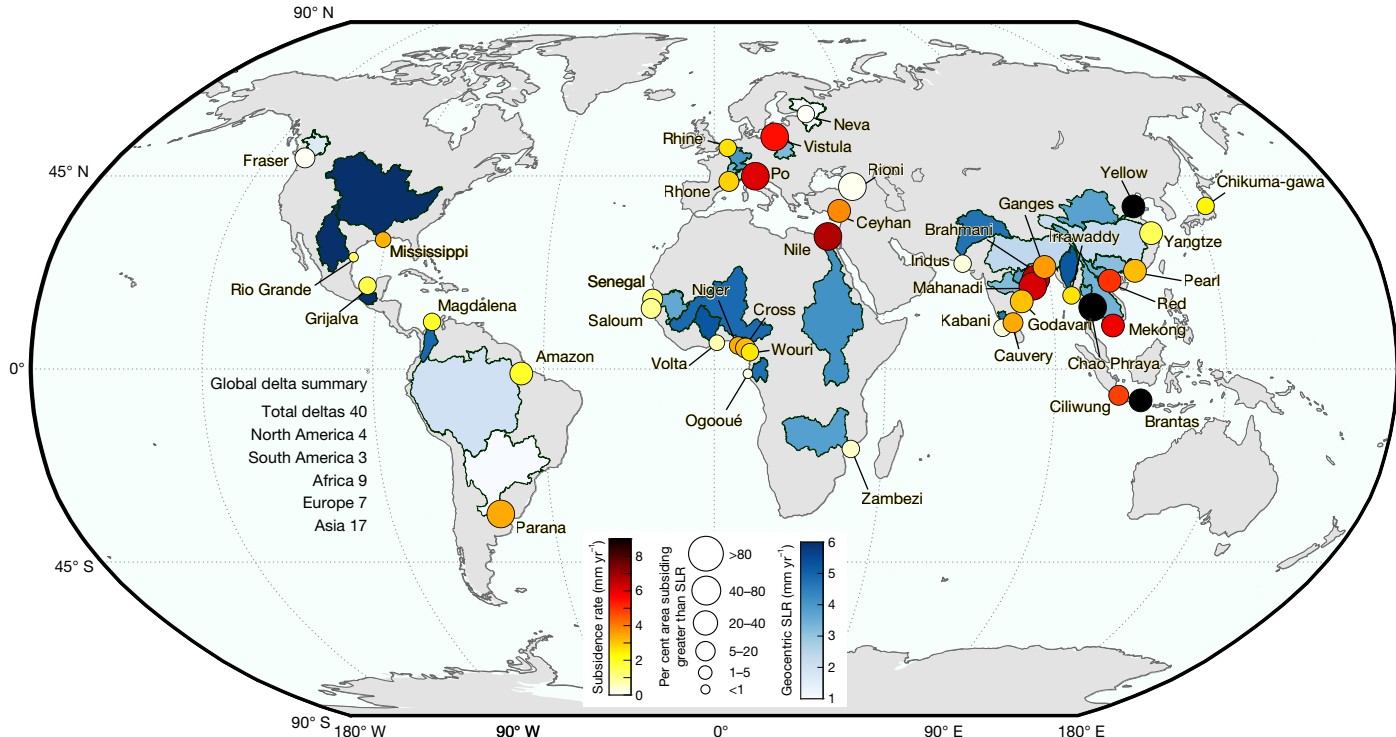

**Fig. 1 | Land subsidence in global deltas.** Each circle represents the location of the 40 deltas evaluated in this study, colour-coded by the average land subsidence rate. The size of the circle represents the percentage of the delta area subsiding faster than geocentric SLR. For visualization purposes, the geocentric SLR rate is shown as a colour gradient over entire watersheds or basins, although this does not represent the actual extent of exposure. Global coastlines are based on public-domain data from the CIA World DataBank II (using GSHHG (Global Self-consistent, Hierarchical, High-resolution Geography Database)), distributed with MATLAB. The delta basin polygons were obtained along with the sediment flux dataset from ref. 29.

3 million people[4], historically recognized sinking deltas[2] and representatives of less-populated, understudied deltas of regional ecological and economic importance (Methods).

## Global analysis of delta subsidence

We measured the spatial patterns and rates of subsidence in 40 deltas by analysing the complete archive of the Sentinel-1 SAR dataset between 2014 and 2023 using advanced multitemporal interferometric SAR (InSAR) analysis (Methods). InSAR measures surface-elevation changes, capturing vertical land motion (VLM), sediment deposition and erosional processes[13,25]. For consistency, to reflect both VLM and surface-elevation change in the deltas, we use the terms VLM or elevation gain or loss to describe net surface-elevation change across all delta environments, with positive values indicating uplift or elevation gain and negative values indicating subsidence or net elevation loss. Throughout this study, negative VLM is quoted with negative signs and references land subsidence rates, whereas only the absolute values are reported when presenting subsidence rates.

Our analysis shows that subsidence threatens deltas globally, with the delta-scale average rate of VLM on all deltas indicating subsidence (Fig. 1). In 12 out of 40 deltas, the average sinking rate is moderate, at less than 2 mm yr$^{-1}$. By contrast, more than half of the deltas exhibit subsidence rates exceeding 3 mm yr$^{-1}$, and in 13 of these deltas (Nile, Po, Vistula, Ceyhan, Brahmani, Mahanadi, Chao Phraya, Mekong, Red, Ciliwung, Brantas, Godavari and Yellow River), the average subsidence rates exceed the current estimates of global SLR (that is, about 4 mm yr$^{-1}$). Among these, the Chao Phraya (Thailand), Brantas (Indonesia) and Yellow River (China) deltas show an average sinking rate of more than twice the current global SLR rate. To further highlight the severity of subsidence in deltas, we compared the subsidence with the regional geocentric SLR rates for the twenty-first century (2001–present).

In 18 of the 40 deltas (the Nile, Po, Vistula, Ceyhan, Rioni, Brahmani, Mahanadi, Ganges–Brahmaputra, Godavari, Chao Phraya, Mekong, Red River, Ciliwung, Brantas, Amazon, Parana, Pearl and Yellow River), the average rate of local land subsidence is greater than the rate of regional geocentric SLR (Fig. 1 and Supplementary Table 1). However, in almost every delta (except Rio Grande) at least 1% of the delta area is subsiding faster than both global and geocentric sea levels (Fig. 1 and Supplementary Table 1).

Among all deltas, we find that at least 35% of the area is sinking, and in 38 deltas (excluding Neva and Fraser), more than 50% of the delta area is sinking (Fig. 2a). Of the 40 deltas, 19 show widespread subsidence patterns, with greater than 90% of the delta area affected by subsidence (for example, Mississippi, Niger, Nile, Rhine–Meuse, Po, Vistula, Brahmani, Mahanadi, Ganges–Brahmaputra, Chao Phraya, Mekong and Brantas deltas). Deltas with notable subsiding areas with greater than 50% of the delta area sinking faster than 5 mm yr$^{-1}$ include the Chao Phraya (94% of delta area), Nile (80%), Brahmani (77%), Po (74%), Mahanadi (69%), Brantas (66%), Vistula (57%), Yellow River (53%) and Mekong (51%) deltas (Fig. 2a and Supplementary Table 1). In sum, we estimate that a total delta area of 460,370 km$^2$ is exposed to subsidence. If we consider a global habitable geomorphic area of 710,000–855,000 km$^2$ for deltas[6,26], approximately 54–65% of global delta areas are sinking just from the analysis of the 40 deltas. By region, South Asia, East Asia and Southeast Asia, with 17 representative deltas, have the greatest exposure to subsidence, with 274,000 km$^2$ of delta area subsiding. Africa, South America, North America and Europe have total subsiding delta areas of 78,800 km$^2$, 39,800 km$^2$, 37,800 km$^2$, and 30,000 km$^2$, respectively. Seven large deltas—Ganges–Brahmaputra, Nile, Mekong, Yangtze, Amazon, Irrawaddy and Mississippi deltas—contribute about 57% of the total subsiding delta area, with a combined area of 265,000 km$^2$. Coastal cities such as Alexandria (Nile), Bangkok (Chao Phraya), Dhaka and Kolkata (Ganges–Brahmaputra), Shanghai

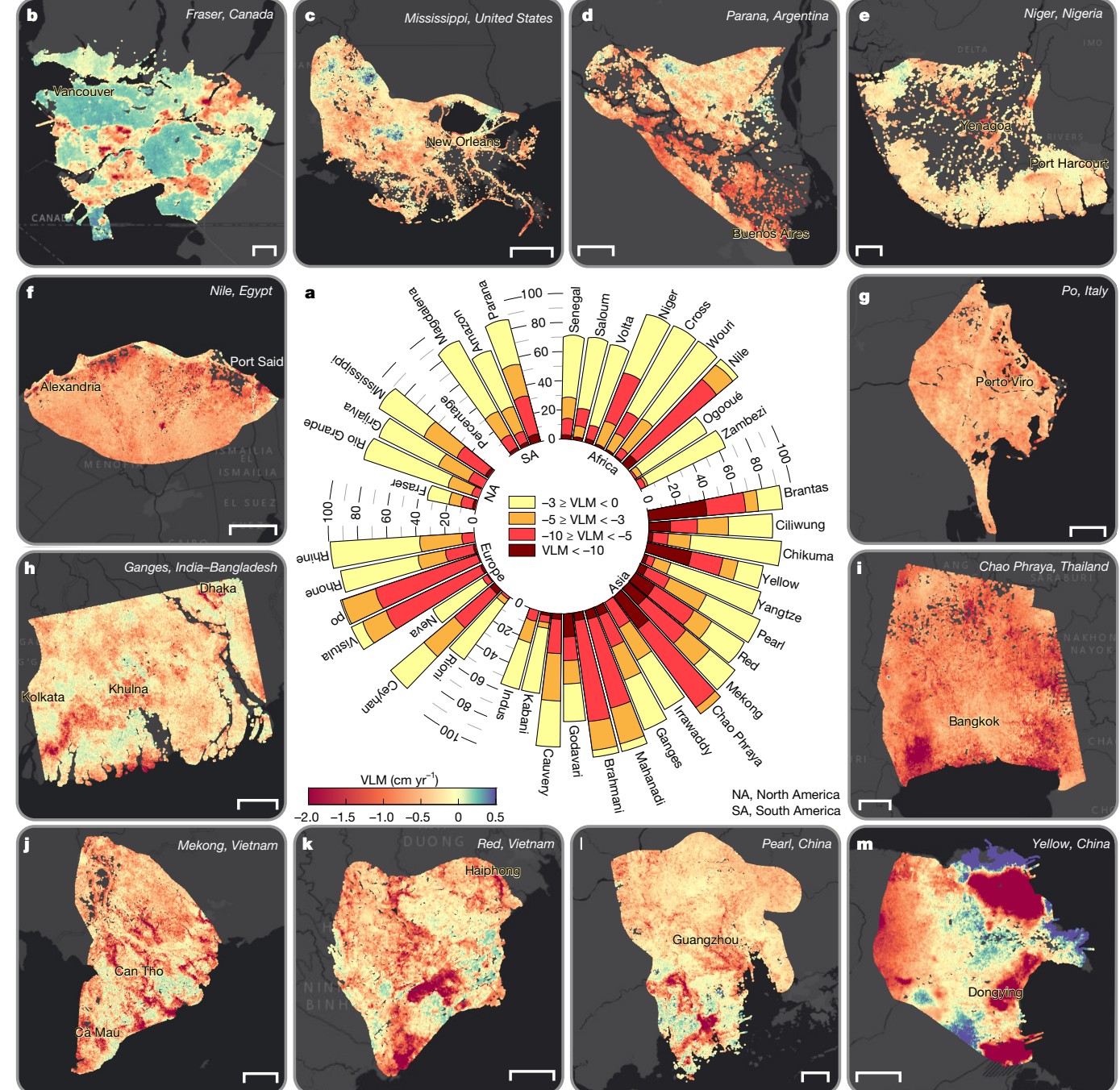

**Fig. 2 | Spatial pattern of VLM across global deltas. a**, Proportion of each delta exposed to different rates of subsidence. Note that only subsiding areas are represented in each bar, and areas of uplift within each delta are omitted to emphasize the extent of elevation loss. **b**–**m**, Spatial maps of VLM rates for the Fraser (Canada) (**b**), Mississippi (the USA) (**c**), Parana (Argentina) (**d**), Niger (Nigeria) (**e**), Nile (Egypt) (**f**), Po (Italy) (**g**), Ganges–Brahmaputra (India–Bangladesh) (**h**), Chao Phraya (Thailand) (**i**), Mekong (Vietnam) (**j**), Red River (Vietnam) (**k**), Pearl (China) (**l**), Yellow River (China) (**m**) deltas. Positive VLM (green–purple hues) suggests uplift or elevation gain, whereas negative VLM (yellow–orange–red hues) indicates land subsidence. The spatial VLM maps for the other 28 deltas are shown in Extended Data Figs. 2–4. Background image in **b**–**m** is Esri, streets-dark. Scale bars, 5 km (**b**); 50 km (**c**,**f**,**h**,**j**); 20 km (**d**,**e**,**i**,**k**,**l**,**m**); 10 km (**g**).

(Yangtze), Yangon (Irrawaddy), Cần Thá (Mekong), Thái Bình (Red River), Niigata (Chikuma-gawa), Jakarta (Ciliwung), Surabaya (Brantas) and Dongying (Yellow River) are experiencing subsidence at rates equal to or exceeding the delta-wide averages, indicative of the intensity of subsidence and elevation loss processes in cities on deltas.

Furthermore, we observe non-uniform spatially variable VLM within individual deltas, reflecting the complex interplay of natural and anthropogenic processes[2,5,13,27] (Fig. 2 and Extended Data Figs. 2–4). Although

all deltas exhibit an overall trend of subsidence, localized and broad zones of uplift, which vary from 0 mm yr⁻¹ to greater than 5 mm yr⁻¹ are observed in some areas (Fig. 2b,d,k,m, and Extended Data Figs. 2e,f,i,j,l and 3c,f). In some deltas (for example, Wouri, Zambezi, Indus, Ciliwung and Yellow River), the observed uplift or elevation-gaining parts correlate with patterns of horizontal land motion (Extended Data Figs. 5–7). Possible mechanisms may include sediment redistribution processes potentially driven by river dynamics or growth faulting, either of which

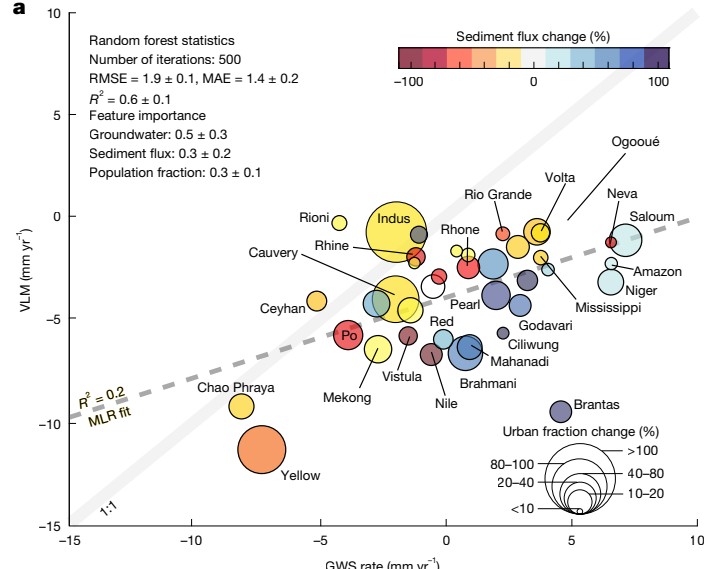

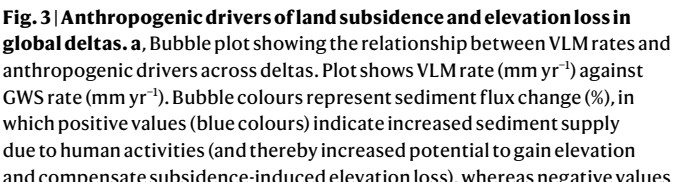

**Fig. 3 | Anthropogenic drivers of land subsidence and elevation loss in global deltas. a**, Bubble plot showing the relationship between VLM rates and anthropogenic drivers across deltas. Plot shows VLM rate (mm yr⁻¹) against GWS rate (mm yr⁻¹). Bubble colours represent sediment flux change (%), in which positive values (blue colours) indicate increased sediment supply due to human activities (and thereby increased potential to gain elevation and compensate subsidence-induced elevation loss), whereas negative values

can cause localized zones of elevation gain even within a predominantly subsiding deltaic system[28,29]. This highlights the necessity of comprehensive assessments and models of delta vulnerability to consider not only overall absolute subsidence rates but also the spatial heterogeneity of elevation change dynamics.

## Anthropogenic drivers of delta subsidence

All deltas, by their inherent nature, subside over time as recently deposited sediments or in situ organic material compact under their weight[30–32], a process further influenced by isostatic adjustments and tectonic activity[13,27]. However, human interventions have accelerated subsidence rates in many of the major deltas of the world, transforming a gradual geological process into an urgent environmental crisis[4,20,32]. The primary anthropogenic drivers that dominate delta subsidence include excessive groundwater extraction, oil and gas exploitation, and land-use changes associated with urbanization and agriculture[4,6,13,20,33,34].

To quantify the relative contributions of anthropogenic factors to delta subsidence and elevation loss, we analysed the relationship between three main anthropogenic drivers—groundwater storage change, sediment flux alteration and urban expansion—and non-glacial isostatic adjustment VLM/subsidence rates across the 40 deltas (Methods and Supplementary Table 2).

Figure 3a and Extended Data Fig. 8 show the interplay of anthropogenic factors and their correlation with subsidence rates across the 40 deltas. Deltas experiencing groundwater storage (GWS) loss (indicative of groundwater extraction), negative sediment flux change (red and yellow hues; reflecting sediment reduction due to upstream human activities) and higher urban population growth tend to have higher rates of subsidence (for example, the Yellow River, Po, Nile, Chao Phraya and Mekong deltas). Conversely, deltas with GWS stability or gain (net increase in groundwater storage), positive sediment flux change (blue colours; sediment surplus) and limited urban expansion show lower subsidence rates (for example, Saloum, Amazon and Ogooué deltas).

The initial multilinear regression (MLR) model, which included interaction terms between the different anthropogenic factors,

(yellow–orange–red colours) indicate a decline in sediment availability. Bubble size indicates urban fraction change (%), with larger circles representing a greater urban expansion over the twenty-first century. The dashed line represents the MLR fit. See Extended Data Fig. 8 for individual pairwise relationships between each anthropogenic driver and VLM. **b**, Ternary plot of subsidence rates with nLIME scores.

poorly captured subsidence dynamics on the deltas ($R^2 = 0.2 \pm 0.1$), as it failed to account for nonlinear interactions between the different processes (Fig. 3a). For instance, urban expansion not only directly increases infrastructure loading but also indirectly elevates groundwater demand, thereby compounding aquifer depletion and extraction-induced subsidence, which are synergistic effects that linear models cannot resolve.

To address these limitations, we used a random forest (RF) machine learning approach designed to capture nonlinear relationships and variable interactions. The RF model shows a moderate to strong relationship between the predictors (GWS, sediment flux and urban expansion) and VLM, achieving improved performance over the MLR model ($R^2 = 0.6 \pm 0.1$; RMSE (root mean square error) = $1.9 \pm 0.1$ mm yr⁻¹; MAE (mean absolute error) = $1.4 \pm 0.2$ mm yr⁻¹), and capturing complex, non-additive relationships between anthropogenic stressors and subsidence rates (Fig. 3a and Supplementary Fig. 1). However, we observe some underestimation at high subsidence rates (>8.0 mm yr⁻¹) (Supplementary Fig. 1), which probably suggests that natural processes or other anthropogenic predictors (not considered in our analysis) may contribute to subsidence in these highly dynamic deltaic environments.

Note that the primary objective in our analysis is not to predict subsidence rates across deltas, but rather to identify and extract key features that explain the dynamic relationships between the three anthropogenic drivers and subsidence across these deltas. Feature importance analysis from the RF model identifies GWS as the dominant anthropogenic predictor of delta subsidence ($0.5 \pm 0.2$), whereas sediment flux change ($0.3 \pm 0.2$) and urbanization ($0.3 \pm 0.1$) have secondary roles as subsidence rate predictors across these deltas (Fig. 3a and Supplementary Fig. 1b). However, the large standard deviations in feature importance values reflect substantial variability in predictor dominance across subsampled delta subsets, suggesting that the primary contributors to subsidence differ locally depending on the anthropogenic or geomorphic context. To resolve delta-specific mechanisms, we applied local interpretable model-agnostic explanations (LIME), which interprets individual predictions by approximating the RF model locally with simpler, interpretable functions. Deltas with

low LIME model fidelity ($R^2 < 0.5$) were excluded from this interpretative analysis, refining the dataset from 40 to 28 deltas (Methods). The low fidelity scores for some deltas could be due to unaccounted processes (natural and/or other anthropogenic) in our RF model. The retained 28 deltas show improved overall model performance ($R^2 = 0.7 \pm 0.1$; RMSE = $0.4 \pm 0.1$ mm yr$^{-1}$; MAE = 0.3 mm yr$^{-1}$), ensuring reliable interpretation of local feature importance. Normalized LIME feature importance scores (nLIME) showed substantial heterogeneity in predictor dominance (Supplementary Table 2). GWS emerged as the most significant factor across the different deltas ($0.6 \pm 0.3$), whereas sediment flux change ($0.3 \pm 0.1$) and urbanization ($0.1 \pm 0.1$) exhibited lower but context-dependent impacts (Supplementary Fig. 1b).

To assess the dominant influence on land motion across individual deltas, the nLIME for each delta was mapped onto a ternary diagram (Fig. 3b). Of the 28 deltas, 35%, including the Mekong, Ganges–Brahmaputra, Rhine–Meuse, Fraser, Cauvery, Irrawaddy and Red River systems, cluster within the GWS portion of the diagram (nLIME$_{GWS} > 0.7$), suggesting that observed GWS changes in these deltas are the primary driver of subsidence among the three anthropogenic variables examined (Fig. 3b and Supplementary Table 2). The Chao Phraya and Yellow River deltas, with the highest average subsidence rates, plot near the centre of the ternary diagram, reflecting relatively balanced contributions from GWS, sediment flux and urban expansion. Sediment flux correlates most closely with elevation changes in deltaic systems, such as the Saloum, Mississippi, Amazon and Rio Grande deltas, suggesting that reduced sediment delivery may exacerbate land elevation loss in these deltas. The Nile, Po, Chikuma-gawa, Mahanadi, Kabani, Niger and Volta deltas exhibit mixed contributions from GWS, sediment flux changes and population change, with GWS slightly outweighing sediment deficits as predictors in the Nile and Po deltas, possibly reflecting reliance on aquifer-dependent irrigation[35]. These findings are consistent with delta-specific studies that attribute accelerated subsidence in densely populated Asian deltas—Mekong, Ganges–Brahmaputra and Chao Phraya—to urbanization and unsustainable groundwater extraction for agriculture, industry and domestic use[6,20,31,32,36]. Moreover, the Nile, Po and Mississippi deltas, which were historically sustained by seasonal floods that deposited sediments, are now documented to experience severe sediment deficits due to dams and levees, accelerating elevation loss[2,20,29].

We acknowledge several limitations. First, GRACE-derived GWS trends (spatial resolution of about 300–400 km) may introduce signal leakage from adjacent basins, particularly affecting smaller deltas. Second, the sediment flux dataset represents percentage changes between pristine and disturbed conditions rather than contemporary absolute rates, potentially masking recent trends. Third, other natural VLM processes (sediment compaction and tectonics) and anthropogenic drivers (hydrocarbon extraction and peat drainage) are not explicitly separated. Fourth, RF model results are inherently dependent on input variable distributions and should be interpreted within the context of these datasets. Last, although the 40 deltas represent a substantial portion of global delta area and population, they are not globally representative. Nevertheless, our analysis focuses on understanding the relative influence of three key anthropogenic variables across these diverse systems rather than providing delta-specific VLM budgets. Future studies incorporating spatially dense, delta-specific datasets will better resolve local-scale processes within individual deltas and enable rigorous partitioning of anthropogenic compared with natural contributions to land motion and elevation change.

## Relative impacts of SLR and subsidence

Globally, deltas face a 'double burden' of climate-induced SLR and sinking land, which together drive relative sea-level rise (RSLR) at rates exceeding global averages[2,3,7,8,18]. Unlike SLR, which reflects global-scale processes and progresses at a relatively uniform rate globally[7,37], subsidence operates at local to regional scales, is highly variable and reflects localized natural and human processes[13,27,30]. In many deltas, contemporary rates of subsidence may surpass the current SLR rates[2,14] (see previous section), creating a compound hazard in which RSLR is dominated not by climate-induced changes in sea surface height but by VLM.

To quantify the contributions of SLR and land subsidence in deltas, we evaluated their relative impact on the exposed delta populations. Our analysis shows that current average subsidence rates exceed geocentric SLR in 18 of the 40 deltas, including the Nile, Mekong, Red River, Ganges–Brahmaputra, Brahmani, Mahanadi, Chao Phraya, Ciliwung, Brantas and Yellow River deltas, affecting approximately 236 million people—a population about 50% larger than those residing in deltas in which the current rates of geocentric SLR outpace the subsidence rates (156.9 million) (Fig. 4a). This disparity is particularly pronounced for vulnerable populations occupying land below 1 m elevation[11]. In these lowest elevation areas, subsidence dominates the contribution to RSLR in about two-thirds of the deltas, including Amazon, Fraser, Niger, Rhone, Vistula, Ganges–Brahmaputra, Mekong, Red River, Pearl, Yangtze and Godavari deltas (Fig. 4b). Of the 76 million people living in delta areas with an elevation below 1 m, 84% (63.7 million people) reside in rapidly sinking areas of the deltas (Fig. 4b). These observations are striking, revealing the current dominance of subsidence over geocentric SLR in global deltas. Moreover, the spatial heterogeneity of VLM creates localized extreme rates of subsidence within deltas, further exacerbating their vulnerability. Under the current trajectory, moderate emission scenarios (shared socioeconomic pathway 2-4.5 (SSP2-4.5)), current maximum subsidence rates in the deltas already surpass projected twenty-first-century SLR rates (no VLM)[38]. Through the end of the twenty-first century, current maximum subsidence rates in all 40 deltas exceed projected SLR rates (Fig. 4c). This disparity extends to the 95th percentile subsidence rates, representing widespread, high-magnitude sinking across the deltas. In 29 deltas, 95th percentile subsidence rates exceed the projected SLR rates by 2050, outpacing SLR by 1.1 (Niger delta) to 10.3 (Yellow River delta) times. By 2100, as the current maximum rate of SLR (SSP2-4.5) accelerates to 0.9 cm yr$^{-1}$, current 95th percentile subsidence rates still dominate in 22 deltas, surpassing geocentric SLR by up to seven times. Even accounting for worst-case, high-emission scenarios (SSP5-8.5), subsidence will exceed projected SLR rates in all deltas (considering maximum subsidence) and in 23 deltas (considering 95th percentile subsidence) through 2050. By 2100, current maximum subsidence rates exceed projected SLR in 38 of 40 deltas, whereas 95th percentile subsidence rates remain dominant in seven deltas (Godavari, Chao Phraya, Mekong, Ciliwung, Brantas, Red River and Yellow River) (Supplementary Table 1).

These findings identify VLM as the principal hazard in deltaic systems and other subsidence-prone low-elevation coastal zones. Although global coastal zones face baseline threats from SLR[38], subsidence in many deltas often dominates RSLR, creating a distinct and more acute risk profile, which is amplified by the high populations in many of these deltas[4,12]. Yet, subsidence remains underprioritized in global coastal risk discourse, a tendency that stems from its perceived tractability. Unlike climate-induced SLR, which can be slowed but not stopped on human policy time scales, human-induced subsidence can theoretically be slowed or halted through targeted interventions[22,23,31,32]. Its responsiveness to human action, however, has paradoxically relegated it to the periphery of international policy[3,31,39]. This disconnect reflects a broader misalignment between the spatial scales of climate impacts and adaptation priorities. Thus, subsidence does not merely compound SLR; it undermines the foundational logic of incremental, SLR-centric adaptation[39]. Addressing this requires shifting adaptation from just a global climate challenge to a regional socio-technical imperative and an integrated approach that prioritizes subsidence mitigation

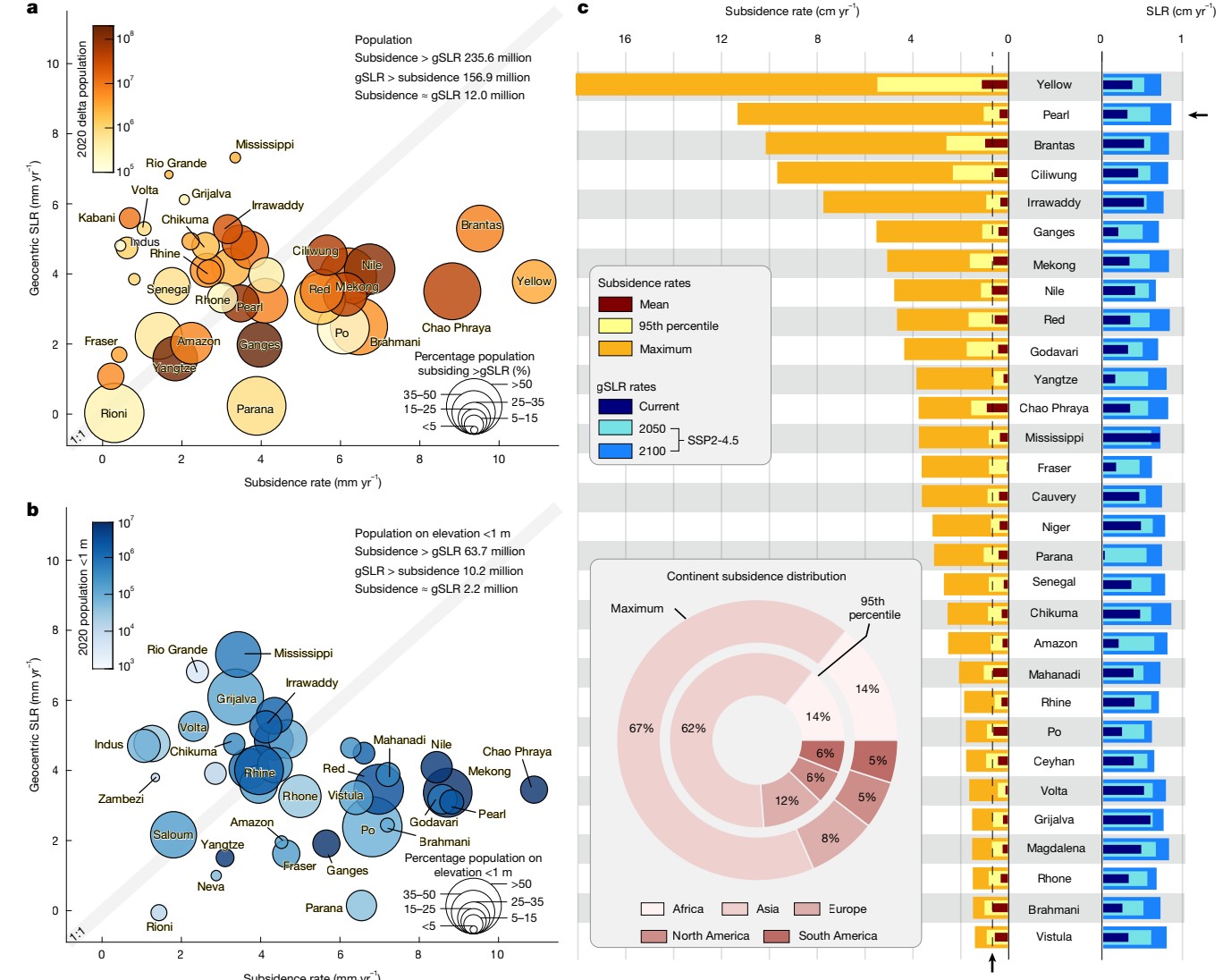

**Fig. 4 | Relative contributions of land subsidence and SLR in global deltas.**
**a**, Bubble plot comparing geocentric (absolute) SLR (mm yr$^{-1}$) and land subsidence (mm yr$^{-1}$) across 40 deltas. Deltas in which subsidence rates exceed geocentric SLR fall to the right of the 1:1 line, whereas those in which geocentric SLR exceeds subsidence fall to the left. Bubble colours indicate the total delta population, ranging from fewer than 100,000 (lighter colours) to more than 100 million (darker colours). Bubble size represents the percentage population living in delta land areas subsiding faster than geocentric SLR. **b**, Same as **a**,

but considering only the population living at elevations below 1 m. Note that the Brantas and Yellow River deltas have values greater than 15 mm yr$^{-1}$ and are not represented on the plot for visual clarity. **c**, Bar plots comparing the range of land subsidence rates, contemporary and projected SLR for 30 representative deltas. The maximum subsidence rate is calculated as the median of the 50 highest rates to avoid biases from single extreme values. The dashed vertical line shows the maximum 2100 projected SLR rate across all deltas.

(for example, groundwater regulation, managed aquifer recharge and sediment management) alongside RSLR adaptation.

## Adaptive capacity in vulnerable deltas

From the Fraser delta in Canada to the Yellow River delta in China, global deltas are sinking, as climate change accelerates SLR, compounding the vulnerabilities of low-lying regions. These combined effects create a multifaceted threat, forcing delta communities to contend with land loss, more frequent flooding and saltwater intrusion[6–9,20]. Whereas the urgency of adaptation is immediate and worldwide, the capacity to act is not. For many deltas, especially those in low- and middle-income countries, adaptive capacity is limited by institutional, social and financial constraints[9]. These systemic barriers are quantified by the Notre Dame Global Adaptation Index (ND-GAIN), a framework that evaluates the vulnerability of countries to climate change and their readiness to

deploy adaptation resources across economic, social and governance dimensions[40,41]. A higher ND-GAIN adaptation readiness score (>0.52) is an indication of the capacity of a country to absorb funds and translate these into actionable strategies[41].

To visualize disparities in adaptive capacity and risk, we mapped global deltas into a two-dimensional (2D) impact matrix defined by RSLR and ND-GAIN adaptation readiness scores (Fig. 5). This framework allows for a comparative assessment of deltas assuming that the adaptation readiness of the delta is reflected by the adaptation readiness of its country, categorizing them into four quadrants: (1) Unprepared Divers (high RSLR (>4 mm yr$^{-1}$), low readiness (<0.52)); (2) Rising Ready (high RSLR, high readiness (>0.52)); (3) Latent Threats (low RSLR, low readiness); and (4) Safe Havens (low RSLR, high readiness). 65% of the deltas (26 out of 40 deltas), predominantly in low- and middle-income nations, fall into the Unprepared Divers group, in which nations have a diminished adaptive capacity and RSLR rates exceeding

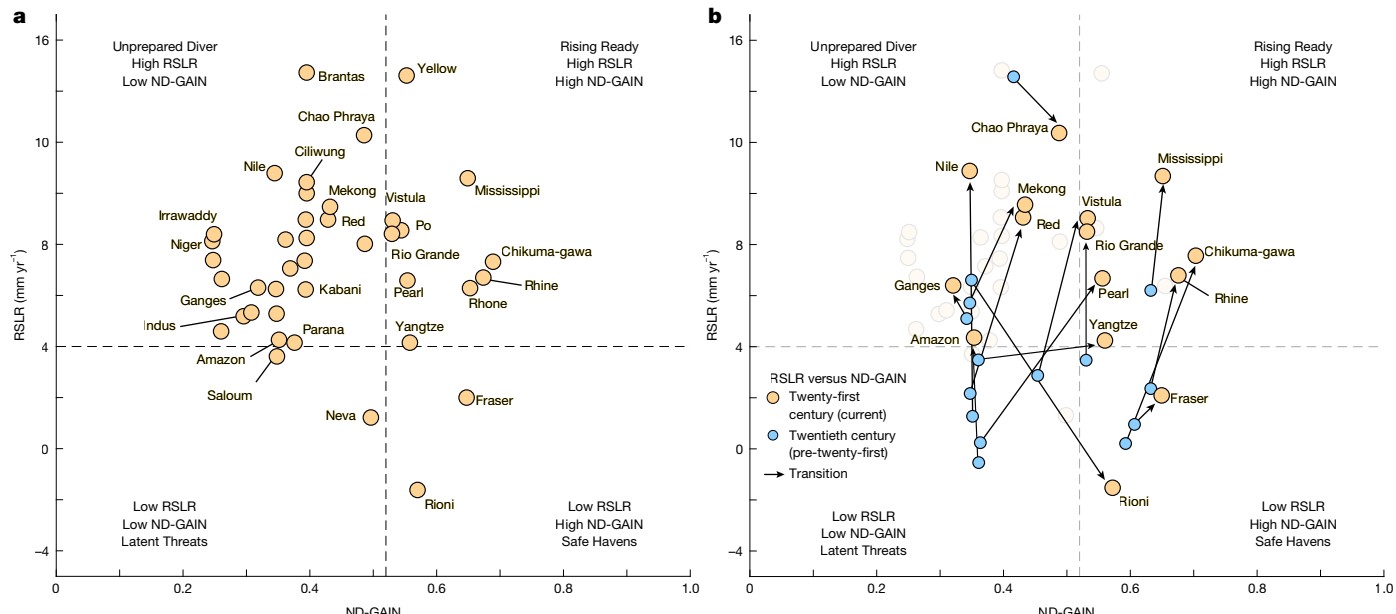

**Fig. 5 | RSLR and adaptive capacity in global deltas. a**, Scatter plot showing the relationship between RSLR and ND-GAIN adaptation readiness score for 40 deltas in the twenty-first century. The horizontal dashed line represents the current global SLR (about 4 mm yr⁻¹), whereas the vertical dashed line indicates the threshold between 'good' and 'very good' readiness categories (0.52), as defined in ref. 41. **b**, Same as **a** but including both twentieth- and twenty-first-century data for 15 deltas. Arrows illustrate the trajectory of the readiness score of each delta from the twentieth century to the twenty-first century. The four quadrants represent Unprepared Divers (deltas with high RSLR, low adaptation readiness), Rising Ready (deltas with high RSLR, high adaptation readiness), Latent Threats (deltas with low RSLR, low adaptation readiness) and Safe Havens (deltas with low RSLR, high adaptation readiness).

current global SLR (Fig. 5a). These challenges are compounded for indigenous communities, who primarily live in the lowest-lying delta areas; lack the resources needed to implement large-scale adaptation; and face relocation barriers due to cultural and subsistence ties despite escalating risks[42,43].

Most deltas in high-income countries, including the Yellow River (China), Vistula (Poland), Po (Italy), Rhine–Meuse (the Netherlands) and Mississippi (the USA) deltas, cluster in the Rising Ready group, demonstrating robust governance (Fig. 5a). For example, the integrated flood management approach of the Dutch delta, which combines ecological restoration with infrastructural fortifications, has become a model for coastal hazard resilience[44]. However, some deltas even within this group face substantial gaps. For instance, the Mississippi delta has lost more than 5,000 km² of land (mainly wetlands) since 1932 because of a lack of adaptation (for example, sediment diversion projects)[45,46], whereas the Po delta struggles with salinization driven by agricultural groundwater extraction, highlighting how economic priorities can undermine adaptation even in high-income regions[47]. Although RSLR exceeds global rates of SLR in most deltas, exceptions exist. The Latent Threats group includes the Saloum and Neva deltas, which exhibit relatively low RSLR and low adaptive capacity (Fig. 5a), indicating their unpreparedness and potential vulnerability to a future rise in sea level (Fig. 4c). The Rioni and the Fraser delta fall into the Safe Havens group, in which lower RSLR is coupled with higher adaptive capacity, indicative of low risk and preparedness for current and future sea-level changes. The Rioni Delta is the only delta in our sample exhibiting negative sea-level trends for the twenty-first century, in which long-term regional sea-level decline masks short-term fluctuations (Methods).

To examine the evolving risk landscape, we compared twentieth-century and present-day impact matrices (Fig. 5b). For our analysis, we used tide gauge data to estimate twentieth-century RSLR rates, which were available for only 15 of the 40 deltas. Our estimates show that 10 deltas previously classified as Latent Threats (low RSLR, low readiness) and Safe Havens (low RSLR, high readiness) groups during the twentieth century have transitioned to Unprepared Divers

(high RSLR, low readiness) and Rising Ready (high RSLR, high readiness) groups in the twenty-first century (Fig. 5b). This shift highlights the accelerating contemporary RSLR trends, driven by land subsidence and SLR[48,49]. Deltas such as the Mississippi, Ganges–Brahmaputra and Mekong show sustained increases in long-term RSLR rates above 4.0 mm yr⁻¹ since the twentieth century, exacerbating vulnerabilities in these densely populated regions. Conversely, the Chao Phraya and the Rioni deltas showed a decline in RSLR and improved adaptive capacity in the twentieth century. However, although the Rioni Delta exhibited a more than 200% decline in RSLR, the Chao Phraya Delta still experiences high RSLR rates (12.3 mm yr⁻¹). The pronounced decrease in RSLR for the twentieth century in the Rioni Delta probably reflects localized subsidence at the tide gauge station rather than a delta-wide RSLR trend[50] (Methods). The greatest change in RSLR was observed in the Nile Delta, surging from 1 mm yr⁻¹ in the twentieth century to more than 10 mm yr⁻¹ in the twenty-first century (Fig. 5b). Moreover, we find that all deltas in low- and middle-income countries in the present-day Unprepared Divers groups, transitioned from the Latent Threats group, suggesting stagnant adaptive capacity despite worsening RSLR. By contrast, deltas such as the Yangtze (China), Pearl (China) and Vistula (Poland) shifted from Latent Threats to Rising Ready, demonstrating increased adaptation readiness due to economic growth, raising governance and institutional capacity to adapt, although RSLR has surged (Fig. 5b). Although deltas in the Rising Ready quadrant showed potential for robust adaptation policies, deltas in the Unprepared Divers remain trapped in cycles of reactive, underfunded responses.

These long-term trajectories reveal a challenging reality in which deltas with strong adaptive capacity still struggle to manage persistent subsidence and climate-driven SLR, whereas those with limited capacity face severe and escalating risks on both fronts. Ideally, the goal for sustained coastal resilience is a transition to Safe Havens, characterized by both low RSLR and high adaptation readiness. However, only two deltas (the Fraser and Rioni) currently occupy this quadrant. As the climate crisis and related threats intensify, the challenge for the up to 500 million people in deltas demands more than incremental

adaptation; it requires global attention to subsidence and other key vulnerability drivers while advancing governance approaches that preserve land elevation and long-term habitability over short-term adaptation.

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

## Methods

### Selection of global river deltas

We selected 40 deltas globally, prioritizing 35 deltaic systems with the greatest exposed area and population currently below sea level, supplemented by five less-exposed deltas of local and regional significance and previously identified risks[9]. To assess the 35 deltas with the greatest exposure among global river deltas, we used 955 delineated delta boundaries in ref. 6 and identified coastal delta elevation below sea level using the DeltaDTM dataset v.1.1 (ref. 11) resampled to 3 arcseconds (100 m) and referenced to mean sea level[51]. Global delta population was estimated by aggregating 100 m resolution WorldPop population count for each delta, which is calibrated to the 2020 national population estimates from the United Nations population data[52].

Our estimates show that globally, 42,000 km² of the delta area at present lies below sea level, containing a population of 10.2 million people (Extended Data Fig. 1). The 35 deltas with the greatest exposure included in this analysis are Nile, Mississippi, Rhine–Meuse, Mekong, Niger, Cauvery, Po, Red River, Vistula, Rhone, Amazon, Ganges–Brahmaputra, Chao Phraya, Kabani, Pearl, Rio Grande, Yangtze, Yellow River, Senegal, Indus, Saloum, Grijalva, Ceyhan/Seyhan, Rioni, Cross, Chikuma-gawa, Volta, Brantas, Neva, Wouri, Irrawaddy, Ogooué, Zambezi, Magdalena and Ciliwung (Extended Data Fig. 1). The cumulative delta area and population below sea level are 38,000 km² and 10.1 million people, respectively, reaching within rounding errors of the global total exposure. Deltas such as the Danube, Orinoco and Shatt-el-Arab met the selection criteria but were excluded due to challenges associated with the SAR imaging and interferometric analysis (including spatial coverage gaps, excessive temporal baselines, poor coherence and limited data availability). The five supplementary deltas are Brahmani, Mahanadi, Godavari, Parana and Fraser deltas.

The final selection of 40 deltas spans five continents (Asia, Africa, Europe, North America and South America) and 29 countries, encompassing deltas with noted and emerging environmental, geophysical and social vulnerabilities[9,24], historically sinking river deltas[2] and densely populated coastal megacities[3,4,53].

### SAR dataset

We analysed 132 SAR frames from the Sentinel-1A/B C-band satellite, spanning September 2016 to May 2023. The SAR datasets include 3,300 images obtained in single-orbit geometry (ascending or descending) for 13 deltas and 10,700 images obtained in both ascending and descending orbits for 27 deltas. See Supplementary Table 3 for the complete inventory of SAR images used in each delta. For each SAR dataset, we applied a multi-looking factor of 32:6 (range:azimuth) to improve the signal-to-noise ratio, obtaining an average pixel resolution of about 75 m. To minimize decorrelation errors, we also constrained the interferometric pairs to a maximum temporal and perpendicular baselines of 300 days and 80 m, respectively. For deltas requiring multi-frame coverage (for example, Amazon, Mississippi, Mekong, Ganges–Brahmaputra, Nile, Red River and Niger), we arranged in a mosaic form the overlapping adjacent frames along a single path before processing or post-processed deltas with coverage spanning multiple paths to ensure full spatial continuity across expansive deltas.

### SAR interferometric analysis

We processed each SAR frame or single-path multiple-frame coverage to generate high-spatial resolution maps of surface deformation for the 40 deltas using a multitemporal wavelet-based InSAR (WabInSAR) algorithm[54–57]. First, we generated 59,000 high-quality interferograms from the coregistered SAR images using GAMMA software[58,59], with an interferogram pair selection algorithm[57] optimized through dyadic downsampling and Delaunay triangulation. To minimize phase errors and to maximize the pixel density associated with dynamic surface changes over deltas (for example, flooding, vegetation growth or soil saturation), we screened the initial set of interferograms based on their coherence stability to exclude interferograms with high coherence variability, while maintaining a 50% temporal baseline coverage. The final selection retained about 55,000 interferometric pairs (93%) for further analysis. Moreover, we implemented a statistical framework to discard noisy pixels with average coherence less than 0.7 for distributed scatterers and amplitude dispersion of greater than 0.35 for permanent scatterers[57]. Next, we used a minimum cost flow phase unwrapping algorithm optimized for sparse coherent pixels[60,61] to estimate the absolute phase changes of the elite (less noisy) pixels in each interferogram. We corrected all unwrapped interferograms for the effects of residual orbital error[62] and minimized the effects of topography-correlated components of atmospheric phase delay and spatially uncorrelated DEM error by applying a suite of wavelet-based filters[54]. Last, we estimated the time series, velocities and standard deviation for each geocoded elite pixel along the line of sight (LOS) of the satellite using a reweighted least-squares optimization[55]. The standard deviation of the LOS velocity corresponds to the uncertainty of the regression slope derived from the least-squares fit. For each delta, the reference point was selected as the pixel corresponding to a global navigation satellite systems (GNSS) station within the processed SAR frame when available. In areas without GNSS stations, a preliminary reference point was randomly selected from pixels with average temporal coherence >0.85. Following initial processing, the reference point was refined by visually identifying stable ground features (for example, bedrock outcrops and deep-foundation structures) and low displacement variability (standard deviation <1 mm yr⁻¹), then reprocessing with this final reference point. For large deltas requiring overlapping SAR frame coverage, the LOS velocities were arranged in a mosaic form to ensure seamless spatial representation across the entire delta.

In the 27 deltas with overlapping spatiotemporal SAR satellite coverage and different orbit geometries (ascending and descending), we estimate the horizontal (east–west) and VLM components of deformation by jointly inverting the LOS time series of the ascending and descending tracks[63–65]. To this end, we identified the co-located pixels of the LOS time series by resampling the pixels from the descending track onto the ascending track to obtain two co-located LOS displacement velocities $\{LOS_{ASC}, LOS_{DES}\}$. Given $\{LOS_{ASC}, LOS_{DES}\}$ and their associated variances $\{\sigma^2_{ASC}, \sigma^2_{DES}\}$ are the LOS displacement and variances for a given pixel, the model to combine the LOS velocities to generate a high-resolution map of the east–west ($E$) and VLM ($U$) displacements are given by

$$\begin{bmatrix} LOS_{ASC} \\ LOS_{DES} \end{bmatrix} = \begin{bmatrix} C^E_{ASC} & C^U_{ASC} \\ C^E_{DES} & C^U_{DES} \end{bmatrix} \begin{bmatrix} E \\ U \end{bmatrix} \tag{1}$$

where, $C$ represents the unit vectors for projecting ($E$) and ($U$) displacements onto the LOS, which is a function of the heading angle of the satellite and incidence angles of each pixel[66]. The solution to the model in equation (1) is given by

$$X = [G^T PG]^{-1} G^T PL \tag{2}$$

where $X$ represents the unknowns ($E$) and ($U$), $G$ is the design matrix comprising the unit vectors for projecting the horizontal and vertical displacements onto the line of sight, $L$ are the observations $\{LOS_{ASC}, LOS_{DES}\}$, and $P$ is the weight matrix, which is inversely proportional to the observant variances $\{\sigma^2_{ASC}, \sigma^2_{DES}\}$. To obtain the parameter variance–covariance matrix ($Q_{XX}$), we use the concept of error propagation[67] to calculate the associated parameter uncertainties given the observation errors as follows:

$$Q_{XX} = [G^T PG]^{-1} \tag{3}$$

For the 13 deltas imaged in single-orbit geometry (ascending or descending), we projected the LOS velocities to the vertical direction, assuming the principal deformation is vertical:

$$\text{VLM}_i = \frac{\text{LOS}_i}{\cos\theta_i} \qquad (4)$$

where, $\cos\theta_i$ is the local incidence angle for each pixel. This assumption of zero gradients in the horizontal components of deformation is tenuous for most coastal areas, given the significant localized horizontal motion noted (up to 10 mm yr$^{-1}$) across the 27 deltas with multiple orbit geometries. Nevertheless, the assumption is necessary given that overlapping ascending and descending orbit geometries are available for less than 50% of global land areas (for European Space Agency Sentinel-1 satellite), limiting the ability to resolve 2D deformation trends. However, under this assumption, it is necessary for the locally referenced VLM estimates to be transformed into a globally consistent reference frame, particularly for comparative studies across multiple regions[13,27].

To transform the VLM rates from a local to a global reference frame, we used the available GNSS datasets for 17 deltas (the Fraser, Mississippi, Rio Grande, Rhine–Meuse, Rhone, Po, Vistula, Red River, Amazon, Parana, Ciliwung, Brantas, Ganges–Brahmaputra, Chao Phraya, Mekong, Pearl and Chikuma-gawa). The GNSS datasets across the 17 deltas were obtained from the Nevada Geodetic Laboratory[68] and previous regional studies[69]. For each delta with GNSS coverage, we calculated the offset between the InSAR-derived vertical velocity at the reference point and the corresponding GNSS vertical velocity, then applied this offset to transform all InSAR velocities in that delta to the IGS14 reference frame. The uncertainty in the final velocity was estimated by propagating both the InSAR velocity uncertainty (from the reweighted least-squares inversion) and the GNSS velocity uncertainty (reported by data sources) through standard error propagation. In deltas without GNSS stations, we used the global VLM model[70], which mainly includes long-wavelength deformation signals due to TWS changes, tectonics and glacial isostatic adjustment (GIA) referenced to the IGS14 global frame. We then applied an affine transformation to align the VLM rates from local to IGS14 global reference frame[23,71]. This approach ensures consistency in VLM rates across global deltas by correcting for local reference biases and should be the standard practice in coastal research using InSAR[27]. When comparing these measurements to other subsidence rate estimation techniques in deltas, such as RSET, marker horizons, sediment cores, repeat lidar or other InSAR measurements, careful consideration must be given to differences in both reference frames and temporal ranges. Reference frame incompatibility may require adjustments to align local or relative measurements with other datasets, whereas mismatches in monitoring periods introduce temporal biases that complicate direct quantitative comparisons.

The distribution of the standard deviations (precision of the results) for all pixels (20.5 million) across the 40 deltas is shown in Supplementary Fig. 2. The standard deviation distribution shows that 99% of the pixels have a value <0.5 mm yr$^{-1}$. We evaluated the accuracy of the results by comparing the averaged VLM rates of pixels within a radius of 100 m with more than 100 independent GNSS data (that is, stations that were not used in the reference frame transformation). The validation included 122 GNSS stations across 23 deltas with historical long-term records (spanning various periods before and/or including the InSAR observation window) and 81 GNSS stations across 15 deltas with time series covering at least 70% of the InSAR observation period (2014–2023) (Supplementary Fig. 3). We found a strong correlation (0.7–0.8), between GNSS and InSAR velocities, with an RMSE of 1.4 mm yr$^{-1}$ for long-term rates (Supplementary Fig. 3a) and 1.2 mm yr$^{-1}$ for rates within the InSAR observation period (Supplementary Fig. 3b). The improved agreement for temporally coincident measurements suggests that nonlinear subsidence behaviour contributes to some scatter when comparing historical GNSS rates to contemporary InSAR measurements, although the overall correlation remains strong in both cases. Note that some GNSS stations used for validation, while within the broader processed SAR frame, are outside the clipped delta boundaries. Note that the final delta extents were delineated using a tiered approach. Primary boundaries were derived from ref. 9, supplemented by ref. 6 for deltas not covered in the former. For extensive deltas in which the entire delta surface is not analysed (for example, the Ganges–Brahmaputra), boundaries were defined using the SAR spatial extent.

## GIA influence on VLM
We estimated VLM trends and the associated uncertainty due to GIA using the model in ref. 72, which was derived from a probabilistic ensemble of 128,000 GIA forward simulations. Each model solves the sea-level equation for a compressible, viscoelastic Maxwell Earth under late-Pleistocene ice-sheet loading, incorporating solid-Earth deformation, geoid change and rotational feedback. The ensemble samples a wide range of Earth rheological structures, including lithospheric thickness, upper and lower mantle viscosities, and scaling factors applied to regional deglaciation histories over the past 122,000 years. Likelihoods were assigned to each simulation based on fit to a global dataset of 11,451 relative sea-level records and 459 GNSS-derived uplift rates using a Bayesian framework that accounts for data uncertainties and spatial correlations. The resulting posterior distributions enable spatially resolved estimates of GIA-driven VLM with formal uncertainty.

For each delta, we extracted the ensemble mean and standard deviation in GIA vertical velocity to correct observed deformation rates and isolate contemporary, non-GIA contributions to VLM. Supplementary Table 1 shows the mean GIA-induced VLM, the associated standard deviation and the per cent contribution of GIA to the total observed VLM magnitude for each delta. GIA accounts for the largest proportion and exceeds (>100%) the total VLM in the Neva (540%) and Fraser (455%) deltas, in which low observed VLM rates are substantially influenced by strong GIA uplift. Moderate GIA contributions (25–55%) are observed in five deltas, including the Rio Grande, Mississippi, Volta, Rhine and Ogooué deltas. Most of the deltas (55%) exhibit minimal GIA influence, with contributions under 10%, indicating that observed VLM is primarily governed by contemporary anthropogenic and natural processes such as groundwater withdrawal, sediment compaction, or tectonics. In 28–67% (accounting for uncertainty) of the deltas, the sign and approximate magnitude of observed and GIA-corrected VLM are consistent, implying limited distortion from GIA and the sustained expression of contemporary processes on the average local subsidence. By contrast, the Fraser and Neva deltas illustrate how substantial GIA-induced uplift in high-latitude, post-glacial regions can obscure contemporary subsidence processes through opposing vertical trends. In both cases, modest observed subsidence rates (Fraser −0.4 mm yr$^{-1}$ and Neva −0.2 mm yr$^{-1}$) are counteracted by substantial GIA uplift of 1.8 ± 2.3 mm yr$^{-1}$ and 1.0 ± 0.3 mm yr$^{-1}$, respectively.

## Anthropogenic drivers datasets
We analysed the relationship between major anthropogenic pressures on global deltas to subsidence and elevation loss by quantifying the contributions of groundwater storage change, sediment flux alteration and urban expansion to the residual rates of sinking (after GIA correction) across the 40 deltas. These globally consistent datasets provide insights into human-induced impacts on land subsidence and elevation change in river deltas (Supplementary Table 2).

**Groundwater storage change.** We derived twenty-first-century groundwater storage trends for all deltas by leveraging Gravity Recovery and Climate Experiment (GRACE) and GRACE Follow-On (GRACE-FO) satellite observations[73,74]. We used the JPL GRACE/GRACE-FO level 3 mascon solutions (RL06.3) (refs. 75,76), which provide monthly global estimates of total water storage (TWS) change

relative to a 2004.9–2009.999 mean baseline. The final solutions span 2002–present and are derived from solving for monthly gravity field variations in terms of 4,551 equal-area 3° spherical cap mass concentration functions rather than global spherical harmonic coefficients. The mascon approach implements geophysical constraints during the level-2 processing step to filter out noise, applies improved accelerometer data and standard corrections, including several geophysical adjustments, such as gravity anomaly due to ocean (GAD), GIA, degree-1, C20 and C30 replacement and representation on ellipsoidal earth[75–77]. We extracted TWS values at 3° mascon resolution (about 300–400 km spatial resolution) covering each delta area to compute representative regional water storage estimates. TWS change from GRACE contains contributions from GWS, soil moisture storage (SMS), snow water equivalent (SWE) and surface water storage (SWS) represented by

$$\Delta TWS = \Delta GWS + \Delta SWS + \Delta SMS + \Delta SWE \quad (5)$$

To isolate GWS change from TWS, we used the 1/4° global land data assimilation system Noah model[78] to remove changes in SMS and SWE contributions and used the WaterGAP Global Hydrology Model (WGHM v.2.2d) (refs. 79,80) to remove SWS contributions. The contribution from SWE was negligible in most deltas, given their prevailing arid and semi-arid climate (Fig. 1), although it was included to maintain consistency across all deltas. SWS components include contributions from rivers, lakes, wetlands and reservoir storage within the GRACE footprint for each delta. The residual signal following removal of SWS, SMS and SWE was interpreted as the GWS anomaly.

To estimate the temporal trend of groundwater storage changes, we applied harmonic analysis to account for annual and semiannual variations in the time series of the GWS anomalies. In standard practice, environmental variables (for example, GRACE data, GNSS data and sea-level anomalies) are modelled as time-invariant seasonal signals. However, the response of Earth to environmental changes represented as seasonal signals is not time-invariant[81–83]. To account for this variability, we adopted the stochastic-seasonal model in the following equation, in which the harmonic amplitudes evolve as random walks, allowing for time-dependent seasonal variations and the seasonal trends are modelled using a Kalman filter[83]:

$$x(t) = x_0 + v(t)(t - t_0)$$
$$+ \sum_{k=1}^{2} [a_k(t)\cos(2\pi k f(t - t_0)) + b_k(t)\sin(2\pi k f(t - t_0))] \quad (6)$$

where $t_0$ is the reference epoch, $x_0$ is the reference intercept at $t_0$, $v(t)$ is the time-varying rates, $k$ indexes the annual ($k = 1$) and semiannual ($k = 2$) components, $a_k$ and $b_k$ are the harmonic amplitudes. $v(t)$, $a_k$, and $b_k$ are modelled as random walk parameters. To estimate the long-term multi-year trend ($v_f$) of GWS from the time-varying rates, we computed the weighted average of the time-varying rates $v(t_i)$ using

$$v_f = \frac{\sum_{i=1}^{m} v(t_i)/\sigma_{v(t_i)}^2}{\sum_{i=1}^{m} 1/\sigma_{v(t_i)}^2} \quad (7)$$

where $m$ is the total number of epochs in the time series and $\sigma_{v(t_i)}^2$ is the variance of the rate at epoch $t_i$, derived from the posterior covariance matrix of the Kalman filter. The uncertainty $\sigma_v^2$ in the rate is given by

$$\sigma_v^2 = \frac{1}{\sum_{i=1}^{m} 1/\sigma_{v(t_i)}^2} \quad (8)$$

Supplementary Figs. 4 and 5 compare the time-invariant model (black curves) with the stochastic-seasonal model (red curves) for GRACE-derived GWS and RSLR from tide gauges in the Mississippi and Chao Phraya deltas. These plots show that a stochastic-seasonal process better represents the observed variability in the time series.

The post-fit residuals of the time-invariant model show some systematic seasonal patterns, particularly during periods when seasonal amplitudes deviate from the assumed constant values (Supplementary Figs. 4b,d and 5b,d). By contrast, the stochastic model accommodates time-dependent variations in seasonal amplitudes, resulting in reduced (often near-zero) residuals (Supplementary Figs. 4b,d and 5b,d), demonstrating the advantage of the stochastic-seasonal model in capturing transient seasonal variations rather than fixed annual and semiannual cycles[83].

The GWS rates for each delta are summarized in Supplementary Table 2, and Fig. 3a and Extended Data Fig. 8a show the relationship with the subsidence rates. Negative GWS trends indicate mass depletion, primarily driven by groundwater extraction, whereas positive trends represent net groundwater accumulation due to recharge processes, reduced extraction or hydrological interventions. To evaluate the reliability of GRACE-derived GWS trends, we compared them with in situ groundwater level trends for 18 deltas (Supplementary Fig. 6). Groundwater levels were compiled from two publicly available sources: 13 deltas from ref. 84 and 5 deltas from the Global Groundwater Monitoring Network[85]. Given the spatial scale discrepancy between GRACE (basin-wide) and well observations (point-scale), we emphasized agreement in trend direction rather than absolute magnitudes. Each site was categorized based on the sign of the GRACE and well trends, and a confusion matrix was constructed to assess consistency. The analysis yielded an overall classification accuracy of 88.9%, with six sites exhibiting positive–positive trends (PPT) and 10 showing negative–negative trends (NNT). Only two sites showed mixed behaviour (NPT or PNT), and no site exhibited fully opposing trends. Moreover, a high correlation ($R = 0.7$) was observed between the GRACE-based GWS and well-derived trends, further supporting the consistency of GRACE estimates at the basin scale despite localized variability in in situ measurements. Although the coarse spatial resolution of GRACE/GRACE-FO may not capture localized variations[84], its basin-scale sensitivity is well-suited to characterizing basin-wide groundwater trends. Moreover, the dominance of groundwater extraction in many deltas[2,20,31] probably ensures that GWS trends are the primary signal captured.

We find a modest linear correlation ($R = 0.5$) between GWS and subsidence rate; however, a cubic regression model ($R = 0.6$) provides a better fit (Extended Data Fig. 8a).

**Sediment flux alteration.** We obtained values for the sediment flux alteration for the 40 deltas from ref. 29. This dataset provides a global assessment of fluvial sediment supply, distinguishing between pristine sediment fluxes (before substantial anthropogenic influences) and disturbed or contemporary sediment fluxes (reflecting human influences such as dam construction and land-use changes) within the contributing delta basins. We quantified the per cent change in sediment flux for each delta using the following equation, which expresses the relative alteration (increase or decrease) in sediment delivery due to human activities:

$$\Delta \text{Sediment flux} = \left( \frac{\text{Disturbed sediment flux}}{\text{Pristine sediment flux}} - 1 \right) \times 100\% \quad (9)$$

The pristine and disturbed sediment flux, along with computed sediment flux changes for each delta, are summarized in Supplementary Table 2. A negative sediment flux change indicates a decline or loss in fluvial sediment supply (disturbed < pristine) due to human activities, whereas a positive sediment flux change reflects an increase or gain (disturbed > pristine). We acknowledge that this framework represents a simplified characterization of complex sediment delivery processes and may not capture all temporal variations in sediment supply. Furthermore, some concerns have been raised about potential errors in global sediment flux datasets[86], which we consider as a limitation in our analysis.

Figure 3a and Extended Data Fig. 8b show the relationship between sediment flux change and subsidence rates. Although a poor correlation ($R < 0.4$) is observed, we find that 62% of the deltas (25 out of 40) exhibit negative sediment flux change, indicating widespread human-induced reductions in sediment supply.

**Urban expansion.** Urban expansion is one of the most visible and rapid types of ongoing anthropogenic changes in river deltas[6]. To assess how population-driven land-use changes may affect subsidence rates across deltas, we used a global 1/8° (about 12.5 km) urban land fraction dataset, derived from high-spatial-resolution remote sensing observations[87]. This dataset tracks the conversion of natural landscapes (that is, wetlands and forests) into built environments and serves as a proxy for land-use changes that may exacerbate subsidence through increased infrastructure loading and increased groundwater demand. We quantified the urban fraction change in deltas in the twenty-first century by calculating the percentage change in the proportion of urban areas relative to total delta area between 2000 and 2020.

Supplementary Table 2 summarizes the urban fraction dataset (2000 and 2020) and the urban fraction change for each delta. Figure 3a and Extended Data Fig. 8c show the subsidence–urban expansion relationship across the 40 deltas. All deltas showed consistent urban expansion in the twenty-first century, ranging from relatively low increases (<1%) in the Ogooué river delta to significant increases (>400%) in the Indus delta. However, despite this rapid expansion, the Indus delta remains one of the least urbanized, with only 0.4% of its total area classified as urban in 2020. By contrast, the Ciliwung (Jakarta) and Neva (Saint Petersburg) deltas exhibit the highest urban fractions, exceeding 50%. A logarithmic fit best describes the full dataset and reveals a moderate but significant nonlinear inverse correlation (correlation, $R = 0.38{-}0.51$), indicating that deltas with significant urban land conversion tend to experience more pronounced land sinking (Extended Data Fig. 8c). Steadily urbanizing deltas, such as the Rio Grande and Rhine–Meuse, exhibit slower subsidence rates, whereas rapidly urbanizing deltas, such as the Brahmani and Yellow River deltas, show faster rates of land sinking. However, regional variability is evident, as some deltas deviate from the overall trend (for example, Indus and Cauvery deltas). When excluding outliers (the Indus and Cauvery deltas), subsidence and urban expansion exhibit a strong linear correlation across deltas (Extended Data Fig. 8c).

We also explored the relationship among the anthropogenic drivers (Extended Data Fig. 8d–f), finding a low ($R = 0.1{-}0.3$) correlation depending on the specific driver.

## RF analysis for identifying anthropogenic drivers of subsidence and elevation loss

Given the nonlinear and interacting relationships among GWS, sediment flux alteration, urban expansion and residual land subsidence (after GIA correction) discussed above, a machine learning framework was implemented to model these complexities. First, we attempted a multilinear regression model, incorporating interaction terms between variables, formulated as

$$VLM = x_0 + \sum_{i=1}^{n} x_i X_i + \sum_{j=1}^{m} \sum_{k=j+1}^{m} x_{jk}(X_j X_k) + \epsilon \tag{10}$$

where VLM is the predicted VLM, $x_0$ is the intercept, $X_{i,j,k}$ are the predictor variables (GWS, sediment flux alteration and urban expansion), $x_{i,j,k}$ are the regression coefficients for each predictor variable, $x_{jk}$ represents the interaction effects between predictor variables and $\epsilon$ is the residual error term. However, this multilinear regression model yielded poor performance (correlation $R = 0.38$; $R^2 = 0.15$; RMSE = 4.7 mm yr$^1$) (Fig. 3a), demonstrating the inefficiency of linear models to capture these complex dependencies and the need for a machine learning model.

Next, we used an RF machine learning model to better account for these complex nonlinear interactions between variables. RF has been widely applied in environmental and hydrological studies to model complex systems with nonlinear dependencies, outperforming traditional regression techniques in similar contexts[88–92]. The RF model is well-suited for this analysis due to its ability to handle small datasets (40 deltas), its simpler hyperparameter tuning, and its ability to compute feature importance. In this study, the primary objective for applying RF is not to predict the subsidence rates, but rather to extract key features that explain the dynamic relationships between anthropogenic drivers and subsidence across global deltas.

The RF algorithm is an ensemble learning method that uses the strength of multiple independent regressor decision trees $\{T\}$, in which each tree $\{T_t\}$ is trained on a randomly sampled subset of the input features ($\{X = X_1, X_2, X_3\}$, representing GWS, sediment flux and urban expansion) through bootstrap aggregation (bagging). Key hyperparameters, including the number of trees, maximum tree depth, minimum samples per split and minimum samples per leaf, were optimized using grid search with five-fold cross-validation to minimize overfitting and maximize predictive accuracy[93]. This ensemble approach enhances predictive performance by creating a learning environment in which a large number of predictors work on various characteristics of the input features and learn to combat overfitting and generate predictions (VLM) by computing the average of all decision tree predictions:

$$VLM = \frac{1}{T} \sum_{t=1}^{T} T_t(X) \tag{11}$$

The RF regressor optimizes each decision tree using the mean square error (MSE) defined as a cost function to identify node splits and model performance during model training and testing:

$$MSE = \frac{1}{N} \sum_{i=1}^{N} (VLM_i - \widehat{VLM_i})^2 \tag{12}$$

where $VLM_i$ is the observed VLM rate for individual delta $i$, $\widehat{VLM_i}$ is the predicted VLM rate and $N$ is the total number of observations. To assess uncertainty, we used Monte Carlo simulations to create multiple holdout fractions (0.1–0.5) across 100 iterations, randomly subsampling the 40 deltas for training and validation in each iteration. This random partitioning ensures that each delta is used in both training and validation phases across iterations, enhancing the robustness against overfitting and sampling bias. The final RF model predictions were obtained by averaging prediction estimates across all iterations. The final model performance was evaluated using the coefficient of determination ($R^2$), RMSE and mean absolute error (MAE):

$$R^2 = 1 - \frac{\sum_{i=1}^{N}(VLM_i - \widehat{VLM_i})^2}{\sum_{i=1}^{N}(VLM_i - \overline{VLM_i})^2} \tag{13}$$

$$RMSE = \sqrt{\frac{1}{N} \sum_{i=1}^{N} (VLM_i - \widehat{VLM_i})^2} \tag{14}$$

$$MAE = \frac{1}{N} \sum_{i=1}^{N} | VLM_i - \widehat{VLM_i}| \tag{15}$$

where $\overline{VLM_i}$ is the mean observed VLM rate, and the other variables are defined in equation (12). The feature importance $I_f$ for input feature $\{X = X_1, X_2, X_3\}$ was computed using the following equation, based on the cumulative reduction in node, $j$ impurity among all the trees:

$$I_f = \sum_{j \in N} \frac{\Delta I_j}{N} \tag{16}$$

where $N$ denotes the total number of trees and $\Delta I_j$ denotes the change in impurity.

Although RF effectively captures nonlinear relationships, its ensemble structure limits delta-specific interpretability. To resolve local insights into delta-specific subsidence drivers, we applied LIME, a technique within the field of explainable artificial intelligence (XAI)[94]. LIME approximates black-box models such as RF by fitting interpretable models to perturbed samples of the input data, allowing for local feature importance estimation. For each delta $X_i$, LIME approximates the RF prediction locally by using a linear surrogate model trained on perturbed instances around $X_i$. The explanation function is obtained by solving the following minimization problem:

$$\xi(X_i) = \arg \min_{g \in G}[L(f, g, \pi_{X_i}) + \Omega(g)] \qquad (17)$$

where $\xi(X_i)$ is the local interpretable model for each delta $X_i$, $g$ is the interpretable model, $f$ is the RF model, $\pi_{X_i}$ is a proximity kernel, $L(f, g, \pi_{X_i})$ is the loss function measuring the differences between $f$ and $g$, and $\Omega(g)$ penalizes complexity. This process was repeated for each delta, and deltas with low LIME model fidelity ($R^2 < 0.5$) were excluded to ensure reliable interpretation (Supplementary Table 2). The final dataset for interpretation consisted of 30 deltas, in which LIME produced more consistent feature importance estimates. The feature importance scores from LIME are normalized to obtain normalized LIME (nLIME) scores:

$$I_f^{\text{LIME}} = \frac{|\omega_f|}{\sum_{f' \in F} |\omega_{f'}|} \qquad (18)$$

where $\omega_f$ is the LIME-derived coefficient for feature $f$ and $F$ is set for all features. The nLIME scores provide an instance-specific (local) explanation rather than a global one to evaluate the relative contributions of GWS, sediment flux alteration and urban expansion in each delta. The nLIME values for each delta are summarized in Supplementary Table 2 and were analysed in a ternary diagram to visualize the heterogeneity in delta-specific subsidence and elevation-loss drivers (Fig. 3b).

It is important to emphasize that machine learning model predictions are inherently dependent on the input variables and their distributions. In this study, the predictor–response relationship implies that variations in predictor magnitudes (for example, subsidence rates and GWS rates), dataset composition (for example, inclusion or exclusion of specific deltas), and the selection of input features could influence the weighted feature importance across deltas. Moreover, localized policy interventions, such as groundwater extraction regulations or sediment management initiatives, may alter subsidence and elevation change trends over time, potentially affecting future predictions. Therefore, although our RF-based analysis provides valuable insights into the anthropogenic drivers of subsidence and elevation loss, these results should be interpreted with an awareness of dataset limitations and the potential for evolving land-use and hydrological management practices. Furthermore, the inclusion of additional deltas, particularly those representing undersampled geographic regions or differing geomorphic, socioeconomic or governance conditions, may shift model behaviour and feature rankings, as is typical in data-driven learning frameworks. Nonetheless, within the context of the current global delta sample and observed subsidence patterns, the RF-derived feature importance values provide a consistent and interpretable estimate of the relative influence of anthropogenic drivers under present conditions for these deltas.

### Historical, current and projected SLR rates
We analysed historical (twentieth century), present-day (early twenty-first century) and projected (2050 and 2100) SLR rates to assess the relative and combined impacts of rising seas and sinking lands on global river deltas.

Historical relative sea-level changes were obtained from the Revised Local Reference database of the Permanent Service for Mean Sea Level[95] (https://psmsl.org), which provides monthly relative sea-level records from globally distributed tide gauge stations. These tide gauge records have undergone quality control procedures, including corrections for datum inconsistencies, jumps and spurious data points, and validation through comparisons with neighbouring tide gauge stations[95,96]. For this study, we selected 20 tide gauge stations across 15 deltas (the Mississippi, Rio Grande, Fraser, Amazon, Chao Phraya, Mekong, Red River, Nile, Ganges–Brahmaputra, Vistula, Rhine–Meuse, Chikuma-gawa, Yangtze, Pearl and Rioni deltas), considering only stations within 100 m of the delta boundary and at least 5 years (twentieth century) of valid record. The RSLR rates for each delta were estimated by applying the stochastic-seasonal model (equations (6–8)) over the full observational record for each tide gauge. For deltas with multiple stations (for example, the Mississippi, Ganges–Brahmaputra and Rhine–Meuse deltas), individual station rates were averaged to provide a delta-wide estimate of twentieth-century RSLR. Note that the representativeness of the derived RSLR may vary for each delta following individual tide gauge characteristics (for example, is the station founded on bedrock or 'floating' in unconsolidated sediments, is the station GNSS corrected). Supplementary Fig. 7 shows the time series of relative sea level over the twentieth century for six representative deltas. Supplementary Table 1 provides a complete summary of the RSLR rates for the 15 deltas. The median twentieth-century RSLR trend across all deltas is 2.9 mm yr$^{-1}$, with measured rates ranging from −0.5 mm yr$^{-1}$ in the Amazon delta (indicating declining twentieth-century sea level) to a maximum rate of 1.5 cm yr$^{-1}$ in the Chao Phraya Delta (Fig. 5b).

To estimate present-day (early twenty-first century) absolute (geocentric) SLR rates, we used the multi-mission satellite altimetry data from 2001 to present, obtained from Copernicus Marine Environment Monitoring Service (CMEMS). This dataset provides 1/8° (about 12.5 km) gridded monthly sea level anomalies (SLA) referenced to a 20-year mean baseline (1993–2012). SLA estimates are derived from optimal interpolation, merging the level 3 along-track measurement from multiple contemporaneous altimeter missions (Jason-3, Sentinel-3A, HY-2A, Saral/AltiKa, Cryosat-2, Jason-2, Jason-1, TOPEX/Poseidon, ENVISAT, GFO and ERS1/2)[97] (https://marine.copernicus.eu/). Several necessary corrections have been applied to the raw altimetry data, including instrumental biases and drifts, geophysical, tidal and atmospheric corrections, to ensure accurate SLA estimates. Monthly mean sea-level anomalies were obtained for each delta by spatially averaging the altimetry grid points within a 100-m radius, culling outliers beyond the 95th percentile. Supplementary Fig. 8 shows the monthly SLA time series in six deltas. We estimated the twenty-first-century trends in sea-level anomalies, using equations (6–8). The altimetry-derived geocentric SLR rates for the twenty-first century show exacerbating regional SLR rates over global sea-level estimates (about 4 mm yr$^{-1}$) for 45% of the deltas (18 out of 40) (Supplementary Table 1). Regional sea-level rates vary from 0.2 mm yr$^{-1}$ in the Parana delta to 7.3 mm yr$^{-1}$ over the Mississippi delta (Fig. 1 and Supplementary Table 1). However, a negative geocentric sea-level rate of −1.9 mm yr$^{-1}$ was observed in the Rioni Delta (Black Sea) (Supplementary Table 1). This long-term sea-level decline in the twenty-first century persists in the background of short-term fluctuations (Supplementary Fig. 8d); a characteristic feature of Black Sea sea-level dynamics[50]. This twenty-first-century decline in geocentric sea level for the Rioni Delta represents more than a 100% reduction compared with historical (twentieth-century) rates, even when accounting for average VLM across the delta. To investigate this anomaly, we estimated VLM at the Poti tide gauge (Rioni Delta) by differencing twenty-first-century RSLR rates obtained from the Poti tide gauge station from geocentric SLR. The resulting VLM rate of −6.7 mm yr$^{-1}$ matches the average InSAR-derived VLM rate (−5.9 ± 0.7 mm yr$^{-1}$) within 100 m of the tide gauge. This rapid subsidence rate at the coast of Poti represents localized conditions and

highlights the need for caution when extrapolating point-based tide gauge measurements to infer delta-wide or city-wide subsidence and exposure. Note that satellite altimetry data, although highly valuable for global sea-level monitoring, were primarily optimized for open ocean conditions. Coastal environments naturally exhibit additional complexity due to processes such as shelf circulation, freshwater discharge and tidal amplification, which contribute to the inherent variability in nearshore sea-level measurements compared with offshore altimetric observations.

We use projected sea-level rates from the Intergovernmental Panel on Climate Change Sixth Assessment Report (AR6)[38,98] to assess future SLR rates across all deltas. The sea-level rate projections integrate process-based models that account for the key contributors to climate-induced sea-level change, such as thermal expansion, ocean dynamics, and glacier and ice sheet mass loss, and consider uncertainties in global temperature change and their influence on sea-level drivers[38]. We focus on the no-VLM 50th percentile (median) projected rates for 2050 (mid-twenty-first century) and 2100 (end of the twenty-first century) under shared socioeconomic pathway 2-4.5 (SSP2-4.5) and SSP5-8.5 scenario. SSP5-8.5 represents a high reference scenario associated with the highest emission levels (global atmospheric $CO_2$ concentrations exceeding 800–1,100 ppm by 2100) and associated warming of 3.3–5.7 °C (refs. 38,99). These projections provide an upper-bound reference scenario, capturing the potential worst-case outcome for future SLR. Figure 4c shows the comparison of projected SLR rates with observed land subsidence rates.

## Data availability

The vertical land motion data for all deltas are available at Zenodo[100] (https://doi.org/10.5281/zenodo.15015923). GRACE data are available from https://podaac.jpl.nasa.gov/dataset/TELLUS_GRAC-GRFO_MASCON_GRID_RL06.3_V4. The Sentinel-1 data used in this study are publicly available through the Alaska Satellite Facility and can be accessed at https://search.asf.alaska.edu. The satellite altimetry data for sea-level change are available from Copernicus Marine Environment Monitoring Service (CMEMS) and are available through http://marine.copernicus.eu/. The population for deltas was estimated using the WorldPop dataset available through https://www.worldpop.org/. Source data are provided with this paper.

## Code availability

The WabInSAR algorithm v.5.6 used to perform the SAR analysis is available at https://www.eoivt.com/software. The code for the RF analysis using MATLAB 2024b is available at Zenodo[100] (https://doi.org/10.5281/zenodo.15015923).

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

**Acknowledgements** We thank T. Jarriel, J. Swartz and P. Passalacqua for their 2021 study 'Global rates and patterns of channel migration in river deltas', which inspired the design of Figs. 1 and 3b. The research is supported by funding from the National Science Foundation and the Department of Defense. S.W. and N.S. were partly funded by the National Aeronautics and Space Administration grant no. 80NSSC21K0061.

**Author contributions** L.O.O. and M.S. conceptualized the study. L.O.O. created the figures and wrote the first draft of the paper with contributions from M.S., J.L.D. and A.T. The analysis was carried out by L.O.O., M.S., A.T., O.D., N.S., S.W., F.O., J.L., C.A., A.A. and J.O.; L.O.O., M.S., J.L.D., A.T., R.N., O.D., N.S., K.S., S.W., A.J.C., F.O., J.L., C.A., S.D., A.A., P.S.J.M., J.O. and G.C.Y. reviewed and edited the paper.

**Competing interests** The authors declare no competing interests.

**Additional information**
**Correspondence and requests for materials** should be addressed to L. O. Ohenhen.

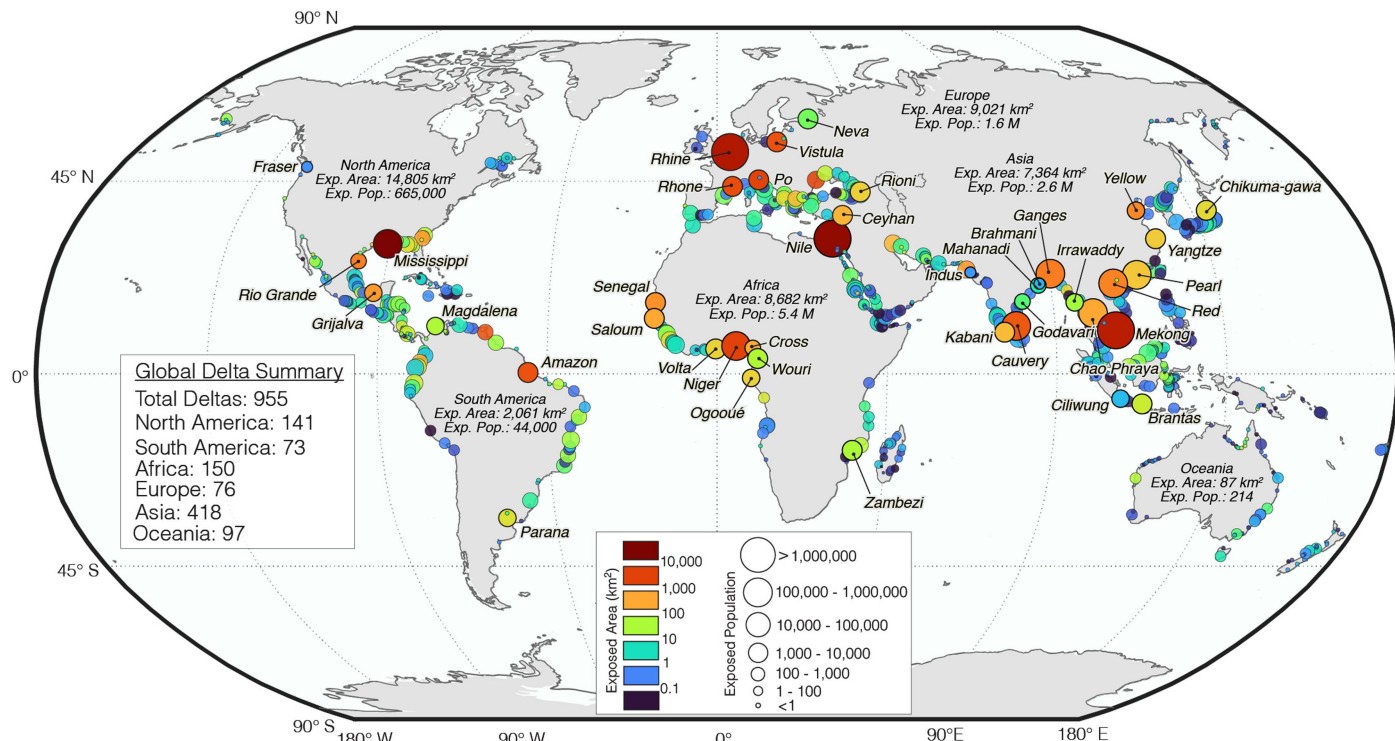

**Extended Data Fig. 1 | Global Distribution of Delta Area and Population Below Mean Sea Level.** Each circle represents one of the 955 global deltas from Edmonds et al.[6], with latitude constrained to below 60°N due to limitations in the digital elevation model dataset. The circle color indicates the land area below mean sea level (exposed area), while the circle sizes represent the population living in those areas (exposed population). The 40 deltas selected for this study are labelled. Global coastlines are based on public-domain data from the World Data Bank II (via GSHHG), distributed with MATLAB.

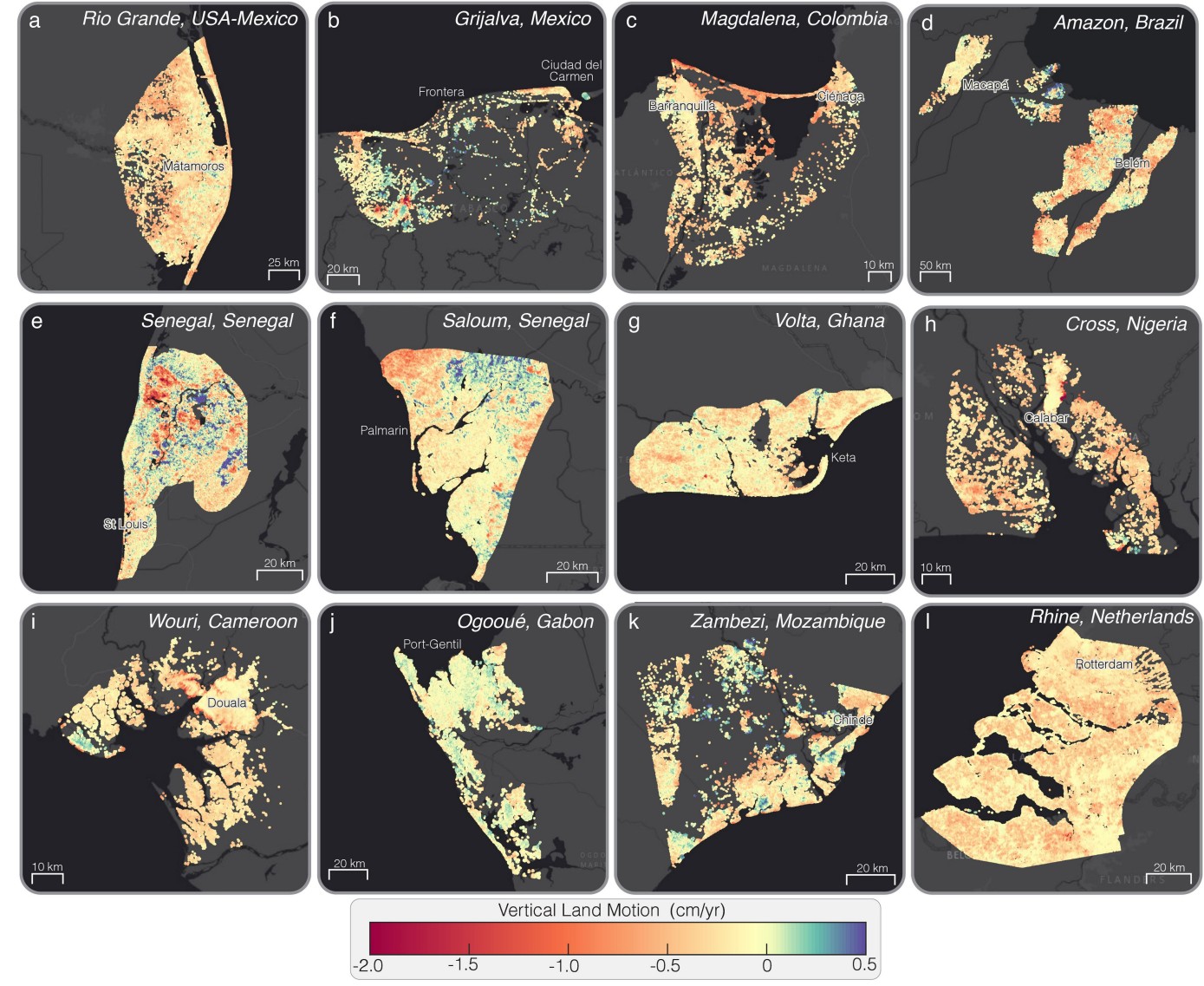

**Extended Data Fig. 2 | Spatial Pattern of Vertical Land Motion (VLM) in Deltas.** Spatial maps of VLM rates for the **(a)** Rio Grande (USA-Mexico), **(b)** Grijalva (Mexico), **(c)** Magdalena (Colombia), **(d)** Amazon (Brazil), **(e)** Senegal (Senegal), **(f)** Saloum (Senegal), **(g)** Volta (Ghana), **(h)** Cross (Nigeria), **(i)** Wouri (Cameroon), **(j)** Ogooué (Gabon), **(k)** Zambezi (Mozambique), and **(l)** Rhine-Meuse (the Netherlands) deltas. Positive VLM (green-purple hues) indicates elevation gain (uplift), while negative VLM (yellow-orange-red hues) indicates elevation loss (land subsidence). Background image is ESRI, streets-dark.

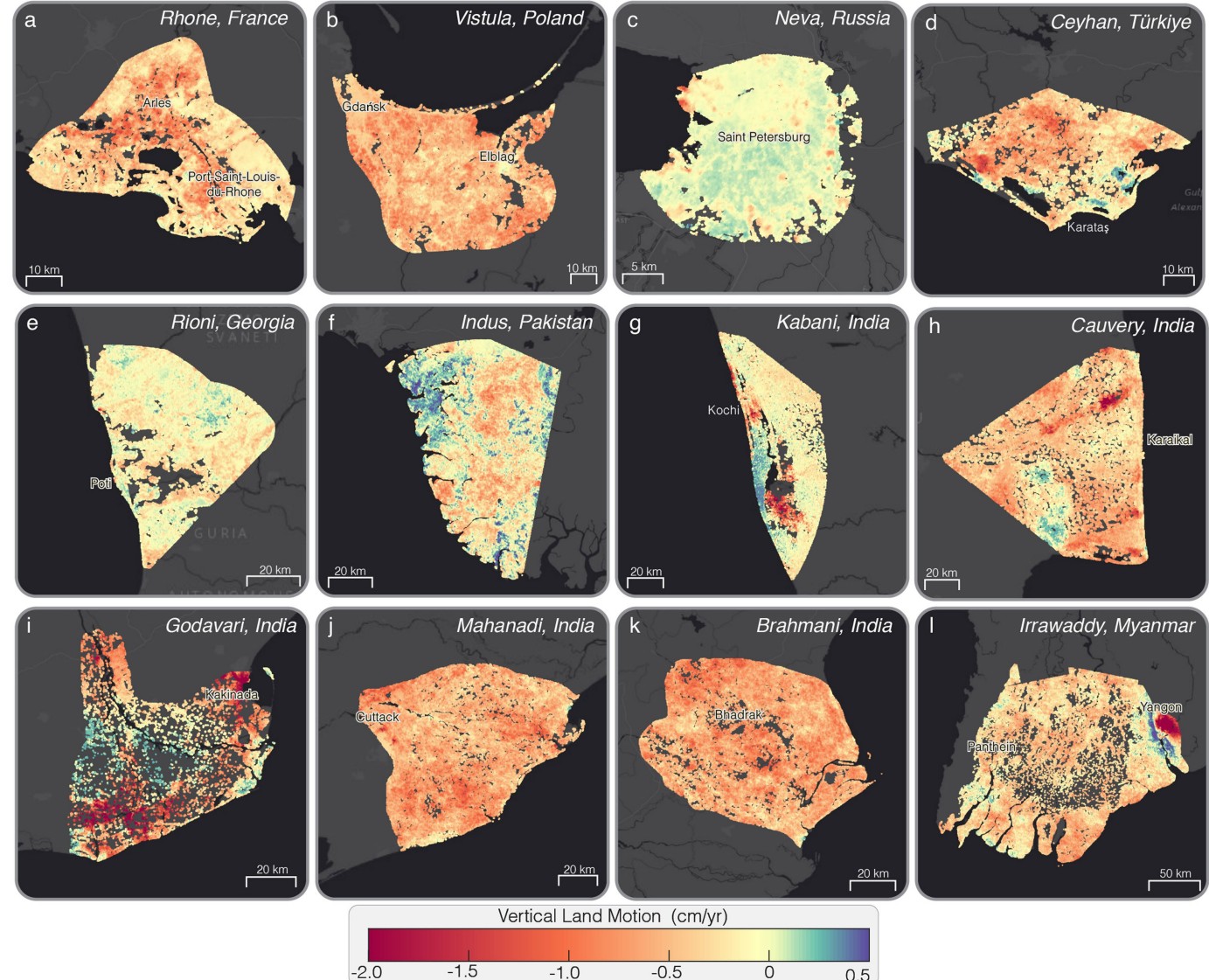

**Extended Data Fig. 3 | Spatial Pattern of Vertical Land Motion (VLM) in Deltas.** Spatial maps of VLM rates for the **(a)** Rhone (France), **(b)** Vistula (Poland), **(c)** Neva (Russia), **(d)** Ceyhan (Türkiye), **(e)** Rioni (Georgia), **(f)** Indus (Pakistan), **(g)** Kabani (India), **(h)** Cauvery (India), **(i)** Godavari (Cameroon), **(j)** Mahanadi (India), **(k)** Brahmani (India), and **(l)** Irrawaddy (Myanmar) deltas. Positive VLM (green-purple hues) indicates elevation gain (uplift), while negative VLM (yellow-orange-red hues) indicates elevation loss (land subsidence). Background image is ESRI, streets-dark.

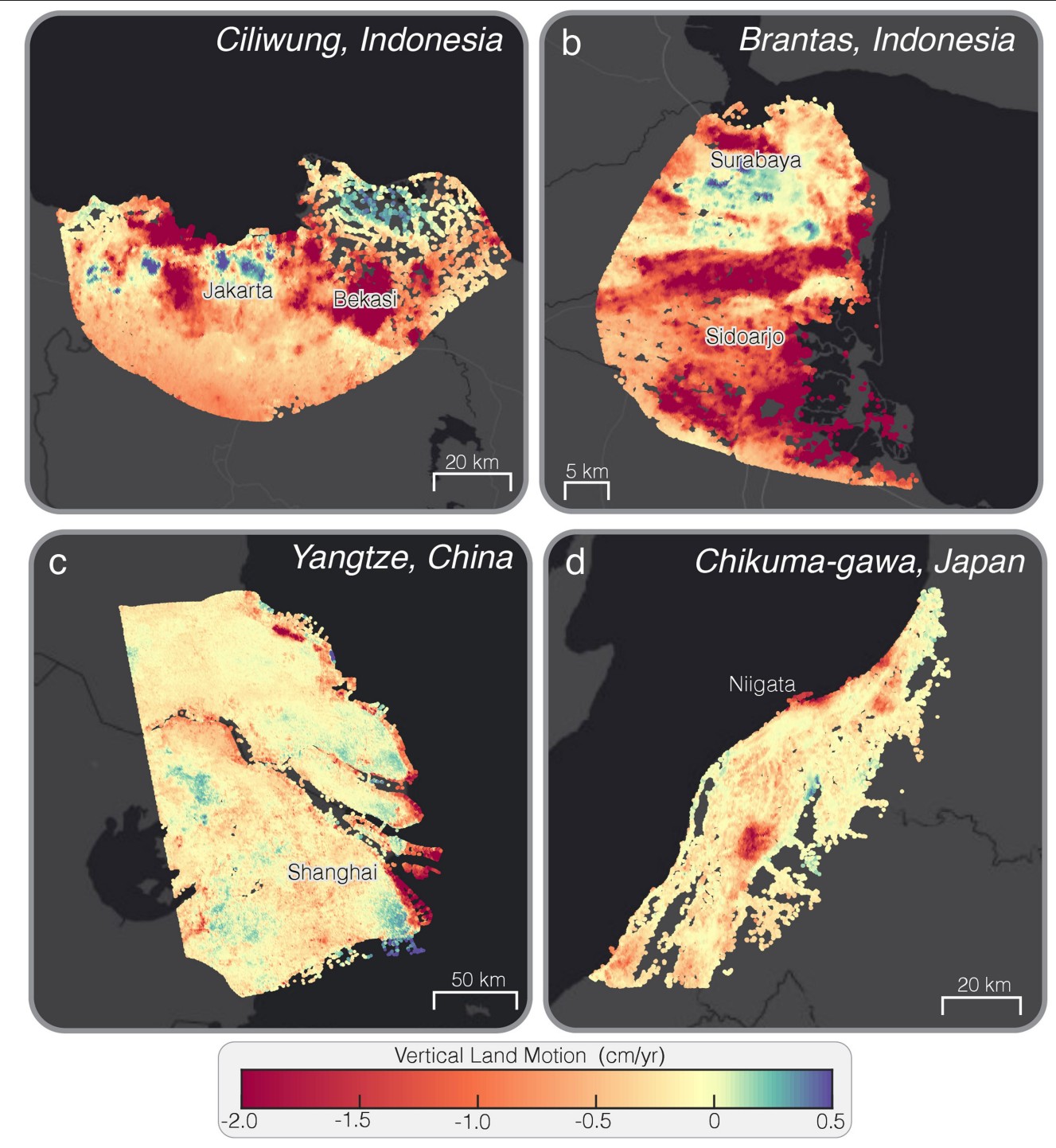

**Extended Data Fig. 4 | Spatial Pattern of Vertical Land Motion (VLM) in Deltas.** Spatial maps of VLM rates for the **(a)** Ciliwung (Indonesia), **(b)** Brantas (Indonesia), **(c)** Yangtze (China), and **(d)** Chikuma-gawa (Japan) deltas. Positive VLM (green-purple hues) indicates elevation gain (uplift), while negative VLM (yellow-orange-red hues) indicates elevation loss (land subsidence). Background image is ESRI, streets-dark.

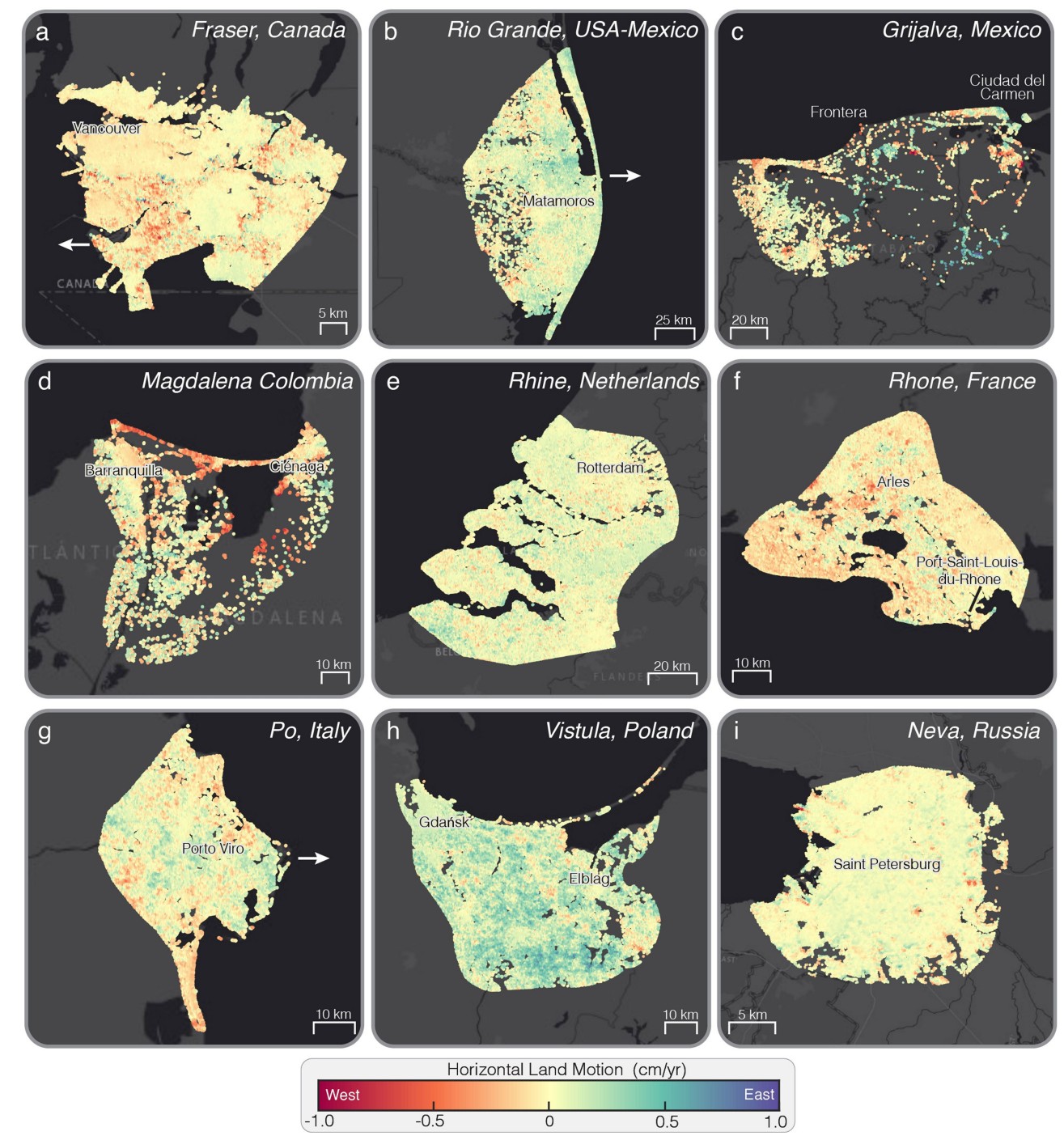

**Extended Data Fig. 5 | Spatial Pattern of Horizontal Land Motion (HLM) in Deltas.** Spatial map of HLM for the **(a)** Fraser (Canada), **(b)** Rio Grande (USA-Mexico), **(c)** Grijalva (Mexico), **(d)** Magdalena (Colombia), **(e)** Rhine (the Netherlands), **(f)** Rhone (France), **(g)** Po (Italy), **(h)** Vistula (Poland), and **(i)** Neva (Russia) deltas. Positive HLM (green-purple hues) indicates eastward motion, while negative HLM (yellow-orange-red hues) indicates westward motion. Near-zero HLM (yellow hues) represents areas with minimal horizontal displacement. Background image is ESRI, streets-dark.

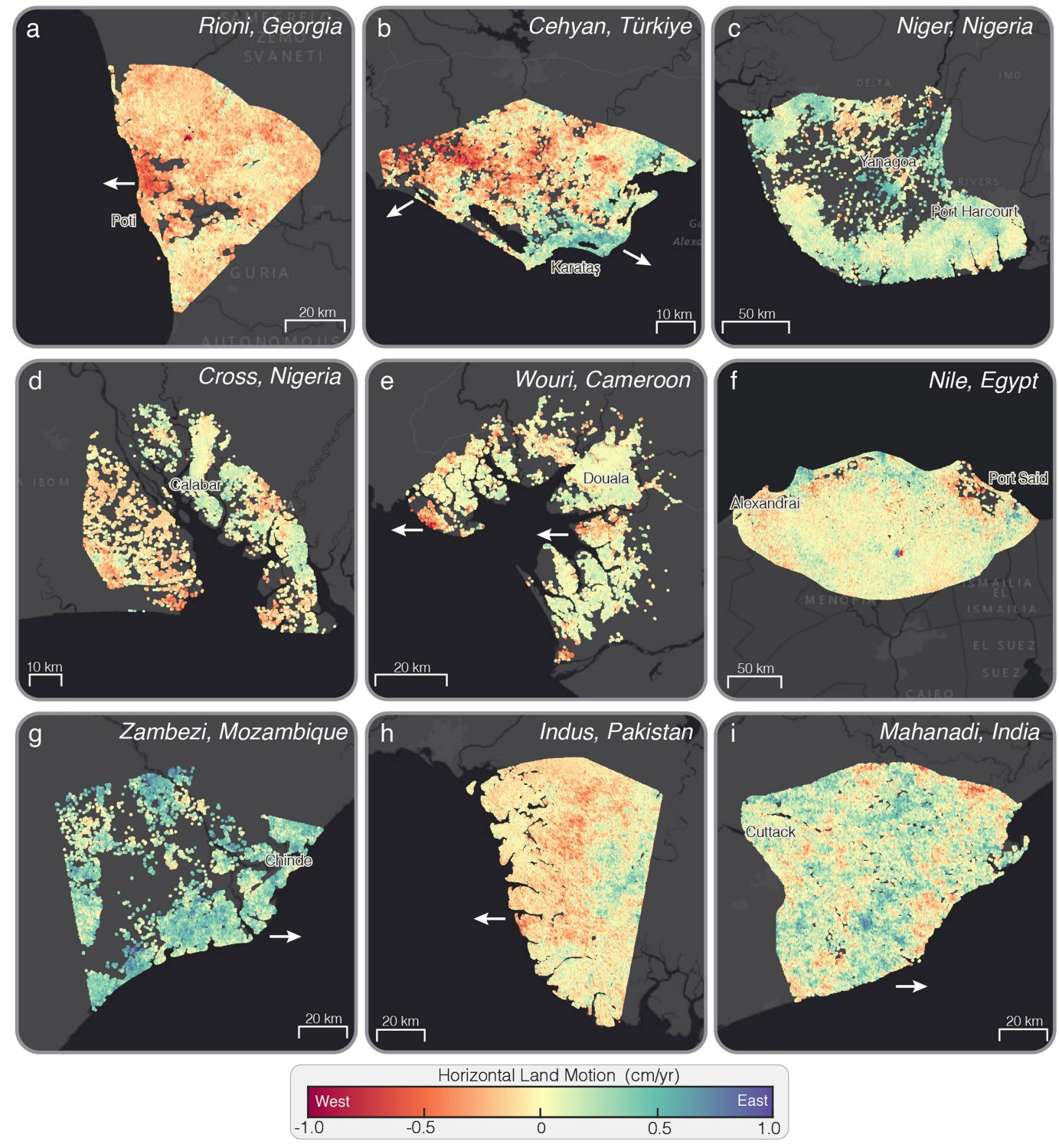

**Extended Data Fig. 6 | Spatial Pattern of Horizontal Land Motion (HLM) in Deltas.** Spatial map of HLM for the **(a)** Rioni (Georgia), **(b)** Ceyhan (Türkiye), **(c)** Niger (Nigeria), **(d)** Cross (Nigeria), **(e)** Wouri (Cameroon), **(f)** Nile (Egypt), **(g)** Zambezi (Mozambique), **(h)** Indus (Pakistan), and **(i)** Mahanadi (India) deltas. Positive HLM (green-purple hues) indicates eastward motion, while negative HLM (yellow-orange-red hues) indicates westward motion. Near-zero HLM (yellow hues) represents areas with minimal horizontal displacement. Background image is ESRI, streets-dark.

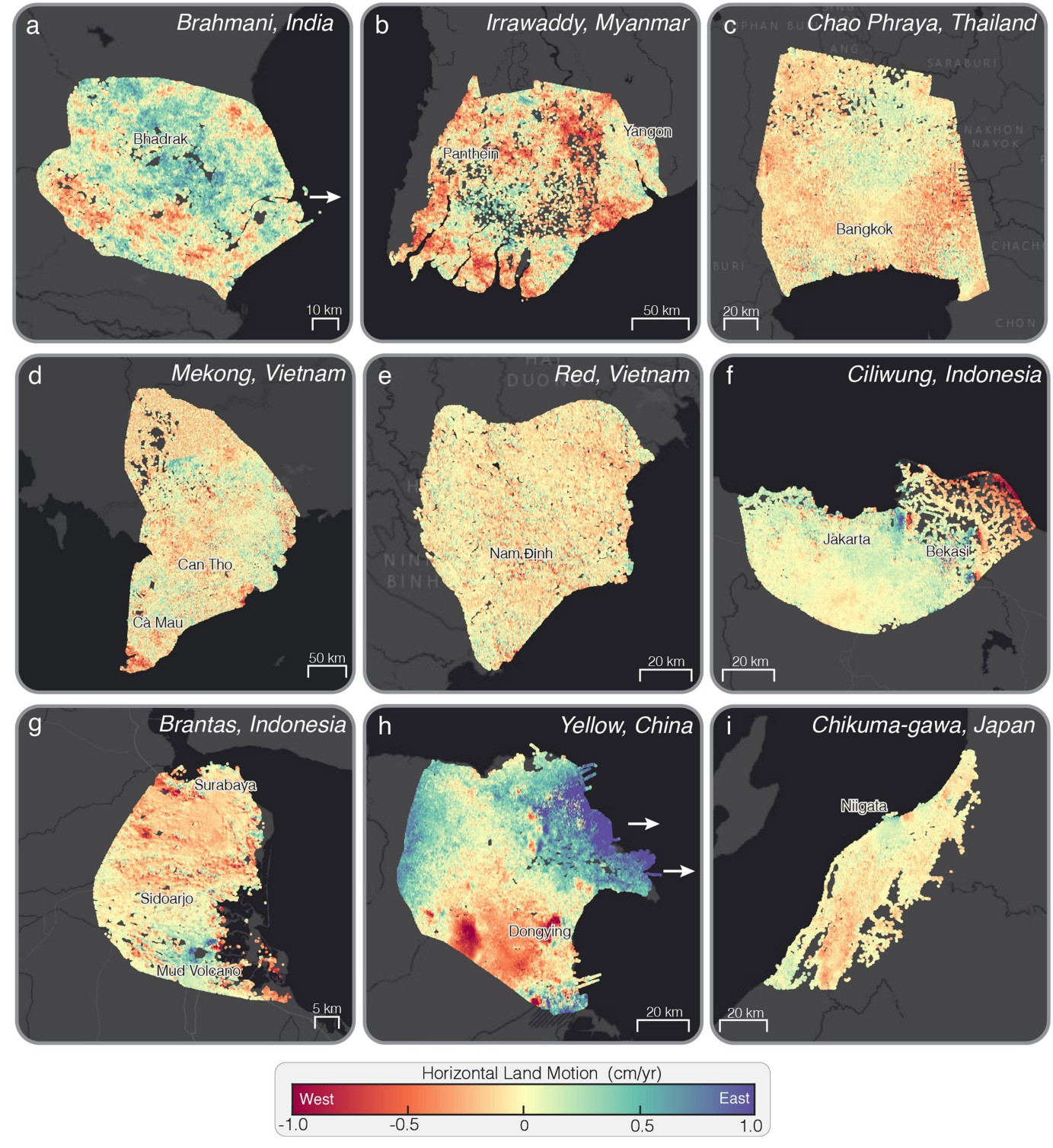

**Extended Data Fig. 7 | Spatial Pattern of Horizontal Land Motion (HLM) in Deltas.** Spatial map of HLM for the **(a)** Brahmani (India), **(b)** Irrawaddy (Myanmar), **(c)** Chao Phraya (Thailand), **(d)** Mekong (Vietnam), **(e)** Red (Vietnam), **(f)** Ciliwung (Indonesia), **(g)** Brantas (Indonesia), **(h)** Yellow (China), and **(i)** Chikuma-gawa (Japan) deltas. Positive HLM (green-purple hues) indicates eastward motion, while negative HLM (yellow-orange-red hues) indicates westward motion. Near-zero HLM (yellow hues) represents areas with minimal horizontal displacement. Background image is ESRI, streets-dark.

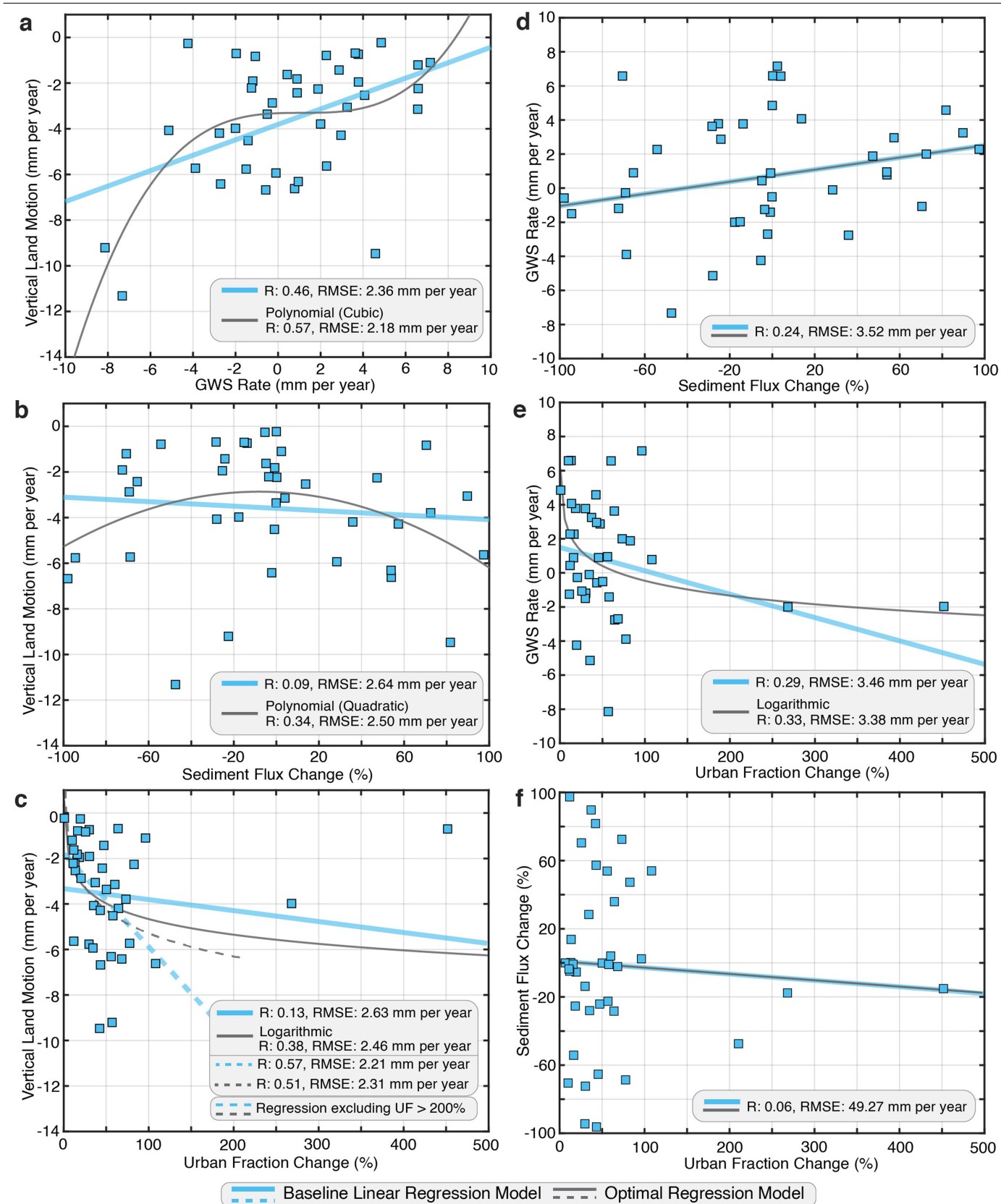

**Extended Data Fig. 8 | Relationship between Vertical Land Motion (VLM) Rates and Anthropogenic Drivers.** Scatter plots of VLM (mm per year) versus **(a)** groundwater storage (GWS) rate (mm per year), **(b)** sediment flux change (%), and **(c)** urban fraction (UF) change (%) for the 40 deltas. Scatter plot of GWS rate (mm per year) versus **(d)** sediment flux change (%) and **(e)** UF change (%). **(f)** Scatter plot of sediment flux change (%) versus UF change (%). Each relationship is analyzed using linear regression as well as polynomial and logarithmic regression models to assess the best-fit representation. Multiple regression fits (linear, quadratic, logarithmic) are shown to illustrate the varied nature of relationships between individual predictors and VLM, demonstrating the need for a nonlinear modeling approach.