## [Peer Review file · Nature]

Global Subsidence of River Deltas

Corresponding Author: Dr Leonard Ohenhen

Version 0:

Reviewer comments:

Referee #1

(Remarks to the Author)

Reviewer Report for "The Global Threat of Sinking Deltas" by Ohenhen et al.,

Key Results

This manuscript offers a comprehensive global evaluation of land subsidence in river deltas, utilizing high-resolution remote sensing data. It reveals that

- $\geq 65\%$ of the mapped delta area is experiencing subsidence. In 18 deltas, the average subsidence rate already surpasses the current geocentric sea-level rise.
- Groundwater storage loss emerges as the primary anthropogenic predictor of VLM in approximately 37% of deltas, while sediment flux decline and urban expansion play secondary, context-dependent roles.
- When subsidence is factored into IPCC AR6 SSP5-8.5 projections, the 95th percentile RSLR in many deltas by 2050–2100 exceeds the ocean-only signal by factors of 2–8, affecting around 236 million people.
- These synoptic figures, the 75 m resolution VLM maps, and the effort to partition anthropogenic drivers represent the manuscript's main contributions.

Validity

The overall strategy—deriving delta wide subsidence from Sentinel 1 InSAR and linking it to putative drivers—is conceptually sound and rests on well established techniques. No fundamental flaw is evident, yet two substantive validity issues must be resolved.

- Groundwater attribution

The assertion that groundwater withdrawal is the primary anthropogenic influence in approximately 37% of deltas relies almost exclusively on GRACE/GRACE-FO trends, which track surface water and groundwater storage. The GRACE mascon footprints, spanning about 300-400 km (approximately 200,000 km²), are significantly larger than most delta plains, often resulting in signal leakage from neighboring aquifers and basins. Given that surface water and groundwater are hydraulically interconnected in many deltas, changes in total water storage also reflect variations in rivers, reservoirs, lakes, and wetlands. Without first subtracting the surface water component (e.g., rivers, wetlands, lakes, and reservoirs) using models that have these components like WGHM or existing in-situ data, and without local well data or pumping records for validation, the connection between the decline in surface water storage plus groundwater storage and aquifer depletion remains speculative. The manuscript should

- recover GRACE data at the native mascon resolution ~ 3 -deg,
- separate groundwater from surface water signals, and
- verify at least a subset of deltas against in situ water level or extraction statistics.

Only then can subsidence be confidently attributed to groundwater extraction.

- Natural subsidence omitted

All deltas experience background "geologic" subsidence: Holocene sediment compaction (typically 1–2 mm yr⁻¹), glacial isostatic adjustment (e.g. ~ 1 mm yr⁻¹ on the U.S. Gulf Coast), sediment load flexure, and, in some settings, tectonic subsidence. For example, roughly one fifth of the 5 mm yr⁻¹ total subsidence measured in the Mississippi Delta is expected from long term GIA. Excluding such natural terms can inflate the anthropogenic share and misallocate residual variance among groundwater, sediment flux, and urban load. The authors should therefore:

- incorporate global GIA fields (e.g., (Caron et al., 2018)) and published estimates of long term compaction (Yu et al., 2012) into their VLM budget, or

- clearly justify and document any assumption that natural rates are negligible or spatially uniform.

Until this baseline is accounted for, statements that subsidence is “largely human caused” need a caveat: part of the observed sinking would occur even in the absence of contemporary human activity. Addressing these two issues will prevent over interpretation of the anthropogenic contribution and greatly strengthen the manuscript’s validity.

- Sea level benchmark:

Subsidence is compared with open ocean altimetry. Coastal altimetry (geocentric) typically shows 20–50 % faster rise; thus, VLM dominance may be exaggerated. Authors need to incorporate coastal altimetry rather than open ocean altimetry, and contextualize their results with their the scope of the regional rates not the global

Cazenave A, Gouzenes Y, Birol F, Leger F, Passaro M, Calafat FM, Shaw A, Nino F, Legeais JF, Oelmann J, Restano M. Sea level along the world’s coastlines can be measured by a network of virtual altimetry stations. *Communications earth & environment*. 2022 May 16;3(1):117.

- Scenario choice

Concentrating exclusively on SSP58.5 (previously RCP 8.5) emphasizes an extreme, upper-limit emissions pathway that already diverges from current trends and observations. Please include results for a midrange scenario—such as SSP24.5—to present readers with a more realistic spectrum of potential futures.

Data and Methodology

Beyond the GRACE-related issues previously mentioned, two additional methodological considerations warrant attention.

- The study employs a multilinear regression to associate subsidence rates with three anthropogenic drivers, subsequently using LIME (Local Interpretable Model-agnostic Explanations) to assess the significance of each driver. With only 40 deltas, any regression or ML model is susceptible to outliers and multicollinearity. The three predictors are likely interrelated—highly urbanized deltas often feature drained wetlands and upstream dams, for instance—resulting in potentially unstable standard coefficients. It is crucial to report variance inflation factors, cross-validated R^2 , and sensitivity to the exclusion of high-leverage deltas. These diagnostics are vital for establishing statistical robustness.

- The ternary diagram of LIME scores holds potential for insight, yet the overplotting of large bubbles hampers readability. Consider cleaner alternatives: plain points with labels, smaller symbols, or separate subplots for the scatter and ternary views. A simplified graphic would convey the attribution results far more effectively.

Reviewer Limits

As a reviewer, I have done my best to evaluate all aspects of this multidisciplinary study, but I acknowledge certain technical limits in my expertise and access:

- I am not an expert in the latest InSAR time-series algorithms. I cannot independently verify the specifics of the InSAR data processing (e.g., unwrapping errors, atmospheric correction algorithms, or how phase ambiguities were resolved) without access to the authors’ methodology description or code. I trust the authors’ expertise in this area (several co-authors are known for InSAR work) and assume standard, validated techniques were used.

- I have focused my review on the geoscientific reasoning, the integration of data, and the alignment of conclusions with evidence, while relying on the authors’ expertise for the specialized technical execution in areas such as InSAR processing. Consequently, my feedback focuses more on interpretation rather than algorithmic validation.

Despite these limitations, I found the manuscript to be highly engaging and generally well-executed. My critique is intended to help clarify and strengthen the work, rather than to question its fundamental approach

Caron, L., et al., 2018. GIA model statistics for GRACE hydrology, cryosphere, and ocean science. *Geophysical Research Letters*. 45, 2203-2212.

Yu, S.-Y., et al., 2012. Quantifying Holocene lithospheric subsidence rates underneath the Mississippi Delta. *Earth and Planetary Science Letters*. 331-332, 21-30.

Referee #2

(Remarks to the Author)

Review of The Global Threat of Sinking Deltas – L. Ohenhen et al.

Summary of the key results

This is a significant and welcome contribution to the underrated but extremely serious problem of subsidence in deltas globally. The authors, bringing together among the best international expertise on the subject, have carried out a thorough scrutiny of the various drivers and implications of subsidence, insisting on the three important components of (dominantly) groundwater extraction, sediment supply, and urban growth.

Originality and significance

The manuscript addresses delta subsidence as a vulnerability factor that not only dominates contemporary sea-level rise because of its impact on delta surface elevation but also because it tends to be neglected in vulnerability assessments. This study will, hopefully, bring to the fore the urgency of further tracking subsidence in deltas and tackling, from a management point of view, its pernicious effects.

Data & methodology: validity of approach, quality of data, quality of presentation

The authors use state-of-the-art databases on: delta surface elevation change from remote sensing images and generated

maps on surface deformation changes, groundwater, sediment supply, urban development and an adaptation index to which they apply a comprehensive suite of analytical tools and models to unravel, with some success (given the complexity of delta environments), global delta subsidence, explore its drivers, its spatial variations and the degree of vulnerability-based subsidence level and the way it is addressed from a management perspective. The authors use available sea-level data to further highlight the dominance of subsidence in driving sea-level rise over climate-change induced sea-level rise.

Application of the term 'subsidence'

Lines 112-118: There is a need for a more rigorous application of where subsidence prevails relative to, respectively, vertical land motion (VLM)/surface elevation change (SES). The former apply to static environments and the latter to dynamic environments (your ref. 17). What are the thresholds of population density applied here? What are the extents of densely populated areas and urban areas relative to non-urban/low population areas in these deltas? Is sedimentation entirely excluded in densely populated areas (where this can be necessary for agriculture with sedimentation occurring during seasonal flooding as in the Mekong)?

Are you sure you are measuring VLM rather than SES?

Methods

In order to establish the primary conclusions of the manuscript, the correct magnitudes of Vertical Land Motion (VLM) with regards to geocentric sea-level rise are paramount.

The authors do a good job in validating InSAR derived VLM rates with GNSS stations. The validation is shown in Suppl. Figure 11 with a good R 0.8. However, it would be useful to validate the data with other high-resolution InSAR products. One available dataset is the European Ground Motion Service (EGMS) <https://egms.land.copernicus.eu/> which provides point displacement data for the whole EU area. Measured average displacements for some European deltas compared with the data in Table 1 show marked differences:

Rhone: -1.23 mm/year (EGMS) vs. around -3mm (paper data).

Po: -4.67 (EGMS) vs. -6 (paper data) ;

Vistula: -1.4 (EGMS) vs. -5.4 (paper data)

SEE REVIEWER ATTACHMENT OF MAP OF VLM for the Rhone. Manuscript data(left) compared to EGMS (right).

Why this negative bias for the VLM data in the manuscript (1.3 to 4 mm/yr)? It is important to clarify this, as the paper compares subsidence with geocentric sea-level rise. Apart from validation with EGMS, there should be a comparison with other high-resolution VLM datasets (if available) including the ref29 of the lead author for the Mississippi for example.

For the Rhone delta, a big part of the high subsidence values 0.5 cm/yr to 1 cm/yr appears over agricultural fields. It is unclear how you identified the surfaces suitable for ground point deformation. Many surfaces lack points in the EGMS because the service obtains data from most reliable and coherent areas like man-made structures or bare stable surfaces. How reliable is calculated VLM in surfaces that are heavily vegetated or subjected to agricultural use? You mention using the (WabInSAR) algorithm that handles these challenging surfaces but it should be more clearly shown how this algorithm separates true deformation from vegetation-induced noise for example.

Also, the manuscript does not seem to provide methods on how the discrete VLM measurements from points were interpreted to produce the continuous velocity map grids shown in the figures. Considering the apparent smoothness in the figures, the mentioned 75 m-resolution seems an overstatement, with the grid data being inherently much smoother.

Datasets and their robustness

Sediment flux data

You use sediment flux changes from the difference between pristine (without humans) and disturbed fluxes (with humans) based on ref 32. This is highly debatable and simplistic, if we are to accept that there is indeed a difference in the type of sediment supply in these deltas between 'pristine' and 'human influence' over the period of subsidence analysis (last decade or so: 2014-2023). Ref. 32 further proposes datasets that are fraught with uncorrected errors (see Zainescu, F. et al., 2023. Concerns about data linking delta land gain to human action. *Nature*, 614, E20–E25. <https://doi.org/10.1038/s41586-022-05624-x>

Another point regarding sediment flux is that of in-situ organic production in deltas with large wetland systems which is not mentioned. We are not sure how important this can be (we are not aware of any global synthesis) but mention should at least be made of this, especially given its added influence on (differential) compaction and subsidence.

-GRACE data processing and suitability

SWS (Surface Water Storage) is included in the GRACE data. The authors could not remove the SWS from Total Water Storage (TWS) to isolate the GWS. However, in large deltas fed by major rivers and subject to large seasonal flooding (e.g., Amazon, Mekong) the SWS signal can be quite substantial; hence the statement that "GWS is probably the primary signal" is an assumption, and this nuance needs to be critically examined. Although in some deltas the GWS signal can be very strong (as in the Chao Phraya), perhaps this contributes to the very low correlation between the GWS and VLM, essentially indicating noise.

Appropriate use of statistics and treatment of uncertainties

The authors first employ simple and multiple regression analyses that fail to convey the complexity of the relationships and drivers of subsidence, and then used more sophisticated non-linear models based on a Random Forest machine-learning framework that turned out to be much more robust in extracting interdependencies among the various parameters. To go

further and resolve delta-specific subsidence drivers, they applied Local Interpretable Model-Agnostic Explanations (LIME), derived from Explainable Artificial Intelligence. Altogether mindful of the complex subject of delta subsidence, the authors have gone at lengths to render their results robust, coherent and useful. We have, however, a number of concerns.

- Machine Learning Application

Small Sample Size (N=40): Although RF may be able to better accommodate smaller sets than many other learning methods, a sample size of N=40 (then later N=30) is quite modest for training a model with global application. The notable standard deviations reported for feature importance (0.4 ± 0.3 for GWS) indicates model instability, and high sensitivity to the specific subsets of deltas used in training. This may be an inherent limitation, but the authors should be more clear in communicating the implications this has on the model robustness and its potential generalisation.

Filtering by LIME Fidelity

The authors excluded 10 deltas (25% of the original dataset) where the local LIME models did not reach a fidelity of $R^2 > 0.5$. This filtering should provide improved reported performance metrics of the retained deltas, but may have introduced a selection bias.

Why were the models failing for these 10 deltas? How are they distinct?

By removing them from the analysis, the conclusions of the paper about "global" drivers become restricted to, technically, only 30 deltas for which the models performed well. Therefore, the limitations of a small sample size should be acknowledged more directly and the claims of the model's generalizability tempered.

Predictor Interdependence

The authors indicate moderate correlation between urban expansion and GWS. This should not be a big surprise, given that urbanization increases demand on groundwater systems. RF handles some collinearity, but the interdependency of factors makes the interpretation of feature importance less certain. The importance attributed to GWS may even overlap with the effects of urbanization to some degree. This distinction would be worth noting for discussion.

L641-642 : What were the exact challenges that could not be overcome in using synthetic aperture radar (SAR) imaging and interferometric analysis in the Danube, Orinoco, and Shatt-el-Arab deltas ?

Suppl Fig 8. A: Correlation R 0.44, in the plot. In lines 925-926 you mention a different value for essentially the same data: "However, this model yielded poor performance (correlation, $R = 0.22$; RMSE = 4.7 mm per year) (Figure 3a)". Which is the correct value?: b,c,d,e,f) All correlations are either low or non-existent. Please check for significance (p values). Some correlations in e,f have data which look more like outliers, so correlations should be carefully interpreted (correlation with outliers removed in c seems more robust).

For non-linear correlations, cubic, quadratic, or logarithmic fit are discretionarily used. There should be an underlying mechanistic reason why one model is chosen over the other. For example what sense does it make for a polynomial (quadratic) to be fitted to plot b? This would imply that increasing sediment flux would lead to negative VLM; contrary to the expected logical outcome of increased sediment input.

Conclusions: robustness, validity, reliability

The study highlights a suite of potential pathways for analysing delta subsidence and vulnerability that is relatively robust, valid and reliable, inasmuch as this reliability is constrained by the quality of the data, delta feature heterogeneity, and uncertainties arising from more or less strong human impacts. We are just not sure whether you should not add a brief comment acknowledging a certain degree of uncertainty in all this, given the complexity of deltas individually and across a given delta, and the uneven quality of various datasets over large swathes of deltas.

Suggested improvements: experiments, data for possible revision

See items in previous sections.

References: appropriate credit to previous work?

Yes, this is a well-referenced study. Justified suggestions for a handful of additional references are indicated in comments to the authors for minor revision.

Some additional references you may want to consider:

-Törnqvist TE and Blum MD (2024). What is coastal subsidence? Cambridge Prisms: Coastal Futures, 2, e2, 1–4. <https://doi.org/10.1017/cft.2024.1>

- Pedretti L, Giarola A, Korff M, Lambert J and Meisina C (2024) Comprehensive database of land subsidence in 143 major coastal cities around the world: overview of issues, causes, and future challenges. *Front. Earth Sci.* 12:1351581. doi: 10.3389/feart.2024.1351581

Clarity and context: lucidity of abstract/summary, appropriateness of abstract, introduction and conclusions

Well organized, well written and cannot be faulted in any way.

Further suggestions for revision

- Lines 48-50: I am not sure references 1 (2006) and 2 (2009) are pertinent regarding current (2025) global delta population, which is estimated at 350 to >500 M. These conflicting estimates (see also Nicholls, R.J., Adger, W.N., Hutton, C.W. & Hanson, S.E. (eds.). *Deltas in the Anthropocene* (Springer, 2020) might also need to be acknowledged, while stating that you are using Edmonds et al. (2020).
- Lines 64-65, 330-336: You refer here to deltas as 'low-lying landforms averaging less than 2 m above sea level'. Can you

provide a reference? There are several areas in the manuscript where delta elevations are referred to, and accurate delta surface elevations are indeed a major gap in current vulnerability studies of deltas. Are these references to elevations derived from the DeltaDTM (reference 62)? If yes, then this reference needs to be cited in the main text.

- Lines 82-86 + Lines 536-541 (reference 26): Please break down this sentence as it is rather confusing as regards the two references. Ref. 26 is on sea level and 25 on issues in the five megadeltas. Why is the comment on definition of sea level attached to this reference in the reference list?
- Line 108: in 40 deltas globally. Globally is superfluous here.
- Line 168: You use a global delta area of 710,187 km² from Edmonds et al. (ref. 6). Again, estimates of global delta vary and are presumably robustly proposed in each study, such that this variation may need to be stated alongside your recourse to Edmonds et al. (2020). Syvitski et al. (2022), for instance, robustly calculated an area of 855,000 km². Syvitski, J. et al. 2022. Large deltas, small deltas: Towards a more rigorous understanding of marine deltas. *Global & Planetary Change*, 2018, 103958. <https://doi.org/10.1016/j.gloplacha.2022.103958>
- Lines 246-247. A missing but relevant reference here that highlights delta complexity and draws attention to many facets of delta dynamics and global differences is Syvitski et al. (2022) cited above.
- Lines 354-355: You state that your findings challenge the prevailing narrative framing climate-induced sea-level rise as the principal hazard in low-elevation coastal zones (ref 45). Low-elevation coastal zones cover well beyond the 1% of coast occupied by deltas, and most low-elevation coastal zones (beaches, beach-ridge plains, dunes, etc..) are not exposed to subsidence. This statement therefore needs to be reframed more cautiously.
- Line 627: Global delta population 'was' estimated.

Florin Zainescu, Edward Anthony, Aix-en-Provence

Referee #3

(Remarks to the Author)

I have finished reviewing the manuscript titled "The Global Threat of Sinking Deltas" which addresses a critical and timely issue concerning the vulnerability of river deltas worldwide. It provides a comprehensive analysis of the drivers of subsidence and their possible implications for coastal populations, infrastructure, and ecosystems.

Overall I am very positive about the work, the paper addresses an important topic, it is well written, with high quality figures and interesting methods.

Summary of the key results

The work underlines the importance of including subsidence in coastal adaptation planning. This is a key point and important finding.

Originality and significance:

The work is novel and significant. It builds on several previous studies that are properly cited in the manuscript

Data & methodology:

The methods seem sound and the results are very well presented. I have to admit that I am not a remote sensing expert and even though I read thoroughly the methods, I may miss some important details. The part of the SAR Interferometric analysis is important, despite the detailed description I would expect the authors to provide code, but in case this is not possible, at least to provide more details about the environment the work is done (toolboxes, etc).

In general, I believe that sharing the code would be important for a work of such importance. I also think it would not be a big challenge since we are not referring to large modelling frameworks, but rather data analysis that I assume that could be easily packaged in a repository (e.g. the RF application).

Appropriate use of statistics and treatment of uncertainties

One that is missing in the manuscript is a description of uncertainty. Starting from VLM, I would expect that such measurements would come with significant error, especially for such extensive areas as deltas. Figure 4c mentions some percentiles (and Figs 9, 11 of the SI), but I think it would be helpful to show confidence intervals for the important variables. The same applies for the Groundwater Storage Change, as I would be surprised if such calculations would not come with substantial uncertainty.

Conclusions:

I found the conclusions solid and the analysis robust

Suggested improvements:

Doesn't Fig 9a of the SI mean that the RF approach underestimates subsidence? Shouldn't these results discredit RF as a predictive tool and just use it to interpret the results? If yes why use RF and not other multivariate techniques including standard multivariate regression? I am not questioning the selection of the method, but I think the above should be discussed.

Figure 2a. Shouldn't the bars have a total of 100%?

Check reference 26 (Gregory et al), page 14

Figure 5. I think the 4 'types' of deltas should be explained in the caption

References:

The authors give appropriate credit to previous work and to my understanding cite all previous major studies

Clarity and context:

I found the manuscript very well written

Referee #4

(Remarks to the Author)

I co-reviewed this manuscript with one of the reviewers who provided the listed reports.

Version 1:

Reviewer comments:

Referee #1

(Remarks to the Author)

The authors adequately addressed key review issues.

In my view, the manuscript is scientifically sound, transparent about scope, and broadly suitable for publication.

Referee #2

(Remarks to the Author)

The authors have carried out the revisions to the letter. The revisions make for a robust manuscript that will be useful to the delta community. Our comments concern completing one reference, and very minor (syntax, punctuation) edits that may even be carried out at the proof stage.

Reference 98: Zainescu, F. et al., Concerns about data linking delta land gain to human action. Nature, 614, E20–E25 (2023). <https://doi.org/10.1038/s41586-022-05624-x>

Line 40: .. groundwater storage 'has' (in lieu of have)

Line 88: ...is just as, or more, influential than (punctuation)

Line 117: For consistency, to reflect .. (punctuation)

Lines 333-334: ... floods that deposited sediments, are documented to now experience (correct experience) severe sediment deficits due to dams and levees, (add comma here) accelerating elevation loss ...

Well done, once again!

Edward Anthony, Florin zainescu, Aix-en-Provence, France.

Referee #3

(Remarks to the Author)

I enjoyed reading the revised manuscript and the authors' interaction with the reviewers.

The paper has been corrected substantially and its limitations are presented more transparently..

So I can recommend publication of the article

Referee #4

(Remarks to the Author)

I co-reviewed this manuscript with one of the reviewers who provided the listed reports.

Referee #5

(Remarks to the Author)

Key Results

The authors of this manuscript present an extensive analysis of elevation changes and their main drivers in selected deltas around the globe. They reveal that approximately 54-65 % of the global deltas are sinking. They analyzed the role of three key anthropogenic drivers and found that groundwater storage loss is the primary driver of elevation changes in 35% of deltas, while groundwater storage, sediment flux decline and urban expansion rather have mixed influences in the other deltas. They show that subsidence is the dominant driver of relative sea level rise in many deltas today, and even under the worst-case, high-emission scenarios (SSP5-8.5), maximum subsidence rates in all deltas will exceed projected sea level rise rates in all deltas. Lastly, they provide a study on the varying adaptation readiness of the selected deltas.

Validity

The individual components of the analysis – the InSAR analysis of elevation changes in the deltas, the machine learning regression of the contribution of three key drivers, and the assessment the future impact of subsidence and sea level rise in deltas based on IPCC scenarios – are based on established and validated techniques, and the analysis seems to be extensive and thorough. I did not find any major flaws, but I have a couple of questions/discussion points which I give in the specific paragraphs below.

Originality and significance

I consider the analysis as an overdue and essential upgrade of the global delta analysis by Syvitski et al. (2009). The elevation change dataset in 40 deltas, achieved with InSAR, is impressive. Together with the analysis of associated anthropogenic drivers and the future impact of subsidence and sea level rise based on different emission scenarios, the analysis highlights the urgency of the overall issue and hopefully contributes to the initialization or further development of adaptation policies and appropriate countermeasures.

Reference:

Syvitski, J., Kettner, A., Overeem, I. et al. Sinking deltas due to human activities. *Nature Geosci* 2, 681–686 (2009). <https://doi.org/10.1038/ngeo629>

Data & methodology

InSAR analysis:

- On page 18, lines 709-711, you describe that you multi-looked the SAR data. Later in line 733, you describe a coherence threshold for persistent scatterers. How did you identify and analyze persistent scatterers when the SAR data has previously been multi-looked with a boxcar? Furthermore, how did you incorporate persistent scatterers in the final aggregated results?
- I am surprised by the apparent quite dense coverage of your InSAR results in almost all deltas, as seen in the figures. In the Mekong Delta, for example, your results seem to even cover parts of forested areas (such as parts of the U Minh Forest), and areas which are widely covered by agriculture and even aquacultural farms with many water bodies. Is this just a matter of the visualization? If not, how did you identify evaluable pixels in such areas, and which kind of phase filtering / phase optimization approaches did you apply for distributed scatterers?
- Multi-temporal multi-look InSAR can be influenced by phase biases, whose magnitude depends on the temporal baseline distribution of selected interferograms and the land cover (Ansari et al., 2021; Maghsoudi et al., 2022). A large percentage of the analyzed areas is covered by agricultural fields. Devlin and Lohman (2025) have shown in a Sentinel-1 InSAR study that agricultural fields in the San Joaquin Valley, CA, can be associated with a bias of ~2–4 cm/yr of apparent subsidence, when only analyzing short-term interferograms of 6 days temporal baselines. You describe that first allowed interferograms with a temporal baseline of up to 300 days, but later you sorted out interferograms based on the estimated coherence, “while maintaining a 50% temporal baseline coverage”. What does this exactly mean? Did you consider the risk of phase bias while selecting the interferograms in your study?

References:

Ansari, H., De Zan, F., & Parizzi, A. (2021). Study of systematic bias in measuring surface deformation with SAR interferometry. *IEEE Transactions on Geoscience and Remote Sensing*, 59(2), 1285-1301.

Maghsoudi, Y., Hooper, A. J., Wright, T. J., Lazecky, M., & Ansari, H. (2022). Characterizing and correcting phase biases in short-term, multilooked interferograms. *Remote Sensing of Environment*, 275, 113022.

Devlin, K. R., & Lohman, R. B. (2025). Evaluation of vegetation bias in InSAR time series for agricultural areas within the San Joaquin Valley, CA. *Earth and Space Science*, 12, e2024EA004062.

- Multi-looked leads to a smoothing of displacements within windows. De Wit et al. (2021) have shown that there is significant differential subsidence on short spatial distances in urban areas in the Mekong Delta, and I assume this also applies to other deltas. The reason behind is the occurrence of different foundation depths of buildings, loading effects, differences in historical land use, etc. As a result, aggregating vertical motion rates mixes motion signals from different depths and sources. Could you discuss if this is a relevant issue/uncertainty in your study?

Reference:

de Wit, K., Lexmond, B. R., Stouthamer, E., Neussner, O., Dörr, N., Schenk, A., & Minderhoud, P. S. (2021). Identifying causes of urban differential subsidence in the Vietnamese Mekong Delta by combining InSAR and field observations. *Remote Sensing*, 13(2), 189.

- Page 5, lines 191-193: “In some deltas (e.g., Wouri, Zambezi, Indus, Ciliwung, and Yellow), the observed uplift or elevation gaining parts correlate with patterns of sediment deposition and horizontal land motion”. As land subsidence is a process originating in the subsurface, the surface backscatter characteristics do not necessarily change during the subsidence process. On the contrary, sedimentation is a process which alters the surface backscatter characteristics, forcing decorrelation of the InSAR measurements. I’m not challenging your results, I am really interested in these findings. I would be interested to see (1) if you can provide/discuss similar literature which describes the measurement of elevation gain by sedimentation with InSAR, (2) a short analysis whether the InSAR coherence is statistically different in areas which are characterized by elevation gain than in others.
- On page 19, lines 736-737 you describe how you estimated and corrected orbital errors. The mentioned approach in reference 76 estimates orbital errors by assuming that no displacement signals exhibit long wavelengths. Can you be sure that this is the case in all of the study areas?
- Page 19, line 737-738: “[...] minimized the effects of topography-correlated components of atmospheric phase delay [...]” Topography-correlated atmospheric disturbances are rather small in flat deltas, while turbulent tropospheric effects can be really strong due to the flat, maritime location of deltas, as well as the frequent low latitude. Have you applied any approach to reduce effects from the turbulent troposphere? Furthermore, did you account for the following long-wave effects, which can have significant contributions to large-scale InSAR displacement estimates: solid earth tides, ocean tide loading, ionospheric disturbances?
- Page 19, lines 764-766: Mention that G is the matrix which comprises the projection vectors
- Page 20, line 793: How was the velocity reference point for the InSAR analyses selected? Did you select them in the near

vicinity of the GNSS stations which were used later for the alignment? How did you mathematically align the reference points to the GNSS-derived rates? How did you do this if several GNSS stations were present in the study areas? Did you incorporate estimate noise statistics into the alignment?

- Page 20, line 794: 17 deltas were mentioned before in line 790, why only 16 now?
- Page 20, line 795: I've exemplarily checked the GNSS stations from the Nevada Geodetic Laboratory in the Mekong Delta, and found that not a single one covers the whole time span between 2016 and 2023, which was analyzed in this study. How did you generally select appropriate GNSS Stations? Also, please describe how you handled data gaps in GNSS time series and the presence of very few stations in large study areas?

GWS:

- Page 24, lines 944-945: I exemplarily checked both sources for groundwater level data in the Mekong Delta, but could not find the data during my rapid search. Where can I find the groundwater data from the Mekong Delta included in Supp. Figure 14?

Urban Expansion:

- Page 25, lines 1010-1012: Have you tested the absolute urban land conversion in km² between 2000 and 2020 for the analysis? I guess the urban fraction change is hard to compare if the absolute urban area in 2000 was largely different among the studied deltas.

Appropriate use of statistics and treatment of uncertainties

- Page 20, line 810-811: Can you briefly explain how you estimated the standard deviation of the land motion results?
- Page 20, line 814 "independent GNSS data": I guess independent means that these are stations which previously have not been considered in the transformation of VLM rates from local to the global reference frame. Please clarify.
- Page 21, line 815: What does "historical recorded rates" mean exactly? What is the time difference between the GNSS and the InSAR measurements? In case of non-linear subsidence, this might have a significant impact. This should be briefly discussed.
- Page 21, line 820-822: I wonder how representative GNSS stations outside the deltas are. The conditions for InSAR and hence its uncertainties might be more challenging in deltas than in adjacent areas. This depends on vegetation, land surfaces changes during the observation time series, and even atmospheric conditions could be different in a delta than adjacent areas which might be more inland. While the validation gives valuable insights in the general performance of the InSAR algorithm, it does not necessarily yield exact uncertainties of the results within the deltas.

Conclusions

- Page 7, line 251-258 "However, we note some deltas deviate from this trend. [...] These outliers and deviations highlight the complex, nonlinear, and interwoven processes driving surface elevation change in deltaic systems". You should discuss if these findings could also be caused by uncertainties in the analyzed parameters.
- Page 9, lines 324-325: "Importantly, no delta exhibits elevation change driven solely (100%) by one variable." Is this something LIME could provide as a result? Is a FI score of 0.01 significant (Example Mekong Delta)?
- Page 11, lines 404-405 "These findings identify VLM as the principal hazard in deltaic systems and some subsidence-prone low-elevation coastal zones." What do you mean by "some subsidence-prone low-elevation coastal zones"?

Suggested improvements

See above

References

The manuscript references the literature appropriately.

Clarity and context

The manuscript is well written and structured. The discussion has been provided with sufficient context and consideration of previous work.

Inflammatory material

-

Reviewer Limits

I was asked to take a particular look at the InSAR analysis for the review, as this is my main research field. This is why my review largely concerns the technical aspects of the InSAR analysis. I still have done my best to evaluate the other parts of the study, but I acknowledge limitations in my expertise relating to the machine learning analysis and the GRACE-derived groundwater storage change. Even though I was focusing on technical aspects, I still found the manuscript to be well written and structured, with a thorough data integration and important conclusions. I did not intend with my comments to question the overall approach and results, but to help clarify and strengthen certain aspects of the analysis.

Nils Dörr, Karlsruhe Institute of Technology

Version 2:

Reviewer comments:

Referee #5

(Remarks to the Author)

The authors satisfactory addressed my suggestions and clarification requests. I support the publication of this manuscript.

Reviewer #1 (Remarks to the Author): (Reviewers comments in normal text, response to reviewers is in bold)

Reviewer Report for “The Global Threat of Sinking Deltas” by Ohenhen et al.,

Key Results

This manuscript offers a comprehensive global evaluation of land subsidence in river deltas, utilizing high-resolution remote sensing data. It reveals that

- $\geq 65\%$ of the mapped delta area is experiencing subsidence. In 18 deltas, the average subsidence rate already surpasses the current geocentric sea-level rise.
- Groundwater storage loss emerges as the primary anthropogenic predictor of VLM in approximately 37% of deltas, while sediment flux decline and urban expansion play secondary, context-dependent roles.
- When subsidence is factored into IPCC AR6 SSP5-8.5 projections, the 95th percentile RSLR in many deltas by 2050–2100 exceeds the ocean-only signal by factors of 2–8, affecting around 236 million people.
- These synoptic figures, the 75 m resolution VLM maps, and the effort to partition anthropogenic drivers represent the manuscript’s main contributions.

Validity

The overall strategy—deriving delta wide subsidence from Sentinel 1 InSAR and linking it to putative drivers—is conceptually sound and rests on well established techniques. No fundamental flaw is evident, yet two substantive validity issues must be resolved.

We sincerely thank the Reviewer for their comprehensive and constructive review of our manuscript. We appreciate the several important points raised, which were useful in strengthening the manuscript and clarifying our methodological approaches. See detailed point-by-point responses to the comments below:

- Groundwater attribution

The assertion that groundwater withdrawal is the primary anthropogenic influence in approximately 37% of deltas relies almost exclusively on GRACE/GRACE-FO trends, which track surface water and groundwater storage. The GRACE mascon footprints, spanning about 300-400 km (approximately 200,000 km²), are significantly larger than most delta plains, often resulting in signal leakage from neighboring aquifers and basins. Given that surface water and groundwater are hydraulically interconnected in many deltas, changes in total water storage also reflect variations in rivers, reservoirs, lakes, and wetlands. Without first subtracting the surface water component (e.g., rivers, wetlands, lakes, and reservoirs) using models that have these components like WGHM or existing in-situ data, and without local well data or pumping records for validation, the connection between the decline in surface water storage plus groundwater storage and aquifer depletion remains speculative. The manuscript should

- recover GRACE data at the native mascon resolution ~3-deg,
- separate groundwater from surface water signals, and
- verify at least a subset of deltas against in situ water level or extraction statistics.

Only then can subsidence be confidently attributed to groundwater extraction.

We thank the Reviewer for their comment. We have now repeated the analysis extracting GRACE data at the native mascon resolution (~3-deg), we employed WGHM to separate surface water storage contributions from total water storage to derive groundwater storage (GWS) trends. Additionally, we validated the GWS trends against in-situ well data for 18 deltas (Supplementary Fig. 14). However, we note that we use this validation to show correlation in trends rather than exact correspondence in rates, as basin-wide groundwater storage changes are expected to differ from localized measurements from in-situ wells.

- Natural subsidence omitted

All deltas experience background “geologic” subsidence: Holocene sediment compaction (typically 1–2 mm yr⁻¹), glacial isostatic adjustment (e.g. ~1 mm yr⁻¹ on the U.S. Gulf Coast), sediment load flexure, and, in some settings, tectonic subsidence. For example, roughly one fifth of the 5 mm yr⁻¹ total subsidence measured in the Mississippi Delta is expected from long term GIA. Excluding such natural terms can inflate the anthropogenic share and misallocate residual variance among groundwater, sediment flux, and urban load. The authors should therefore:

- incorporate global GIA fields (e.g., (Caron et al., 2018)) and published estimates of long term compaction (Yu et al., 2012) into their VLM budget, or
- clearly justify and document any assumption that natural rates are negligible or spatially uniform.

Until this baseline is accounted for, statements that subsidence is “largely human caused” need a caveat: part of the observed sinking would occur even in the absence of contemporary human activity. Addressing these two issues will prevent over interpretation of the anthropogenic contribution and greatly strengthen the manuscript’s validity.

We thank the Reviewer for their comments. We agree that natural processes such as GIA, tectonics, and sediment compaction constitute a significant part of background subsidence rates in deltas and coastal regions. In fact, we acknowledge this in the opening sentence of the section “Anthropogenic Drivers of Subsidence in Deltas”: ‘All deltas, by their inherent nature, subside over time as recently deposited sediments compact under their weight, a process further influenced by isostatic adjustments and tectonic activity.’ We also agree that proper VLM budgeting requires addressing all components (natural and anthropogenic) across all deltas.

However, this presents a significant challenge for a global study spanning multiple deltas, as globally consistent datasets for processes like sediment compaction rates are not available and goes beyond the scope of this study. In this study, we did not set out to

investigate VLM budgets for the deltas but rather to understand the relative influence of three key anthropogenic drivers (groundwater storage, sediment change due to humans, and urbanization) on elevation change across these 40 deltas using globally consistent datasets. We acknowledge this objective, and the conclusions thereof were not properly framed in the original abstract.

To address these concerns, we have: (1) incorporated the Caron et al. GIA dataset to remove glacial isostatic adjustment contributions from our VLM measurements before analyzing the anthropogenic drivers, (2) modified the abstract and other sections to clarify that we analyze the relative influence of three anthropogenic drivers rather than claiming comprehensive attribution across deltas, (3) retained references to subsidence being largely human driven only in the introduction section when citing relevant previous foundational studies, and (4) explicitly added limitation statements in lines 336 – 357 to acknowledge the caveats regarding natural VLM processes. We recognize that comprehensive VLM budgeting requires additional datasets and delta-specific analyses, which is beyond the scope of our current study.

• Sea level benchmark:

Subsidence is compared with open ocean altimetry. Coastal altimetry (geocentric) typically shows 20–50 % faster rise; thus, VLM dominance may be exaggerated. Authors need to incorporate coastal altimetry rather than open ocean altimetry, and contextualize their results with their the scope of the regional rates not the global

Cazenave A, Gouzenes Y, Birol F, Leger F, Passaro M, Calafat FM, Shaw A, Nino F, Legeais JF, Oelsmann J, Restano M. Sea level along the world's coastlines can be measured by a network of virtual altimetry stations. *Communications earth & environment*. 2022 May 16;3(1):117.

We thank the Reviewer for this suggestion. While coastal satellite altimetry offers improved sea-level estimates near the shoreline, we explored its applicability as recommended and identified several limitations that constrain its use in our global delta-scale analysis.

First, spatial data availability is inconsistent. For 6 of the 40 deltas in our study, no viable virtual coastal altimetry stations exist within 500 km of the delta boundary. Incorporating altimetry-derived sea-level trends for only a subset of deltas would introduce methodological inconsistency. Second, we find that the conclusion that coastal altimetry systematically yields higher sea-level rise rates than gridded products is generally untrue in the 40 deltas. Among the 34 deltas with usable coastal altimetry stations, 67% exhibit lower rates than those derived from our current combination of gridded altimetry and VLM data (see the figure below). Thus, using these coastal altimetry values would demonstrate even greater dominance of VLM over sea-level rise, not less. In cases where altimetry values are comparable to our estimates, virtual stations tend to be located close to the delta margin, supporting the validity of our original approach. Third, the Parana delta shows anomalously high sea-level rise rate (5.0 mm per year), and its coastal altimetry station is located more than 250 km from the delta boundary, raising concerns about its representativeness. A co-located tide gauge records show RSLR of

4.6 ± 0.5 mm per year, within errors of our estimate of 4.0 ± 0.8 mm per year derived from delta-averaged VLM (-3.8 mm per year) plus regional sea-level trends (0.2 mm per year). Using the 5.0 mm per year sea-level rate would yield a RSLR of ~9 mm per year, which is inconsistent with observed tide gauge records.

For these reasons, we have chosen to retain the gridded altimetry dataset to ensure methodological consistency and global coverage across all 40 deltas. However, we acknowledge the limitations of the satellite altimetry dataset in lines 1201–1206.

Figure 1: Scatter plot comparing ASL using gridded satellite altimetry dataset versus virtual tide gauge coastal altimetry dataset.

- Scenario choice

Concentrating exclusively on SSP58.5 (previously RCP 8.5) emphasizes an extreme, upper-limit emissions pathway that already diverges from current trends and observations. Please include results for a midrange scenario—such as SSP24.5—to present readers with a more realistic spectrum of potential futures.

We thank the Reviewer for this suggestion. We have now analyzed the SSP2-4.5 midrange scenario and presented these results in lines 385-393 and Figure 4c.

Data and Methodology

Beyond the GRACE-related issues previously mentioned, two additional methodological considerations warrant attention.

- The study employs a multilinear regression to associate subsidence rates with three anthropogenic drivers, subsequently using LIME (Local Interpretable Model-agnostic Explanations) to assess the significance of each driver. With only 40 deltas, any regression or ML model is susceptible to outliers and multicollinearity. The three predictors are likely interrelated—highly urbanized deltas often feature drained wetlands and upstream dams, for instance—resulting in potentially unstable standard coefficients. It is crucial to report variance inflation factors, cross-validated R^2 , and sensitivity to the exclusion of high-leverage deltas. These diagnostics are vital for establishing statistical robustness.

We thank the Reviewer for their comment and the opportunity to clarify our methodological approach. We believe there may be a misinterpretation of our analysis, as we did not employ multilinear regression as the primary modeling framework. As noted in lines 260 – 266, we evaluated a multilinear regression model and found that it performed poorly ($R^2 = 0.1 \pm 0.1$ now = 0.2 ± 0.1 after removing GIA and SWS from GRACE data), motivating our choice of a Random Forest (RF) machine learning approach, which is better suited to capturing nonlinear interactions among variables.

To assess the potential for multicollinearity, we examined pairwise correlations among the three anthropogenic predictors (Supplementary Fig. 8d–f), which showed low to moderate correlations ($R = 0.1–0.3$). These relationships, as well as the interactions between each predictor and the VLM dataset, are discussed in lines 961 – 1023 of the Methods section. We further evaluated the statistical robustness of the RF model through a Monte Carlo cross-validation framework, using holdout fractions ranging from 0.1 to 0.5 over 100 iterations. This approach randomly excluded different deltas in each iteration, allowing us to report average model performance ($R^2 = 0.6 \pm 0.1$; $RMSE = 1.8 \pm 0.2 \text{ mm yr}^{-1}$; $MAE = 1.6 \pm 0.2 \text{ mm yr}^{-1}$) and assess variability in feature importance (e.g., groundwater storage: 0.4 ± 0.3). As is well-established in the literature, RF models are relatively robust to multicollinearity and outliers, making them an appropriate choice for our multivariate, nonparametric analysis.

- The ternary diagram of LIME scores holds potential for insight, yet the overplotting of large bubbles hampers readability. Consider cleaner alternatives: plain points with labels, smaller symbols, or separate subplots for the scatter and ternary views. A simplified graphic would convey the attribution results far more effectively.

We thank the Reviewer for this suggestion. We believe the comment may refer to Figure 3a rather than the ternary diagram in Figure 3b. Figure 3a presents a simplified, integrated view of the relationships among the three anthropogenic drivers and vertical land motion (VLM). By plotting VLM against groundwater storage trends and encoding urban fraction change as bubble size and sediment flux change as bubble color, the figure allows us to distinguish how deltas experiencing groundwater loss (i.e., more negative GWS trends), reductions in sediment supply (red/yellow hues), and higher urban expansion (larger bubbles) tend to exhibit higher rates of subsidence; compared to deltas with minimal groundwater depletion, stable sediment flux, or limited urban growth. While we have reduced the maximum bubble size to improve readability, some overlaps are unavoidable due to clustering of deltas with similar anthropogenic pressures. To enhance interpretability, we refer readers to Supplementary Figure 8, which disaggregates the individual pairwise relationships between each anthropogenic driver and VLM. Additionally, we have updated Figure 3b to include explicit delta labels

and improved marker distinction. The caption for Figure 3 now explicitly references Supplementary Figure 8 to guide readers to these complementary visualizations. Supplementary Figure 8 will be included in the Extended Data of the final manuscript not the supplementary Figures.

Reviewer Limits

As a reviewer, I have done my best to evaluate all aspects of this multidisciplinary study, but I acknowledge certain technical limits in my expertise and access:

- I am not an expert in the latest InSAR time-series algorithms. I cannot independently verify the specifics of the InSAR data processing (e.g., unwrapping errors, atmospheric correction algorithms, or how phase ambiguities were resolved) without access to the authors' methodology description or code. I trust the authors' expertise in this area (several co-authors are known for InSAR work) and assume standard, validated techniques were used.
- I have focused my review on the geoscientific reasoning, the integration of data, and the alignment of conclusions with evidence, while relying on the authors' expertise for the specialized technical execution in areas such as InSAR processing. Consequently, my feedback focuses more on interpretation rather than algorithmic validation.

Despite these limitations, I found the manuscript to be highly engaging and generally well-executed. My critique is intended to help clarify and strengthen the work, rather than to question its fundamental approach

Caron, L., et al., 2018. GIA model statistics for GRACE hydrology, cryosphere, and ocean science. *Geophysical Research Letters*. 45, 2203-2212.

Yu, S.-Y., et al., 2012. Quantifying Holocene lithospheric subsidence rates underneath the Mississippi Delta. *Earth and Planetary Science Letters*. 331-332, 21-30.

We sincerely thank the Reviewer for their valuable feedback on our manuscript. We have greatly enjoyed responding to their thoughtful comments and suggestions, which have significantly improved the clarity and scientific rigor of our work.

Reviewers #2 (Remarks to the Author): (Reviewers comments in normal text, response to reviewers is in bold)

Review of The Global Threat of Sinking Deltas – L. Ohenhen et al.

Summary of the key results

This is a significant and welcome contribution to the underrated but extremely serious problem of subsidence in deltas globally. The authors, bringing together among the best international expertise on the subject, have carried out a thorough scrutiny of the various drivers and implications of subsidence, insisting on the three important components of (dominantly) groundwater extraction, sediment supply, and urban growth.

Originality and significance

The manuscript addresses delta subsidence as a vulnerability factor that not only dominates contemporary sea-level rise because of its impact on delta surface elevation but also because it tends to be neglected in vulnerability assessments. This study will, hopefully, bring to the fore the urgency of further tracking subsidence in deltas and tackling, from a management point of view, its pernicious effects.

Data & methodology: validity of approach, quality of data, quality of presentation

The authors use state-of-the-art databases on: delta surface elevation change from remote sensing images and generated maps on surface deformation changes, groundwater, sediment supply, urban development and an adaptation index to which they apply a comprehensive suite of analytical tools and models to unravel, with some success (given the complexity of delta environments), global delta subsidence, explore its drivers, its spatial variations and the degree of vulnerability-based subsidence level and the way it is addressed from a management perspective. The authors use available sea-level data to further highlight the dominance of subsidence in driving sea-level rise over climate-change induced sea-level rise.

We thank Drs. Florin Zainescu and Edward Anthony for their excellent and comprehensive review of our manuscript. We appreciate the time and effort they put into this review and are grateful for their thoughtful engagement with the manuscript. Their detailed feedback and expert insights have been invaluable in strengthening the manuscript. Below, we provide point-by-point responses to their comments.

Application of the term 'subsidence'

Lines 112-118: There is a need for a more rigorous application of where subsidence prevails relative to, respectively, vertical land motion (VLM)/surface elevation change (SES). The former apply to static environments and the latter to dynamic environments (your ref. 17). What are the thresholds of population density applied here? What are the extents of densely populated areas and urban areas relative to non-urban/low population areas in these deltas? Is sedimentation entirely excluded in densely populated areas (where this can be necessary for agriculture with sedimentation occurring during seasonal flooding as in the Mekong)?

Are you sure you are measuring VLM rather than SES?

We cannot absolutely exclude surface elevation change processes in our measurements even in densely populated region, as InSAR inherently captures the integrated signal of vertical land motion and surface processes. Therefore, throughout the manuscript we use both 'uplift and elevation gain' or 'subsidence and elevation loss' to acknowledge that our measurements may include both VLM and surface elevation change processes in deltas. We have modified lines 113-121 accordingly to clarify this distinction and our terminology usage.

Methods

In order to establish the primary conclusions of the manuscript, the correct magnitudes of Vertical Land Motion (VLM) with regards to geocentric sea-level rise are paramount.

The authors do a good job in validating InSAR derived VLM rates with GNSS stations. The validation is shown in Supl. Figure 11 with a good R 0.8. However, it would be useful to validate the data with other high-resolution InSAR products.

One available dataset is the European Ground Motion Service (EGMS) <https://egms.land.copernicus.eu/> which provides point displacement data for the whole EU area. Measured average displacements for some European deltas compared with the data in Table 1 show marked differences:

Rhone: -1.23 mm/year (EGMS) vs. around -3mm (paper data).

Po: -4.67 (EGMS) vs. -6 (paper data);

Vistula: -1.4 (EGMS) vs. -5.4 (paper data)

SEE REVIEWER ATTACHMENT OF MAP OF VLM for the Rhone. Manuscript data(left) compared to EGMS (right).

[REDACTED]

Why this negative bias for the VLM data in the manuscript (1.3 to 4 mm/yr)? It is important to clarify this, as the paper compares subsidence with geocentric sea-level rise. Apart from validation with EGMS, there should be a comparison with other high-resolution VLM datasets (if available) including the ref29 of the lead author for the Mississippi for example.

We thank the Reviewers for their comments. Attempts at comparison between different datasets and temporal ranges is one of the major methodological flaws in current understanding of coastal processes between different scientific communities that we highlight in our methods section. As noted in lines 800 – 808 of the manuscript and Minderhoud et al. (2025), there is an increasing number of scientific studies that compare VLM globally or compare different studies or instrumentation without properly accounting for reference frame differences and temporal variability. Firstly, the comparison of different VLM products must be made against the backdrop of accurate reference frame transformation. Without appropriate transformation to accurate reference frames, comparisons of global InSAR data in a single study or against other studies are fundamentally methodologically inaccurate. For example, most InSAR measurements are created and reported in local reference frames relative to local stable areas. To compare such InSAR measurements accurately, all datasets must be transformed into the same or a uniform global reference frame (e.g., IGS14/ITRF2014). Secondly, comparison of datasets with different temporal ranges assumes non-evolving VLM characteristics. Environmental datasets including VLM exhibit nonlinear behavior over time, meaning that rates change as underlying drivers evolve. This temporal variability constitutes the scientific basis for continuously updating VLM measurements across global coastlines to assess changes in VLM rates. For example, consider the differences between the EGMS dataset between 2018-2022 and 2019-2023 for 3 deltas.

Figure 1: Bivariate plot comparing EGMS VLM for 2018-2022 with EGMS VLM for 2019-2023 for (a) Rhone, (b) Rhine, and (c) Po deltas. The correlation, R values are shown in the figures.

Note that the bivariate plot shows significant differences with the correlation (R) values ranging between 0.6 – 0.9. Additionally, we also note differences between the historical GNSS records and GNSS records from 2014-2023 in our validation (Supplementary Figure 11). These differences demonstrate that VLM rates are not temporally static and datasets with different temporal ranges may not be useful for robust quantitative accuracy analysis. In Ohenhen et al., 2024 (ref 29), we calculated rates from 2007 to 2021 using two different satellite products (ALOS-1 and Sentinel-1), presently, we utilize Sentinel-1 product 2016-2023, representing only a 35% temporal overlap between the two datasets. Any comparison of these products would be useful to indicate temporal

evolution of subsidence processes and changes in underlying drivers rather than measurement validation.

Accurate and robust comparisons can only be made if datasets are: (1) transformed to the same reference frame, (2) from the similar temporal period, and (3) account for differences the physical processes being measured by different instrumentation (e.g., deep subsidence versus elevation change). Regarding the specific EGMS comparison, Thiéblemont et al (2024) noted that the “geodetic reference frame used to calibrate EGMS strongly influences coastal vertical land velocity estimates at the millimeter per year level and this needs to be considered with caution.” Here, we applied an affine transformation using the available GNSS dataset with temporal range from 2016-2023 utilized in our analysis to transform the EGMS vertical velocity estimates to the IGS14 Reference Frame to enable comparison. Note that we do not include this InSAR comparison in our study, but present it here per the reviewers' request, while emphasizing that such comparative approaches should not be used for robust accuracy assessment. Robust accuracy assessment for VLM (between InSAR products or other instrumentation) or other environmental variables must meet the above criteria.

The comparisons for the four European deltas are shown below, demonstrating a correlation (R) between 0.5 – 0.9, indicating moderate to great agreement after reference frame transformation.

Figure 2: Bivariate plot comparing EGMS VLM for 2019-2023 with VLM from this study for 2016-2023 for (a) Po, (b) Rhine, (c) Vistula, and (d) Rhone deltas. The correlation, R values are shown in the figures.

Additionally, comparison of the mean rates of collocated and non-collocated pixels for each delta shows differences of less than 1 mm per year and 2 mm per year, respectively:

Delta	EGMS*	This study (collocated pixels)*	This study (non-collocated pixels)*
Rhine	-2.22	-1.91 (199,073)	-2.67 (413,785)
Rhone	-1.26	-1.56 (68,641)	-3.01 (141,228)
Po	-4.30	-4.13 (20,634)	-6.06 (99,702)
Vistula	-4.21	-4.52 (51,114)	-5.47 (207,511)

*Values are in mm per year, number of pixels are shown in parentheses.

The greater differences between the mean VLM rates in our study (non-collocated pixels) and EGMS after this reference frame adjustment likely reflects the inclusion of additional distributed pixels (2 – 5 times greater) over urban areas, wetlands, and agricultural areas not captured in the EGMS dataset (further discussed below). Note that the spatial resolution of both datasets: 75m for this study compared to 100m or more for EGMS dataset may also contribute to the differences in pixel density. The EGMS dataset also utilizes different spatial sampling for non-urban areas (see EGMS whitepaper), reducing its applicability for non-urban regions. Furthermore, our study uses rates from 2016-2023, while the EGMS dataset used here covers 2019-2023, representing different temporal periods that may further contribute to observed differences. Below we plot the VLM for Rhone delta from this study and the EGMS data. Note higher level speckle noise in the EGMS dataset, compared to our analysis, which applied robust wavelet-based temporal filtering to minimize uncertainties of multi-looked pixels (see methods section and refs. 68–71, and 76).

Figure 3: Spatial VLM from (a) EGMS and (b) this study.

For the Rhone delta, a big part of the high subsidence values 0.5 cm/yr to 1 cm/yr appears over agricultural fields. It is unclear how you identified the surfaces suitable for ground point deformation. Many surfaces lack points in the EGMS because the service obtains data from most reliable and coherent areas like man-made structures or bare stable surfaces. How reliable is calculated VLM in surfaces that are heavily vegetated or subjected to agricultural use ? You mention using the (WabInSAR) algorithm that handles these challenging surfaces but it should be more clearly shown how this algorithm separates true deformation from vegetation-induced noise for example.

We thank the Reviewers for their comments. To clarify this point, the EGMS is a product created by a consortium comprising companies e-GEOS, TRE Altamira, NORCE, and GAF, as well as several subcontractors, such as the German Aerospace Center (DLR). These partners operated their own processing chains, based on advanced PS and DS InSAR processing techniques. While all of them demonstrated the ability to create high-quality InSAR deformation maps, it is well-established that each processing technique suffers from some limitations, mainly resulting in a different number of high-quality pixels and overall uncertainties depending on the dataset's characteristics and terrain. To perform the validation, overlaps between adjacent processing areas were used to check and ensure seamless harmonization between the adjacent S-1 tracks and the different processing chains. To achieve a uniform quality, a significant emphasis is placed on urban areas, built environments, and land areas with minimal vegetation; otherwise, the results from DS and PS techniques could be very different in the number of selected pixels and possibly contain errors. That is why the EGMS data set has a much smaller number of pixels and they are particularly sparse over agricultural areas. In the WabInSAR algorithm, we develop a novel pixel selection technique that accounts for both PS and DS pixels, making it applicable to all types of terrains (see Lee and Shirzaei, 2021, RSE). In this method, we apply thresholds that allow us to discard low-coherence and noisy pixels, obtaining a set of pixels that carry reliable phase information suitable for generating maps of the LOS deformation field with high accuracy and precision. Our validation tests in Lee & Shirzaei (2021) show that this method yields a set of reliable pixels that are significantly larger than that of PS or DS alone. Additionally, as described in lines 730 – 733, we screened the initial set of interferograms based on their coherence stability to exclude interferograms with high coherence variability. This additional statistical screening minimizes the loss of pixels due to coherence degradation and ensures that we retain the maximum number of reliable scatterers by maintaining consistently high average coherence across the interferogram stack. As observed above in the table above, the WabInSAR algorithm – optimized for non-urban areas retains more backscattered pixels across deltas. While we do retain more data points, we have sparse pixels over agricultural fields compared to other areas, with the plotting scale giving a false representation of uniformity or pixel density over these areas. The EGMS dataset, on the other hand, employs different processing parameters and excludes more pixels in nonurban areas. Additionally, the higher subsidence values observed over agricultural areas in the Rhone delta likely represent elevation loss due to processes such as drainage-induced compaction or groundwater extraction.

Also, the manuscript does not seem to provide methods on how the discrete VLM measurements from points were interpreted to produce the continuous velocity map grids

shown in the figures. Considering the apparent smoothness in the figures, the mentioned 75 m-resolution seems an overstatement, with the grid data being inherently much smoother.

We thank the Reviewer for their comments. We clarify that our InSAR-derived vertical land motion (VLM) rates were not spatially smoothed during processing or post-processing. The apparent smoothness of the maps arises from the visualization scale, the color ramp used for display purposes, and the scale of the image (high resolution VLM product at 75m resolution are provided in the repository). We adopted a uniform colormap across all deltas with a truncated range (typically -20 to +5 mm/year) to emphasize the spatial distribution of significant subsidence noted in some deltas while avoiding rainbow colormaps, which are discouraged in scientific publishing due to their perceptual artifacts. The maps appear smooth in areas where VLM values fall within similar ranges, reflecting the actual spatial distribution of coherent pixel measurements at native 75m resolution rather than interpolated data (see the comparison with a different colormap below). Additionally, the rendered pixel appearance in our figure generation may contribute to the visual impression of smoothness when viewed at manuscript scale.

[REDACTED]

Figure 4: Spatial VLM using color range of (a) -5 to +5 mm per year and (b) -20 to +5 mm per year.

Datasets and their robustness

Sediment flux data

You use sediment flux changes from the difference between pristine (without humans) and disturbed fluxes (with humans) based on ref 32. This is highly debatable and simplistic, if we are to accept that there is indeed a difference in the type of sediment supply in these deltas between 'pristine' and 'human influence' over the period of subsidence analysis (last decade or so: 2014-2023). Ref. 32 further proposes datasets that are fraught with uncorrected errors (see

Zainescu, F. et al., 2023. Concerns about data linking delta land gain to human action. Nature, 614, E20–E25. <https://doi.org/10.1038/s41586-022-05624-x>

Another point regarding sediment flux is that of in-situ organic production in deltas with large wetland systems which is not mentioned. We are not sure how important this can be (we are not aware of any global synthesis) but mention should at least be made of this, especially given its added influence on (differential) compaction and subsidence.

We thank the Reviewers for these important points regarding the limitations of sediment flux data. We acknowledge the concerns raised about the Nienhuis et al. (2020) dataset in Zainescu et al. (2023), and we agree that the pristine vs. disturbed flux framework represents a simplified approach to characterizing the complex changes in sediment delivery. We note that our analysis utilizes percentage changes between these conditions in the 40 deltas, specifically to assess the relative direction and magnitude of human influence on sediment supply, rather than absolute flux rates or detailed process mechanisms. We acknowledge this limitation in our analysis and have added caveats about the simplified nature of this dataset in lines 980 – 984.

Regarding in-situ organic production in deltaic wetland systems, we thank the Reviewers for highlighting this important process. We are also not aware of global syntheses that quantify their contribution to elevation change. We acknowledge that organic matter production and subsequent decomposition/compaction could influence differential subsidence patterns in deltas with extensive wetland systems. We have added a brief mention of this process as an additional factor that may contribute to elevation change dynamics in organic-rich deltaic environments.

-GRACE data processing and suitability

SWS (Surface Water Storage) is included in the GRACE data. The authors could not remove the SWS from Total Water Storage (TWS) to isolate the GWS. However, in large deltas fed by major rivers and subject to large seasonal flooding (e.g., Amazon, Mekong) the SWS signal can be quite substantial; hence the statement that "GWS is probably the primary signal" is an assumption, and this nuance needs to be critically examined. Although in some deltas the GWS signal can be very strong (as in the Chao Phraya), perhaps this contributes to the very low correlation between the GWS and VLM, essentially indicating noise.

We thank the Reviewer for this valuable comment. We agree that in flood-prone deltas surface water storage (SWS) can contribute significantly to total water storage (TWS), and that assuming groundwater storage (GWS) as the dominant GRACE signal without disaggregation may lead to attribution uncertainty. In response, we have now refined our GRACE-based GWS estimates by subtracting surface water storage derived from the WaterGAP Global Hydrology Model (WGHM), thereby better isolating the groundwater signal. Additionally, we validated these revised GWS trends against available in situ well data for 10 deltas. While absolute values differ due to spatial scale mismatch between point-based observations and GRACE footprints, the temporal trends show reasonable agreement, supporting the representativeness of our GWS estimates for broader regional dynamics. Accordingly, we have revised the manuscript to clarify this methodological update and to temper language around global GWS attribution. We have also added a limitations section in lines 336 – 357.

Appropriate use of statistics and treatment of uncertainties

The authors first employ simple and multiple regression analyses that fail to convey the complexity of the relationships and drivers of subsidence, and then used more sophisticated non-linear models based on a Random Forest machine-learning framework that turned out to be much more robust in extracting interdependencies among the various parameters. To go further and resolve delta-specific subsidence drivers, they applied Local Interpretable Model-Agnostic Explanations (LIME), derived from Explainable Artificial Intelligence. Altogether mindful of the complex subject of delta subsidence, the authors have gone at lengths to render their results robust, coherent and useful. We have, however, a number of concerns.

We thank the Reviewers for their comments and have responded to the specific concerns raised below.

- Machine Learning Application

Small Sample Size (N=40): Although RF may be able to better accommodate smaller sets than many other learning methods, a sample size of N=40 (then later N=30) is quite modest for training a model with global application. The notable standard deviations reported for feature importance (0.4 ± 0.3 for GWS) indicates model instability, and high sensitivity to the specific subsets of deltas used in training. This may be an inherent limitation, but the authors should be more clear in communicating the implications this has on the model robustness and its potential generalisation.

We thank the Reviewer for this important point. We acknowledge that N=40 (and subsequently N=30 (now 28) for LIME interpretation) constitutes a modest sample size for machine learning applications. However, our objective was not to build a predictive model with global generalization, but to explore the relative influence of three key anthropogenic drivers within a specific and spatially diverse sample of deltas (see lines 285 – 288). We agree that the reported standard deviation in feature importance for GWS (0.4 ± 0.3) reflects variability in model outcomes across subsampled training sets. Rather than indicating instability per se, this variability highlights the spatial heterogeneity in driver influence across deltaic systems. To quantify and transparently communicate this uncertainty, we employed a Monte Carlo cross-validation framework using a range of holdout fractions, which allowed us to report mean performance metrics and associated variability. The standard deviation served as the empirical and methodological basis for applying LIME to identify delta-specific drivers, rather than relying solely on global feature importance scores. We have clarified in the manuscript that our findings should be interpreted as insights into anthropogenic driver relationships for this specific set of deltaic systems, rather than globally generalizable predictions. We have also made the limitations of the modest sample size more explicit in our discussion of model robustness.

Filtering by LIME Fidelity

The authors excluded 10 deltas (25% of the original dataset) where the local LIME models did not reach a fidelity of $R^2 > 0.5$. This filtering should provide improved reported performance metrics of the retained deltas, but may have introduced a selection bias. Why were the models failing for these 10 deltas? How are they distinct? By removing them from the analysis, the

conclusions of the paper about "global" drivers become restricted to, technically, only 30 deltas for which the models performed well. Therefore, the limitations of a small sample size should be acknowledged more directly and the claims of the model's generalizability tempered.

We thank the Reviewer for this important clarification. Yes, excluding 10 deltas from the LIME interpretation stage reduces the sample to 30 deltas and improves the RF model performance. Note that LIME was not a different modeling outcome but was used to explain the RF model. The intent of this exclusion was not to improve model performance metrics but to ensure interpretative integrity. Additionally, the improved R^2 value was not obtained from a rerun of the model, but rather from repeating the fitting of the predicted versus actual values without including the 10 (now 12) deltas. Specifically, we excluded deltas where local surrogate models yielded LIME $R^2 < 0.5$, indicating insufficient fidelity in approximating the complex Random Forest model. In such cases, attempting to assign a specific predictor influence n score based on low-fidelity LIME results may risk misleading conclusions. These 10 (or 12) deltas likely reflect systems where vertical land motion (VLM) is influenced by processes not well captured by our three-variable framework (such as natural processes or unrepresented human interventions) and therefore fall outside the explanatory capacity of the model. We now add a clarifying note for model interpretable fidelity to explain the refining of the dataset from 40 to 30 (or 28) deltas, and we temper claims of global generalizability accordingly throughout the manuscript. As indicated above, we also note in the revised text that while our 40-delta dataset covers a substantial proportion of global delta population, it does not represent the full spectrum of geomorphic, climatic, or socioeconomic conditions across all deltas worldwide.

Predictor Interdependence

The authors indicate moderate correlation between urban expansion and GWS. This should not be a big surprise, given that urbanization increases demand on groundwater systems. RF handles some collinearity, but the interdependency of factors makes the interpretation of feature importance less certain. The importance attributed to GWS may even overlap with the effects of urbanization to some degree. This distinction would be worth noting for discussion.

We thank the Reviewers for this observation. We acknowledge the interdependency between urban expansion and groundwater storage, which we briefly discuss in our manuscript in lines 263 – 266: 'urban expansion not only directly increases infrastructure loading but also indirectly elevates groundwater demand, compounding aquifer depletion and extraction-induced subsidence.' We agree that this interdependency makes feature importance interpretation more nuanced, as part of the variance attributed to GWS may indeed reflect urbanization effects. To better reflect this nuance, we have added a sentence in lines 324–328 acknowledging that elevation change in many deltas arises from overlapping anthropogenic pressures, rather than focusing on a single dominant driver. This is further supported by the LIME results and ternary plot in Figure 3b, which show that multiple interacting variables influence most deltas.

L641-642 : What were the exact challenges that could not be overcome in using synthetic aperture radar (SAR) imaging and interferometric analysis in the Danube, Orinoco, and Shatt-el-Arab deltas?

We thank the Reviewer for this question. The primary challenges included: (1) insufficient spatial overlap between adjacent SAR frames, resulting in fragmented coverage; (2) long temporal baselines between acquisitions, which exceeded coherence thresholds; (3) consistently low coherence caused by dense vegetation; and (4) limited SAR data availability during the study period. We have added these details to lines 695–696 of the revised manuscript.

Suppl Fig 8. A: Correlation R 0.44, in the plot. In lines 925-926 you mention a different value for essentially the same data: “However, this model yielded poor performance (correlation, $R = 0.22$; RMSE = 4.7 mm per year) (Figure 3a)”. Which is the correct value?: b,c,d,e,f) All correlations are either low or non-existent. Please check for significance (p values). Some correlations in e,f have data which look more like outliers, so correlations should be carefully interpreted (correlation with outliers removed in c seems more robust). For non-linear correlations, cubic, quadratic, or logarithmic fit are discretionarily used. There should be an underlying mechanistic reason why one model is chosen over the other. For example what sense does it make for a polynomial (quadratic) to be fitted to plot b? This would imply that increasing sediment flux would lead to negative VLM; contrary to the expected logical outcome of increased sediment input.

We thank the Reviewers for the opportunity to clarify this. The different correlation (R) values refer to different analyses. Supplementary Figure 8a shows the bivariate correlation between VLM and GWS alone ($R = 0.44$), whereas the $R = 0.22$ (or $R^2 = 0.1$) mentioned in lines 925-926 (1038-1039) refers to the multilinear regression (MLR) model using all three predictors simultaneously (as shown in Figure 3a). In lines 961 to 1023, we discuss the possible relationship between each predictor and VLM, and between the different predictors. We have now added clarifying text to Supplementary Figure 8 and lines 987.

We agree that some correlations are moderate to non-existent, which we acknowledge in lines 961 – 1023. In Supplementary Figure 8, we did not advocate for any specific fit or choose one model over another, the figure is intended as an exploratory visualization rather than a basis for inferential statistics. We presented multiple fit options (linear, quadratic, logarithmic) to demonstrate the varied and often nonlinear relationships between individual predictors and VLM. In some cases, linear fits performed best; in others, quadratic or logarithmic fits were more appropriate, or none.

To address the reviewer's specific question about the quadratic fit in panel b. While we do not propose a mechanistic model for this, the Exner equation (Paola & Voller, 2005) that relates sediment flux and elevation change in fluvial systems predicts a quadratic relationship between sediment flux and land elevation gain. So, in places where the quadratic relation holds the highest correlation, one can speculate that land elevation gain may dominate the observed VLM. Mechanistically, both strong increases and decreases in sediment flux may be associated with enhanced elevation loss on deltas—increased flux may promote rapid loading and compaction in young deltas, while decreased flux in mature systems may promote elevation loss through other mechanisms. However, we emphasize that this exploratory visualization does not constitute a mechanistic model, and the apparent relationship likely reflects complex interplay of multiple factors rather than direct causation.

We included this figure and analysis specifically to justify why a Random Forest approach, which captures complex, nonlinear interactions without requiring priori assumptions about functional relationships, was the most suitable method for our multi-variable analysis. The poor performance of simple linear/quadratic models (both bivariate and multivariate) reinforced our decision to use machine learning approaches that can handle the complex, non-additive relationships inherent in deltaic systems.

Conclusions: robustness, validity, reliability

The study highlights a suite of potential pathways for analysing delta subsidence and vulnerability that is relatively robust, valid and reliable, inasmuch as this reliability is constrained by the quality of the data, delta feature heterogeneity, and uncertainties arising from more or less strong human impacts. We are just not sure whether you should not add a brief comment acknowledging a certain degree of uncertainty in all this, given the complexity of deltas individually and across a given delta, and the uneven quality of various datasets over large swathes of deltas.

We thank the Reviewers for their comments. We have added a paragraph acknowledging the inherent uncertainties in our analysis including the heterogeneous nature of the delta, the dataset and the model in lines 336 – 357. We also discuss the uncertainty in the RF model and indeed all ML models in the methods in lines 1122 – 1139.

Suggested improvements: experiments, data for possible revision

See items in previous sections.

Thank you! Responded above. We thank you for your feedback.

References: appropriate credit to previous work?

Yes, this is a well-referenced study. Justified suggestions for a handful of additional references are indicated in comments to the authors for minor revision.

Some additional references you may want to consider:

-Törnqvist TE and Blum MD (2024). What is coastal subsidence? Cambridge Prisms: Coastal Futures, 2, e2, 1–4. <https://doi.org/10.1017/cft.2024.1>

- Pedretti L, Giarola A, Korff M, Lambert J and Meisina C (2024) Comprehensive database of land subsidence in 143 major coastal cities around the world: overview of issues, causes, and future challenges. *Front. Earth Sci.* 12:1351581. doi: 10.3389/feart.2024.1351581

We thank the Reviewers for their comments and for suggesting these additional relevant references. We have added these two citations in the manuscript.

Clarity and context: lucidity of abstract/summary, appropriateness of abstract, introduction and conclusions

Well organized, well written and cannot be faulted in any way.

We thank the Reviewers for their positive feedback on our manuscript.

Further suggestions for revision

• Lines 48-50: I am not sure references 1 (2006) and 2 (2009) are pertinent regarding current (2025) global delta population, which is estimated at 350 to >500 M. These conflicting estimates (see also Nicholls, R.J., Adger, W.N., Hutton, C.W. & Hanson, S.E. (eds.). Deltas in the Anthropocene (Springer, 2020) might also need to be acknowledged, while stating that you are using Edmonds et al. (2020).

We thank the Reviewer for this important point. We have updated our population references to acknowledge the uncertainty in these global estimates. Explicitly referencing the 350 to 500 million people range in lines 50-52 and including the Edmonds et al. (2020) citation.

• Lines 64-65, 330-336: You refer here to deltas as 'low-lying landforms averaging less than 2 m above sea level'. Can you provide a reference? There are several areas in the manuscript where delta elevations are referred to, and accurate delta surface elevations are indeed a major gap in current vulnerability studies of deltas. Are these references to elevations derived from the DeltaDTM (reference 62)? If yes, then this reference needs to be cited in the main text.

We thank the Reviewer for pointing this out. Yes, this is the case, we have added the reference to DeltaDTM to support our statement about delta elevations in lines 67 and 377.

• Lines 82-86 + Lines 536-541 (reference 26): Please break down this sentence as it is rather confusing as regards the two references. Ref. 26 is on sea level and 25 on issues in the five megadeltas. Why is the comment on definition of sea level attached to this reference in the reference list?

We thank the Reviewer for this feedback. We have broken down the complex sentences in lines 82-86 for clarity and removed reference 26 (added to define relative sea level rise) from the reference list.

• Line 108: in 40 deltas globally. Globally is superfluous here.

We have removed the word 'globally' from Line 108.

• Line 168: You use a global delta area of 710,187 km² from Edmonds et al. (ref. 6). Again, estimates of global delta vary and are presumably robustly proposed in each study, such that this variation may need to be stated alongside your recourse to Edmonds et al. (2020). Syvitski et al. (2022), for instance, robustly calculated an area of 855,000 km².

Syvitski, J. et al. 2022. Large deltas, small deltas: Towards a more rigorous understanding of marine deltas. *Global & Planetary Change*, 2018, 103958.

<https://doi.org/10.1016/j.gloplacha.2022.103958>

We thank the Reviewer for the additional point about global delta area estimates. We have included this variation in line 168 – 170 of our manuscript citing both studies.

• Lines 246-247. A missing but relevant reference here that highlights delta complexity and draws attention to many facets of delta dynamics and global differences is Syvitski et al. (2022) cited above.

We thank the Reviewer for suggesting this relevant reference. We have added Syvitski et al. (2022) to the citation here.

• Lines 354-355: You state that your findings challenge the prevailing narrative framing climate-induced sea-level rise as the principal hazard in low-elevation coastal zones (ref 45). Low-elevation coastal zones cover well beyond the 1% of coast occupied by deltas, and most low-elevation coastal zones (beaches, beach-ridge plains, dunes, etc..) are not exposed to subsidence. This statement therefore needs to be reframed more cautiously.

We thank the Reviewer for this important clarification. We have reframed our statement in lines 404-408, acknowledging that our conclusions do not apply to all low-elevation coastal environments.

• Line 627: Global delta population 'was' estimated.

We have corrected the verb tense in line 680.

Florin Zainescu, Edward Anthony, Aix-en-Provence

References

Paola, C. and Voller, V.R., 2005. A generalized Exner equation for sediment mass balance. Journal of Geophysical Research: Earth Surface, 110.

Reviewer #3 (Remarks to the Author): (Reviewers comments in normal text, response to reviewers is in bold)

I have finished reviewing the manuscript titled “The Global Threat of Sinking Deltas” which addresses a critical and timely issue concerning the vulnerability of river deltas worldwide. It provides a comprehensive analysis of the drivers of subsidence and their possible implications for coastal populations, infrastructure, and ecosystems. Overall I am very positive about the work, the paper addresses an important topic, it is well written, with high quality figures and interesting methods.

Summary of the key results

The work underlines the importance of including subsidence in coastal adaptation planning. This is a key point and important finding.

Originality and significance:

The work is novel and significant. It builds on several previous studies that are properly cited in the manuscript

We sincerely thank the Reviewer for their positive and encouraging assessment of our work. We also appreciate the reviewer's insightful comments, which were useful to improve the clarity of our manuscript. Below, we provide a detailed point-by-point response.

Data & methodology:

The methods seem sound and the results are very well presented. I have to admit that I am not a remote sensing expert and even though I read thoroughly the methods, I may miss some important details. The part of the SAR Interferometric analysis is important, despite the detailed description I would expect the authors to provide code, but in case this is not possible, at least to provide more details about the environment the work is done (toolboxes, etc).

In general, I believe that sharing the code would be important for a work of such importance. I also think it would not be a big challenge since we are not referring to large modelling frameworks, but rather data analysis that I assume that could be easily packaged in a repository (e.g. the RF application).

We thank the Reviewer for their feedback and for highlighting the importance of code availability for reproducibility. We have added a code availability statement for both the SAR interferometric analysis and the random forest application in lines 1380 – 1382.

Appropriate use of statistics and treatment of uncertainties

One that is missing in the manuscript is a description of uncertainty. Starting from VLM, I would expect that such measurements would come with significant error, especially for such extensive areas as deltas. Figure 4c mentions some percentiles (and Figs 9, 11 of the SI), but I think it would be helpful to show confidence intervals for the important variables. The same applies for the Groundwater Storage Change, as I would be surprised if such calculations would not come with substantial uncertainty.

We thank the Reviewer for this important point. All measurement uncertainties are reported in the supplementary tables. We have now added uncertainty quantification where appropriate throughout the manuscript.

Conclusions:

I found the conclusions solid and the analysis robust

Suggested improvements:

Doesn't Fig 9a of the SI mean that the RF approach underestimates subsidence? Shouldn't these results discredit RF as a predictive tool and just use it to interpret the results? If yes why use RF and not other multivariate techniques including standard multivariate regression? I am not questioning the selection of the method, but I think the above should be discussed.

We thank the Reviewer for this important methodological question. We acknowledge that Figure 9a shows some underestimation at the highest subsidence rates (>8 mm/year) while other data points are generally distributed around the 1:1 line. This underestimation of values > 8 mm per year may be indicative that other natural or other human processes not captured by our three anthropogenic predictors (groundwater storage, sediment flux change, and urban expansion) contribute to extreme subsidence in certain deltas. We have now acknowledged this limitation in lines 280–283 of the revised manuscript. Furthermore, the limited number of extreme subsidence values in our dataset may reduce the model's sensitivity to such cases, which is a well-known challenge in machine learning when applied to imbalanced or underrepresented outcomes.

As stated in our methods (now added in lines 285 - 288 of the main manuscript), the primary objective for applying RF is not to predict subsidence rates, but rather to identify and extract key features that explain the dynamic relationships between anthropogenic drivers and subsidence across global deltas. We selected Random Forest over standard multivariate regression because our initial multilinear regression model performed poorly ($R^2 = 0.2 \pm 0.1$), failing to capture the nonlinear interactions between anthropogenic factors. RF effectively handles these complex, non-additive relationships and provides robust feature importance estimates, enabling us to identify which anthropogenic drivers dominate in each delta.

Figure 2a. Shouldn't the bars have a total of 100%?

We thank the Reviewer for this observation. The bars in Figure 2a do not total 100% because we omitted areas of uplift on each delta and only display the subsidence. We have added a clarifying note about this in the figure caption.

Check reference 26 (Gregory et al), page 14

We thank the Reviewer for pointing this out. The note was meant to clarify the terminology used in our study and has been removed from the reference list.

Figure 5. I think the 4 'types' of deltas should be explained in the caption

We thank the Reviewer for this suggestion. We have added explanations of the four delta categories to the Figure 5 caption for clarity.

References:

The authors give appropriate credit to previous work and to my understanding cite all previous major studies

Clarity and context:

I found the manuscript very well written

We thank the Reviewer for their comments. The manuscript has been improved by their insightful comments.

Reviewer #4 (Remarks to the Author): (Reviewers comments in normal text, response to reviewers is in bold, italicized text)

I co-reviewed this manuscript with one of the reviewers who provided the listed reports.

We thank you for your reviews and have addressed the co-reviews in review 2.

Reviewer #1 (Remarks to the Author): (Reviewers comments in normal text, response to reviewers is in bold)

The authors adequately addressed key review issues.

In my view, the manuscript is scientifically sound, transparent about scope, and broadly suitable for publication.

We sincerely thank the Reviewer for their excellent feedback on our manuscript.

Reviewers #2 (Remarks to the Author): (Reviewers comments in normal text, response to reviewers is in bold)

The authors have carried out the revisions to the letter. The revisions make for a robust manuscript that will be useful to the delta community. Our comments concern completing one reference, and very minor (syntax, punctuation) edits that may even be carried out at the proof stage.

We thank Drs. Florin Zainescu and Edward Anthony for their wonderful review of our manuscript. We have updated the manuscript as discussed below.

Reference 98: Zainescu, F. et al., Concerns about data linking delta land gain to human action. Nature, 614, E20–E25 (2023). <https://doi.org/10.1038/s41586-022-05624-x>

We thank the reviewers for catching this error. It has been updated in the manuscript.

Line 40: .. groundwater storage 'has' (in lieu of have) ***It has been corrected in the manuscript.***

Line 88: ...is just as, or more, influential than (punctuation) ***It has been corrected in the manuscript.***

Line 117: For consistency, to reflect .. (punctuation) ***It has been edited in the manuscript.***

Lines 333-334: ... floods that deposited sediments, are documented to now experience (correct experience) severe sediment deficits due to dams and levees, (add comma here) accelerating elevation loss ... ***It has been updated in the manuscript.***

Well done, once again!

Edward Anthony, Florin zainescu, Aix-en-Provence, France.

We sincerely thank you Drs. Florin Zainescu and Edward Anthony for your reviews. We enjoyed interacting with the comments and the detailed and excellent review comments, which were invaluable to getting the manuscript publication ready.

Reviewer #3 (Remarks to the Author): (Reviewers comments in normal text, response to reviewers is in bold)

I enjoyed reading the revised manuscript and the authors' interaction with the reviewers. The paper has been corrected substantially and its limitations are presented more transparently..

So I can recommend publication of the article

We thank the Reviewer for their valuable comments. We enjoyed responding to the reviewer's comments and engaging with the Reviewer.

Reviewer #4 (Remarks to the Author): (Reviewers comments in normal text, response to reviewers is in bold, italicized text)

I co-reviewed this manuscript with one of the reviewers who provided the listed reports.

We thank you again for your excellent reviews and have addressed the suggested minor edits in review 2.

Reviewer #5 (Remarks to the Author): (Reviewers comments in normal text, response to reviewers is in bold, italicized text)

Key Results

The authors of this manuscript present an extensive analysis of elevation changes and their main drivers in selected deltas around the globe. They reveal that approximately 54-65 % of the global deltas are sinking. They analyzed the role of three key anthropogenic drivers and found that groundwater storage loss is the primary driver of elevation changes in 35% of deltas, while groundwater storage, sediment flux decline and urban expansion rather have mixed influences in the other deltas. They show that subsidence is the dominant driver of relative sea level rise in many deltas today, and even under the worst-case, high-emission scenarios (SSP5-8.5), maximum subsidence rates in all deltas will exceed projected sea level rise rates in all deltas. Lastly, they provide a study on the varying adaptation readiness of the selected deltas.

Validity

The individual components of the analysis – the InSAR analysis of elevation changes in the deltas, the machine learning regression of the contribution of three key drivers, and the assessment the future impact of subsidence and sea level rise in deltas based on IPCC scenarios – are based on established and validated techniques, and the analysis seems to be extensive and thorough. I did not find any major flaws, but I have a couple of questions/discussion points which I give in the specific paragraphs below.

Originality and significance

I consider the analysis as an overdue and essential upgrade of the global delta analysis by Syvitski et al. (2009). The elevation change dataset in 40 deltas, achieved with InSAR, is impressive. Together with the analysis of associated anthropogenic drivers and the future impact of subsidence and sea level rise based on different emission scenarios, the analysis highlights the urgency of the overall issue and hopefully contributes to the initialization or further development of adaptation policies and appropriate countermeasures.

Reference:

Syvitski, J., Kettner, A., Overeem, I. et al. Sinking deltas due to human activities. *Nature Geosci* 2, 681–686 (2009). <https://doi.org/10.1038/ngeo629>

We thank you Dr. Nils Dörr for your excellent reviews, which helped in clarifying the InSAR methodology and strengthening the manuscript. Below, we provide detailed responses to each comment.

Data & methodology

InSAR analysis:

- On page 18, lines 709-711, you describe that you multi-looked the SAR data. Later in line 733, you describe a coherence threshold for persistent scatterers. How did you identify and analyze

persistent scatterers when the SAR data has previously been multi-looked with a boxcar? Furthermore, how did you incorporate persistent scatterers in the final aggregated results?

We thank the reviewer for their comments. To clarify, we did not use a coherence threshold for persistent scatterers (PS). Instead, PS were identified using an amplitude dispersion index, while the coherence threshold was applied specifically to distributed scatterers (DS).

Our approach follows the framework established by Ref. 73 (Lee & Shirzaei, 2023). After multi-looked the SAR data, we applied a dual-criteria scatterer selection strategy. Persistent scatterers were identified using an amplitude dispersion threshold <0.35 . Distributed scatterers were evaluated using temporal mean coherence across the interferogram stack. DS pixels with average coherence >0.70 and low temporal coherence variance were retained. These DS pixels were then assigned to Voronoi cells constructed around adjacent PS pixels. Within each cell, a Fisher's F-test evaluated the statistical similarity of amplitude behavior between DS and PS pixels. DS pixels passing this test (indicating stable, PS-like backscatter characteristics) were reclassified as reliable distributed scatterers. This filtering ensures that the final dataset comprises only scatterers with sufficient phase stability for time-series analysis, whether they exhibit point-target or distributed-scatterer characteristics. Both PS and DS pixels meeting these criteria comprise the elite pixels. Please see Lee & Shirzaei, 2023 for further details.

To the reviewers' questions: (1) Multilooking sets the representation scale (~75 m) but does not preclude PS identification: a PS in our workflow is a temporally stable, high-SNR multi-look pixel whose amplitude is low-dispersion across the time series. This is standard practice when moving from object-scale scatterers to neighborhood-scale "persistent" cells in areas where single-look speckle would otherwise suppress stability. We acknowledge that in the original PS InSAR method, it was necessary to avoid multi-looked to achieve the highest density of PS pixels. However, later advances, such as SQUEE-SAR (Ferretti et al., 2011) and WabInSAR, resolved this issue by integrating both PS and DS pixels.

(2) How PS contributes to aggregated results. After classification, PS and vetted DS are treated uniformly as elite pixels.

Reference:

Lee, J.-C. and Shirzaei, M. (2023). Novel algorithms for pair and pixel selection and atmospheric error correction in multitemporal InSAR. Remote Sensing of Environment, 286, 113447.

• I am surprised by the apparent quite dense coverage of your InSAR results in almost all deltas, as seen in the figures. In the Mekong Delta, for example, your results seem to even cover parts of forested areas (such as parts of the U Minh Forest), and areas which are widely covered by agriculture and even aquacultural farms with many water bodies. Is this just a matter of the visualization? If not, how did you identify evaluable pixels in such areas, and which kind of phase filtering / phase optimization approaches did you apply for distributed scatterers?

We thank the reviewer for this observation. The apparently dense coverage in forested and aquacultural areas is a matter of visualization. Figure 1 shows a zoomed-in map of the U Minh Forest region. As this reveals, coverage in forested areas is sparse and

primarily limited to forest edges, roads, and cleared patches rather than continuous canopy cover.

Figure 1. Vertical Land Motion (VLM) of Mekong Delta zoomed into parts of the U Minh Forest.

• Multi-temporal multi-look InSAR can be influenced by phase biases, whose magnitude depends on the temporal baseline distribution of selected interferograms and the land cover (Ansari et al., 2021; Maghsoudi et al., 2022). A large percentage of the analyzed areas is covered by agricultural fields. Devlin and Lohman (2025) have shown in a Sentinel-1 InSAR study that agricultural fields in the San Joaquin Valley, CA, can be associated with a bias of ~2–4 cm/yr of apparent subsidence, when only analyzing short-term interferograms of 6 days temporal baselines. You describe that first allowed interferograms with a temporal baseline of up to 300 days, but later you sorted out interferograms based on the estimated coherence, “while maintaining a 50% temporal baseline coverage”. What does this exactly mean? Did you consider the risk of phase bias while selecting the interferograms in your study?

References:

Ansari, H., De Zan, F., & Parizzi, A. (2021). Study of systematic bias in measuring surface deformation with SAR interferometry. *IEEE Transactions on Geoscience and Remote Sensing*, 59(2), 1285-1301.

Maghsoudi, Y., Hooper, A. J., Wright, T. J., Lazecky, M., & Ansari, H. (2022). Characterizing and correcting phase biases in short-term, multilooked interferograms. *Remote Sensing of Environment*, 275, 113022.

Devlin, K. R., & Lohman, R. B. (2025). Evaluation of vegetation bias in InSAR time series for agricultural areas within the San Joaquin Valley, CA. *Earth and Space Science*, 12, e2024EA004062.

We appreciate this important comment, which has been the subject of scientific debate for a while (see also Manunta et al. (2019) and De Luca et al. (2021)). As discussed in Lee and Shirzaei (2021), we are aware of this problem and have implemented a dyadic down sampling method coupled with Delaunay triangulation when selecting interferometric pairs. This procedure allows for creating an ensemble of interferograms with baselines spanning from a few days to several months, with a uniform distribution. Additionally, the selection pairs that form a closed path (triangles) enable the adjustment of the so-called “phase closure error” and prevent it from accumulating. Our validation tests have demonstrated that our processing framework is robust and not affected by the errors mentioned by the reviewer.

Regarding interferogram selection, "maintaining 50% temporal baseline coverage" means we retained interferograms that collectively span at least 50% of the available temporal baselines after applying our coherence-based quality screening. Specifically, we first generated interferograms with temporal baselines up to 300 days. We then screened this initial set based on coherence stability, excluding interferograms with high spatial variance in coherence, which often indicates phase noise or decorrelation issues. However, this screening removed only 7% of interferograms ensuring that the remaining network maintained sufficient temporal connectivity by requiring at least 50% coverage of the original temporal baseline distribution. This prevents temporal gaps that could introduce bias or limit our ability to resolve time-dependent deformation.

• Multi-looking leads to a smoothing of displacements within windows. De Wit et al. (2021) have shown that there is significant differential subsidence on short spatial distances in urban areas in the Mekong Delta, and I assume this also applies to other deltas. The reason behind is the occurrence of different foundation depths of buildings, loading effects, differences in historical land use, etc. As a result, aggregating vertical motion rates mixes motion signals from different depths and sources. Could you discuss if this is a relevant issue/uncertainty in your study?

Reference:

de Wit, K., Lexmond, B. R., Stouthamer, E., Neussner, O., Dörr, N., Schenk, A., & Minderhoud, P. S. (2021). Identifying causes of urban differential subsidence in the Vietnamese Mekong Delta by combining InSAR and field observations. *Remote Sensing*, 13(2), 189.

We thank the reviewer for their comment. While we agree that averaging within look windows may mix signals from structures with different foundations or loading histories, multi-looking is a standard and necessary procedure applied when the goal is not to resolve meter-scale differential motion but rather to characterize regional-scale deformation patterns. For our study area spanning hundreds of square kilometers across 40 deltas, a single-look product would be computationally prohibitive and unnecessary for mapping delta-wide subsidence and elevation change trends. Multi-looking (32:6 range:azimuth, yielding ~75 m resolution) reduces speckle noise and improves signal-to-

noise-ratio across highly dynamic surfaces, which is essential for obtaining reliable measurements in the challenging environments of vegetated deltas.

Conceptually, multi-looking improves precision (lower phase noise) at the cost of spatial resolution; it does not inflate the formal time-series uncertainty of a given averaged pixel, but it changes the representativeness scale of that pixel from object- to neighborhood-scale. Our scientific focus is on delta-scale patterns and drivers, rather than parcel-level contrasts, so the ~75 m areal-average velocity is the quantity of interest for cross-delta comparisons and identifying relationships between anthropogenic drivers and subsidence. In fact, any within-window mixing would bias against detecting sharp extremes, making our estimates conservative with respect to localized highs and lows. We acknowledge that in densely built urban cores, our measurements cannot resolve the building-to-building variations documented in high-resolution studies, such as De Wit et al. (2021). Therefore, our results in such areas should be interpreted as neighborhood-scale averages.

• Page 5, lines 191-193: “In some deltas (e.g., Wouri, Zambezi, Indus, Ciliwung, and Yellow), the observed uplift or elevation gaining parts correlate with patterns of sediment deposition and horizontal land motion”. As land subsidence is a process originating in the subsurface, the surface backscatter characteristics do not necessarily change during the subsidence process. On the contrary, sedimentation is a process which alters the surface backscatter characteristics, forcing decorrelation of the InSAR measurements. I’m not challenging your results, I am really interested in these findings. I would be interested to see (1) if you can provide/discuss similar literature which describes the measurement of elevation gain by sedimentation with InSAR, (2) a short analysis whether the InSAR coherence is statistically different in areas which are characterized by elevation gain than in others.

We thank the reviewer for this insightful question regarding the measurement of sedimentation-driven elevation gain using InSAR. The reviewer correctly notes that active sedimentation can alter surface backscattering properties and cause decorrelation, which is indeed a challenge for InSAR measurements in highly dynamic depositional environments. In the literature, several decorrelation models have been proposed (e.g., Zebker et al., 1992), which are widely used to assess the impact of temporal change in surface properties on interferometric coherence. Particularly, Zebker et al. (1992) show that for a significant drop in coherence, the surface motion (including elevation gain) must be several centimeters between two acquisitions. Given the 6-12 day S1 repeat time and the far slower rate of uplift we detected here, a significant decorrelation error is not expected. Regarding coherence differences, we conducted the analysis suggested by the reviewer, comparing the average temporal coherence in areas showing elevation gain (positive VLM > 0 mm/year) with those showing subsidence (negative VLM < 0 mm/year) across the Mekong and Yellow River deltas. We found that areas with elevation gain show statistically similar average coherence (Mekong mean = 0.71 ± 0.10 , Yellow mean = 0.77 ± 0.07) compared to subsiding areas (Mekong mean = 0.69 ± 0.11 , Yellow mean = 0.75 ± 0.08), indicating the reliability of the measurements (Figure 2). In fact, the uplifting areas have slightly higher average coherence compared to subsiding regions.

[REDACTED]

Figure 2. Average temporal coherence for Mekong (top left) and Yellow (top right) deltas. Boxplots of average coherence of subsiding and uplifting regions in Mekong (bottom left) and Yellow (bottom right) deltas.

Regarding literature on InSAR measurement of elevation gain in deltas, recent studies documenting sediment-induced elevation gain in deltaic settings remain scarce. Huang and Sinclair (2025) recently demonstrated the use of InSAR measurements to assess sediment aggradation rates in Himalayan River channels, utilizing differential residual topographic phase, although this study focused on proximal fluvial environments rather than delta plains. Most InSAR delta studies focus predominantly on subsidence while ignoring or not addressing uplift regions. For example, in Minderhoud et al. (2020, Figure 1; reproduced below), localized zones of positive VLM are visible in the Mekong Delta, particularly along some river channels and delta lobes (a pattern comparable to this study), yet remain unaddressed in the interpretation. Similarly, InSAR studies in the Yellow River Delta (Wang et al., 2022, Figure shown below) reveal uplift zones across most regions, with the highest rates observed in recently prograding delta lobes. However, these patterns are not discussed in the published interpretation.

In our view, sediment accretion is a plausible, though not exclusive, explanation for such positive VLMs where coherence is maintained, particularly in distributary channels and young lobes. We also note alternative contributors (e.g., poroelastic rebound linked to

groundwater recharge/management, anthropogenic ground raising, shallow fault-related deformation in rapidly loaded strata). Discriminating among these requires complementary data (e.g., stratigraphy, construction records, or fault mapping) beyond InSAR alone. We have modified this section to include alternate hypothesis.

References:

Huang, J. and Sinclair, H. D. (2025). Sediment aggradation rates in Himalayan rivers revealed through the InSAR differential residual topographic phase. Earth Surface Dynamics, 13, 531-547.

Minderhoud, P. S., Hlavacova, I., Kolomaznik, J., & Neussner, O. (2020). Towards unraveling total subsidence of a mega-delta—the potential of new PS InSAR data for the Mekong delta. Proceedings of the International Association of Hydrological Sciences, 382, 327-332.

Wang, G., Li, P., Li, Z., Liang, C., & Wang, H. (2022). Coastal subsidence detection and characterization caused by brine mining over the Yellow River Delta using time series InSAR and PCA. International Journal of Applied Earth Observation and Geoinformation, 114, 103077.

Zebker, H., & Villasenor, J. (1992). Decorrelation in interferometric radar echoes. IEEE Trans. Geosci. Rem. Sensing, 30, 950-959

[REDACTED]

[REDACTED]

Figure 3. Vertical Land Motion (VLM) of Top: Mekong Delta (Minderhoud et al., 2020). Bottom: Yellow River Delta (Wang et al., 2022).

• On page 19, lines 736-737 you describe how you estimated and corrected orbital errors. The mentioned approach in reference 76 estimates orbital errors by assuming that no displacement signals exhibit long wavelengths. Can you be sure that this is the case in all of the study areas?

We appreciate the comment. Note that the Method of Ref. 76 is designed to precisely extract and remove the signal component with the longest spatial wavelength, given the dimension of the SAR image. This procedure removes long-wavelength horizontal motions, solid Earth tides, ocean tide loading, and ionospheric disturbances, which is our desire, as well as long-wavelength vertical motions. The long-wavelength vertical motion mainly includes a broad signal due to GIA and elastic loading effects, which must be restored. To this end, we utilize coarse-resolution model of global VLM and apply an affine transformation, as described in the Methods section. This procedure allows for the restoration of most of the long-wavelength VLM that was removed, as evident from validation against independent GNSS observations (previously supplementary Fig. 11; now supplementary Fig. 4).

• Page 19, line 737-738: “[...] minimized the effects of topography-correlated components of atmospheric phase delay [...]” Topography-correlated atmospheric disturbances are rather small in flat deltas, while turbulent tropospheric effects can be really strong due to the flat, maritime location of deltas, as well as the frequent low latitude. Have you applied any approach to reduce effects from the turbulent troposphere? Furthermore, did you account for the following long-wave effects, which can have significant contributions to large-scale InSAR displacement estimates: solid earth tides, ocean tide loading, ionospheric disturbances?

Another great comment, and we apologize for the lack of details on InSAR processing and filtering steps. As we used already published and carefully vetted methods, we

assumed that citing relevant papers would be sufficient. To clarify, we implement a two-step atm error correction procedure. Step 1 mimics the fractal structure of spatially correlated ATM errors to identify and remove any existing errors of this characteristic (Lee and Shirzaei, 2021). In flat areas, this filter applies little to no correction. Step 2 correction applies a low-pass filter that utilizes a continuous wavelet transform and employs a soft thresholding procedure to refine the wavelet coefficients, thereby estimating and removing the temporally uncorrelated component of the atmospheric error (Shirzaei, 2013). The combination of these two steps yields an InSAR time series that is corrected for both types of atm error.

Yes, we account for long-wave effects as explained above.

• Page 19, lines 764-766: Mention that G is the matrix which comprises the projection vectors

We thank the reviewer for this suggestion. We have clarified this in the manuscript (line 764-766). G is now explicitly defined as the design matrix which comprises the projection vectors.

• Page 20, line 793: How was the velocity reference point for the InSAR analyses selected? Did you select them in the near vicinity of the GNSS stations which were used later for the alignment? How did you mathematically align the reference points to the GNSS-derived rates? How did you do this if several GNSS stations were present in the study areas? Did you incorporate estimate noise statistics into the alignment?

We thank the reviewer for their comments and suggestions. We have clarified and expanded the Methods section to address all points raised. Regarding reference point selection, for the 17 deltas with GNSS coverage, reference points were selected at GNSS station locations within the processed SAR frames. This ensures co-location between the InSAR reference, and the point used for IGS14 alignment. We aligned the reference frames by calculating the offset between the InSAR-derived vertical velocity at the reference point and the corresponding GNSS vertical velocity. Then, we applied this offset to transform all InSAR velocities in that delta to the IGS14 reference frame using an affine transformation.

When multiple GNSS stations were present in a delta, one co-located with the reference point was used for reference frame transformation. Additional GNSS stations were used as independent validation points, as shown in Supplementary Figure 11 (previous) or supplementary Fig. 4 (currently). Else, we would be validating against points utilized in the reference frame alignment.

Regarding noise statistics, yes, we incorporated estimated uncertainties into the alignment. The uncertainty in the final transformed velocity was estimated by propagating both the InSAR velocity uncertainty (from the reweighted least-squares inversion) and the GNSS velocity uncertainty through standard error propagation. These clarifications have been added to the revised manuscript.

• Page 20, line 794: 17 deltas were mentioned before in line 790, why only 16 now?

We thank the reviewer for catching this typo. This has been corrected. 17 deltas had GNSS data available for reference frame transformation (line 790 and line 794 now both state 17).

• Page 20, line 795: I've exemplarily checked the GNSS stations from the Nevada Geodetic Laboratory in the Mekong Delta, and found that not a single one covers the whole time span between 2016 and 2023, which was analyzed in this study. How did you generally select appropriate GNSS Stations? Also, please describe how you handled data gaps in GNSS time series and the presence of very few stations in large study areas?

We thank the reviewer for their comments. Our selection criteria prioritized stations with sufficient data density rather than requiring complete temporal overlap. Specifically, we included GNSS stations with at least 70% temporal coverage within the InSAR observation period and a minimum of 500 daily observations to ensure robust velocity estimation. We have clarified this in the methods section. For the velocity estimation, we applied the MIDAS (Median Interannual Difference Adjusted for Skewness) robust trend estimator (Blewitt et al., 2016), which is specifically designed to handle data gaps, outliers, and seasonality in geodetic time series without requiring gap-filling or detrending.

Reference:

Blewitt, G., Kreemer, C., Hammond, W. C., & Gazeaux, J. (2016). MIDAS robust trend estimator for accurate GPS station velocities without step detection. Journal of Geophysical Research: Solid Earth, 121(3), 2054-2068.

GWS:

• Page 24, lines 944-945: I exemplarily checked both sources for groundwater level data in the Mekong Delta, but could not find the data during my rapid search. Where can I find the groundwater data from the Mekong Delta included in Supp. Figure 14?

We thank the reviewer for their comment. The groundwater data for the Mekong data was obtained from Jasechko et al. (2024). In Figure 1a (Jasechko et al. (2024)), note that we can already see points over the delta at this scale. The data can be obtained by downloading the source file for the figure. When plotted we obtain the figure below.

[REDACTED]

[REDACTED]

Figure 4. Top: Global groundwater level trends from Jasechko et al. (2024). Zoomed in area of the Mekong delta showing groundwater level dataset used in this study. Note that positive trends indicates lowering groundwater levels.

Urban Expansion:

• Page 25, lines 1010-1012: Have you tested the absolute urban land conversion in km² between 2000 and 2020 for the analysis? I guess the urban fraction change is hard to compare if the absolute urban area in 2000 was largely different among the studied deltas.

We thank the reviewer for their comments. We tested both the absolute change in urban land area (km²) between 2000 and 2020 and the fractional change (%). Because the two metrics are strongly correlated, including both in the analysis would introduce collinearity. We therefore opted to use fractional change, which normalizes across deltas of very different sizes and facilitates direct global comparison. Additionally, the fractional change better captures the intensity of urbanization relative to each delta's baseline condition; a metric more conceptually aligned with how urban expansion might stress existing groundwater systems or alter subsurface loading conditions in proportion to the delta's capacity. We note that Supplementary Table 2 provides the absolute urban fraction values for 2000 and 2020 for readers interested in the absolute magnitudes.

Appropriate use of statistics and treatment of uncertainties

• Page 20, line 810-811: Can you briefly explain how you estimated the standard deviation of the land motion results?

The standard deviation of the land motion results arises from multiple sources, depending on the processing stage. For the initial line-of-sight (LOS) velocities, the standard deviation corresponds to the uncertainty of the regression slope from the reweighted least-squares fit. For the 27 deltas with both ascending and descending orbit coverage, vertical and horizontal velocity uncertainties were calculated through error propagation in the two-dimensional decomposition (Equation 3), which combined the LOS uncertainties from both geometries. For the 17 deltas with GNSS stations used for

reference frame alignment, an additional uncertainty term from the transformation to IGS14 was included by standard error propagation of both the InSAR-derived and GNSS velocity uncertainties.

• Page 20, line 814 “independent GNSS data”: i guess independent means that these are stations which previously have not been considered in the transformation of VLM rates from local to the global reference frame. Please clarify.

Yes, "independent GNSS data" refers to stations that were not used in the transformation of VLM rates from local to the global reference frame. For each of the 17 deltas with GNSS coverage, one station was used for the IGS14 alignment, while all additional GNSS stations within the SAR frames were utilized as independent validation points. This ensures that the validation assessment is not circular, but we are testing the InSAR accuracy against measurements that did not influence the reference frame transformation. This has been clarified in the revised manuscript.

• Page 21, line 815: What does “historical recorded rates” mean exactly? What is the time difference between the GNSS and the InSAR measurements? In case of non-linear subsidence, this might have a significant impact. This should be briefly discussed.

We thank the reviewer for their comment. "Historical recorded rates" refers to GNSS stations with long-term velocity estimates that span periods longer than or ending prior to our InSAR observation window (2014-2023). Note that this is different for the various datasets and the historical dataset. We acknowledge that temporal mismatches could introduce discrepancies if subsidence is non-linear. To address this, we conducted two separate validation analyses, as shown in Supplementary Figure 11: one using 122 GNSS stations with historical long-term rates (Supplementary Figure 11a, RMSE = 1.4 mm/year, $R = 0.7$), and another using 81 GNSS stations with time series overlapping at least 70% of the InSAR observation period (Supplementary Figure 11b, RMSE = 1.2 mm/year, $R = 0.8$). The improved agreement for temporally matched data suggests that non-linear subsidence contributes to some of the scatter in the historical comparison, although the overall correlation remains strong in both cases.

• Page 21, line 820-822: I wonder how representative GNSS stations outside the deltas are. The conditions for InSAR and hence its uncertainties might be more challenging in deltas than in adjacent areas. This depends on vegetation, land surfaces changes during the observation time series, and even atmospheric conditions could be different in a delta than adjacent areas which might be more inland. While the validation gives valuable insights in the general performance of the InSAR algorithm, it does not necessarily yield exact uncertainties of the results within the deltas.

We thank the reviewer for this important point. First, we agree that uncertainty (the statistical precision of velocity estimates) and accuracy (the closeness to true values) are conceptually distinct. In our analysis, the formal uncertainties are derived at the pixel scale from regression residuals, LOS decomposition, and GNSS referencing, and directly reflect the noise characteristics within each delta scene. The comparison with GNSS stations serves as an independent assessment of systematic bias and overall accuracy, rather than as the source of our uncertainty estimates. We acknowledge that local factors in deltas (e.g., vegetation, land-cover change, atmospheric heterogeneity) may affect InSAR performance; these are already incorporated in the pixel-level uncertainties

through the time-series analysis and reported standard deviation values. Therefore, while GNSS validation provides confidence in the general accuracy of our results, the uncertainties reported in the manuscript are derived directly from the InSAR processing and not from GNSS validation. Additionally, greater than 90% of validation GNSS stations are located within or immediately adjacent to delta boundaries, providing validation in the actual environments of interest. The strong GNSS-InSAR correlation ($R = 0.7-0.8$) across diverse delta environments suggests that our results reliably capture ground motion in deltaic settings.

Conclusions

• Page 7, line 251-258 “However, we note some deltas deviate from this trend. [...] These outliers and deviations highlight the complex, nonlinear, and interwoven processes driving surface elevation change in deltaic systems”. You should discuss if these findings could also be caused by uncertainties in the analyzed parameters.

We thank the reviewer for this suggestion. We agree that uncertainties in the analyzed parameters may also contribute to observed deviations from expected trends. However, this section has now been removed for word limit considerations. Note that we comprehensively acknowledge uncertainties in all predictor variables in the limitations section.

• Page 9, lines 324-325: “ Importantly, no delta exhibits elevation change driven solely (100%) by one variable.” Is this something LIME could provide as a result? Is a FI score of 0.01 significant (Example Mekong Delta)?

We thank the reviewer for this question. Yes, LIME could theoretically assign 100% feature importance to a single variable if one predictor dominated the local approximation around that data point. However, we do not observe this in any of our 28 deltas with sufficient LIME fidelity, indicating that subsidence drivers are genuinely mixed rather than singular across all analyzed systems.

Regarding the statistical significance of small feature importance scores, LIME produces normalized importance values that sum to 1.0 across all predictors for each delta. A feature importance of 0.01 (1%) indicates that the variable contributes minimally to explaining subsidence in that delta compared to the other predictors. While small, these values are not arbitrary but rather reflect the local model's assessment of relative contributions. The small urban expansion score for the Mekong delta reflects that urbanization contributes relatively little to Mekong subsidence compared to groundwater extraction, consistent with previous delta-specific studies. We acknowledge that very small feature importance values approach the noise floor of the model, but our interpretation focuses on the dominant and secondary drivers rather than treating all non-zero values as equally meaningful.

• Page 11, lines 404-405 “These findings identify VLM as the principal hazard in deltaic systems and some subsidence-prone low-elevation coastal zones.” What do you mean by “some subsidence-prone low-elevation coastal zones”?

We thank the reviewer for requesting clarification. The phrase "some subsidence-prone low-elevation coastal zones" was intended to indicate that VLM dominates relative sea-level rise not universally across all low-elevation coasts, but specifically in those areas experiencing significant subsidence, which includes most deltas in our study. We have revised this sentence (line 404-405) to read: "These findings identify VLM as the principal hazard in deltaic systems and other subsidence-prone low-elevation coastal zones." This clarifies that we are referring to coastal areas where subsidence is actively occurring, rather than suggesting VLM is the principal hazard for all low-elevation coasts globally, regardless of subsidence conditions.

Suggested improvements

See above

References

The manuscript references the literature appropriately.

Clarity and context

The manuscript is well written and structured. The discussion has been provided with sufficient context and consideration of previous work.

Inflammatory material

-

Reviewer Limits

I was asked to take a particular look at the InSAR analysis for the review, as this is my main research field. This is why my review largely concerns the technical aspects of the InSAR analysis. I still have done my best to evaluate the other parts of the study, but I acknowledge limitations in my expertise relating to the machine learning analysis and the GRACE-derived groundwater storage change. Even though I was focusing on technical aspects, I still found the manuscript to be well written and structured, with a thorough data integration and important conclusions. I did not intend with my comments to question the overall approach and results, but to help clarify and strengthen certain aspects of the analysis.

Nils Dörr, Karlsruhe Institute of Technology

We thank you once again for your excellent reviews and positive comments on our manuscript.

Referee #5 (Remarks to the Author): (Reviewers comments in normal text, response to reviewers is in bold)

The authors satisfactorily addressed my suggestions and clarification requests. I support the publication of this manuscript.

We sincerely thank the Reviewer for their feedback.

Reviewer Report for “The Global Threat of Sinking Deltas” by *Ohenhen et al.*,

Key Results

This manuscript offers a comprehensive global evaluation of land subsidence in river deltas, utilizing high-resolution remote sensing data. It reveals that

- $\geq 65\%$ of the mapped delta area is experiencing subsidence. In 18 deltas, the average subsidence rate already surpasses the current geocentric sea-level rise.
- Groundwater storage loss emerges as the primary anthropogenic predictor of VLM in approximately 37% of deltas, while sediment flux decline and urban expansion play secondary, context-dependent roles.
- When subsidence is factored into IPCC AR6 SSP5-8.5 projections, the 95th percentile RSLR in many deltas by 2050–2100 exceeds the ocean-only signal by factors of 2–8, affecting around 236 million people.
- These synoptic figures, the 75 m resolution VLM maps, and the effort to partition anthropogenic drivers represent the manuscript’s main contributions.

Validity

The overall strategy—deriving delta-wide subsidence from Sentinel-1 InSAR and linking it to putative drivers—is conceptually sound and rests on well-established techniques. No fundamental flaw is evident, yet two substantive validity issues must be resolved.

- **Groundwater attribution**

The assertion that groundwater withdrawal is the primary anthropogenic influence in approximately 37% of deltas relies almost exclusively on GRACE/GRACE-FO trends, which track surface water and groundwater storage. The GRACE mascon footprints, spanning about 300–400 km (approximately 200,000 km²), are significantly larger than most delta plains, often resulting in signal leakage from neighboring aquifers and basins. Given that surface water and groundwater are hydraulically interconnected in many deltas, changes in total water storage also reflect variations in rivers, reservoirs, lakes, and wetlands. Without first subtracting the surface water component (e.g., rivers, wetlands, lakes, and reservoirs) using models that have these components like WGHM or existing in-situ data, and without local well data or pumping records for validation, **the connection between the decline in surface water storage plus groundwater storage and aquifer depletion remains speculative.** The manuscript should

- recover GRACE data at the native mascon resolution ~ 3 -deg,
- separate groundwater from surface-water signals, and
- verify at least a subset of deltas against in-situ water-level or extraction statistics.

Only then can subsidence be confidently attributed to groundwater extraction.

- **Natural subsidence omitted**

All deltas experience background “geologic” subsidence: Holocene sediment compaction (typically 1–2 mm yr⁻¹), glacial-isostatic adjustment (e.g. ~ 1 mm yr⁻¹ on the U.S. Gulf Coast), sediment-load flexure, and, in some settings, tectonic subsidence. For example, roughly one-fifth of the 5 mm yr⁻¹ total subsidence measured in the Mississippi Delta is expected from long-term GIA. **Excluding such natural terms can inflate the anthropogenic share and misallocate residual variance among groundwater, sediment flux, and urban load.** The authors should therefore:

- incorporate global GIA fields (e.g., (Caron et al., 2018)) and published estimates of long-term compaction (Yu et al., 2012) into their VLM budget, or
- clearly justify and document any assumption that natural rates are negligible or spatially uniform.

Until this baseline is accounted for, statements that subsidence is “largely human-caused” need a caveat: part of the observed sinking would occur even in the absence of contemporary human activity. Addressing these two issues will prevent over-interpretation of the anthropogenic contribution and greatly strengthen the manuscript’s validity.

- **Sea-level benchmark:**

Subsidence is compared with open-ocean altimetry. Coastal altimetry (geocentric) typically shows 20–50 % faster rise; thus, **VLM dominance may be exaggerated.** Authors need to incorporate coastal altimetry rather than open ocean altimetry, and contextualize their results with their the scope of the regional rates not the global

Cazenave A, Gouzenes Y, Birol F, Leger F, Passaro M, Calafat FM, Shaw A, Nino F, Legeais JF, Oelmann J, world’s coastlines can be measured by a network of virtual altimetry stations. Communications earth & enviro

- **Scenario choice**

Concentrating exclusively on SSP58.5 (previously RCP 8.5) emphasizes an extreme, upper-limit emissions pathway that already diverges from current trends and observations. Please include results for a midrange scenario—such as SSP24.5—to present readers with a more realistic spectrum of potential futures.

Data and Methodology

Beyond the GRACE-related issues previously mentioned, two additional methodological considerations warrant attention.

- The study employs a multilinear regression to associate subsidence rates with three anthropogenic drivers, subsequently using LIME (Local Interpretable Model-agnostic Explanations) to assess the significance of each driver. With only 40 deltas, any regression or ML model is susceptible to outliers and multicollinearity. The three predictors are likely interrelated—highly urbanized deltas often feature drained wetlands and upstream dams, for instance—resulting in potentially unstable standard coefficients. It is crucial to report variance inflation factors, cross-validated R^2 , and sensitivity to the exclusion of high-leverage deltas. These diagnostics are vital for establishing statistical robustness.
- The ternary diagram of LIME scores holds potential for insight, yet the overplotting of large bubbles hampers readability. Consider cleaner alternatives: plain points with labels, smaller symbols, or separate subplots for the scatter and ternary views. A simplified graphic would convey the attribution results far more effectively.

Reviewer Limits

As a reviewer, I have done my best to evaluate all aspects of this multidisciplinary study, but I acknowledge certain technical limits in my expertise and access:

- I am not an expert in the latest InSAR time-series algorithms. I cannot independently verify the specifics of the InSAR data processing (e.g., unwrapping errors, atmospheric correction algorithms, or how phase ambiguities were resolved) without access to the authors' methodology description or code. I trust the authors' expertise in this area (several co-authors are known for InSAR work) and assume standard, validated techniques were used.
- I have focused my review on the geoscientific reasoning, the integration of data, and the alignment of conclusions with evidence, while relying on the authors' expertise for the specialized technical execution in areas such as InSAR processing. Consequently, my feedback focuses more on interpretation rather than algorithmic validation.

Despite these limitations, I found the manuscript to be highly engaging and generally well-executed. My critique is intended to help clarify and strengthen the work, rather than to question its fundamental approach

Caron, L., et al., 2018. GIA model statistics for GRACE hydrology, cryosphere, and ocean science. Geophysical Research Letters. 45, 2203-2212.

Yu, S.-Y., et al., 2012. Quantifying Holocene lithospheric subsidence rates underneath the Mississippi Delta. Earth and Planetary Science Letters. 331-332, 21-30.